# GRADIENTS EXPLODE
# - DEEP NETWORKS ARE SHALLOW
# - RESNET EXPLAINED

**George Philipp**[*]
Carnegie Mellon University
`george.philipp@email.de`

**Dawn Song**
University of California, Berkeley
`dawnsong@gmail.com`

**Jaime G. Carbonell**
Carnegie Mellon University
`jgc@cs.cmu.edu`

## ABSTRACT

Whereas it is believed that techniques such as Adam, batch normalization and, more recently, SeLU nonlinearities "solve" the exploding gradient problem, we show that this is not the case in general and that in a range of popular MLP architectures, exploding gradients exist and that they limit the depth to which networks can be effectively trained, both in theory and in practice. We explain why exploding gradients occur and highlight the *collapsing domain problem*, which can arise in architectures that avoid exploding gradients.

ResNets have significantly lower gradients and thus can circumvent the exploding gradient problem, enabling the effective training of much deeper networks, which we show is a consequence of a surprising mathematical property. By noticing that *any neural network is a residual network*, we devise the *residual trick*, which reveals that introducing skip connections simplifies the network mathematically, and that this simplicity may be the major cause for their success.

## 1 INTRODUCTION

Arguably, the primary reason for the recent success of neural networks is their "depth", i.e. their ability to compose and jointly train nonlinear functions so that they co-adapt. A large body of work has detailed the benefits of depth (e.g. Montafur et al. (2014); Delalleau & Bengio (2011); Martens et al. (2013); Bianchini & Scarselli (2014); Shamir & Eldan (2015); Telgarsky (2015); Mhaskar & Shamir (2016)).

The exploding gradient problem has been a major challenge for training very deep feedforward neural networks at least since the advent of gradient-based parameter learning (Hochreiter, 1991). In a nutshell, it describes the phenomenon that as the gradient is backpropagated through the network, it may grow exponentially from layer to layer. This can, for example, make the application of vanilla SGD impossible for networks beyond a certain depth. Either the step size is too large for updates to lower layers to be useful or it is too small for updates to higher layers to be useful. While this intuitive notion is widely understood, there are important gaps in the foundational understanding of this phenomenon. In this paper, we take a significant step towards closing those gaps.

To begin with, there is no well-accepted metric for determining the presence of pathological exploding gradients. Should we care about the length of the gradient vector? Should we care about the size of individual components of the gradient vector? Should we care about the eigenvalues of the Jacobians of individual layers? Depending on the metric used, different strategies arise for combating exploding gradients. For example, manipulating the width of layers a suggested by e.g. Anonymous (2018d); Han et al. (2017) can greatly impact the size of gradient vector components but leaves the length of the gradient vector relatively unchanged.

---

[*]work done while at University of California, Berkeley

The underlying problem is that it is unknown whether exploding gradients according to any of these metrics necessarily lead to training difficulties. There is a large body of evidence that gradient explosion defined by some metric when paired with some optimization algorithm on some architectures and datasets is associated with poor results (e.g. Schoenholz et al. (2017); Yang & Schoenholz (2017)). But, can we make general statements about entire classes of algorithms and architectures?

Algorithms such as RMSprop (Tieleman & Hinton, 2012), Adam (Kingma & Ba, 2015) or vSGD (Schaul et al., 2013) are light modifications of SGD that rescale different parts of the gradient vector and are known to be able to lead to improved training outcomes. This raises an another important unanswered question. Are exploding gradients merely a numerical quirk to be overcome by simply rescaling different parts of the gradient vector or are they reflective of an inherently difficult optimization problem that cannot be easily tackled by simple modifications to a stock algorithm?

It has become a common notion that techniques such as introducing normalization layers (e.g. Ioffe & Szegedy (2015), Ba et al. (2016), Chunjie et al. (2017), Salimans & Kingma (2016)) or careful initial scaling of weights (e.g. He et al. (2015), Glorot & Bengio (2015), Saxe et al. (2014), Mishking & Matas (2016)) largely eliminate exploding gradients by stabilizing forward activations. This notion was espoused in landmark papers. The paper that introduced batch normalization (Ioffe & Szegedy, 2015) states:

*In traditional deep networks, too-high learning rate may result in the gradients that explode or vanish, as well as getting stuck in poor local minima. Batch Normalization helps address these issues.*

The paper that introduced ResNet (He et al., 2016b) states:

*Is learning better networks as easy as stacking more layers? An obstacle to answering this question was the notorious problem of vanishing/exploding gradients, which hamper convergence from the beginning. This problem, however, has been largely addressed by normalized initialization and intermediate normalization layers, ...*

We argue that these claims are overly optimistic. While scaling weights or normalizing forward activations can reduce gradient growth defined according to certain metrics in certain situations, these techniques are not effective in general and can cause other problems even when they are effective. We intend to add nuance to these ideas which have been widely adopted by the community (e.g. Chunjie et al. (2017); Balduzzi et al. (2017)). In particular, we intend to correct the misconception that stabilizing forward activations is sufficient for avoiding exploding gradients (e.g. Klambauer et al. (2017)).

ResNet (He et al., 2016b) and other neural network architectures utilizing skip connections (e.g. Huang et al. (2017), Szegedy et al. (2016)) have been highly successful recently. While the performance of networks without skip connections starts to degrade when depth is increased beyond a certain point, the performance of ResNet continues to improve until a much greater depth is reached. While favorable changes to properties of the gradient brought about by the introduction of skip connections have been demonstrated for specific architectures (e.g. Yang & Schoenholz (2017); Balduzzi et al. (2017)), a general explanation for the power of skip connections has not been given.

Our contributions are as follows:

1. We introduce the *'gradient scale coefficient' (GSC)*, a novel measurement for assessing the presence of pathological exploding gradients (section 2). It is robust to confounders such as network scaling (section 2) and layer width (section 3) and can be used directly to show that training is difficult (section 4). Therefore, we propose the unification of research on the exploding gradient problem under this metric.

2. We demonstrate that exploding gradients are in fact present in a variety of popular MLP architectures, including architectures utilizing techniques that supposedly combat exploding gradients. We show that introducing normalization layers may even exacerbate the exploding gradient problem (section 3).

3. We show that exploding gradients as defined by the GSC are not a numerical quirk to be overcome by rescaling different parts of the gradient vector, but are indicative of an inherently complex optimization problem and that they limit the depth to which MLP archi-

tectures can be effectively trained, rendering very deep MLPs effectively much shallower (section 4). To our knowledge, this is the first time such a link has been established.

4. For the first time, we show why exploding gradients are likely to occur in deep networks even when the forward activations do not explode (section 5). We argue that this is a fundamental reason for the difficulty of constructing very deep trainable networks.

5. For the first time, we define the *'collapsing domain problem'* for training very deep feed-forward networks. We show how this problem can arise precisely in architectures that avoid exploding gradients via careful initial scaling of weights and that it can be at least as damaging to the training process (section 6).

6. For the first time, we show that the introduction of skip connections has a strong gradient-reducing effect on deep network architectures in general. We detail the surprising mathematical relationship that makes this possible (section 7).

7. We introduce the *'residual trick'* (section 4), which reveals that ResNets are a mathematically simpler version of networks without skip connections and thus approximately achieve what we term the *'orthogonal initial state'*. This provides, we argue, the major reason for their superior performance at great depths as well as an important criterion for neural network design in general (section 7).

In section 8, we conclude and derive practical recommendations for designing and training deep networks as well as key implications of our work for deep learning research.

In the appendix in section B, we provide further high-level discussion. In section B.1, we discuss related work including the relationship of exploding gradients with other measures of network trainability, such as eigenspectrum analysis (Saxe et al., 2014), shattering gradients (Balduzzi et al., 2017), trajectory lengths (Raghu et al., 2017), covariate shift (e.g. (Ioffe & Szegedy, 2015)) and Hessian conditioning (e.g. (Luo, 2017)). Recently, the behavior of neural networks at great depth was analyzed using mean field theory (Poole et al., 2016; Schoenholz et al., 2017; Yang & Schoenholz, 2017; Anonymous, 2018d) and dynamical systems theory (Haber et al., 2017; Haber & Ruthotto, 2017; Chang et al., 2017; Anonymous, 2018a). We discuss these lines of work in relation to this paper in sections B.1.1 and B.1.2 respectively. We discuss the implications of our work for the vanishing gradient problem in section B.2. We compare the exploding gradient problem as it occurs in feedforward networks to the exploding and vanishing gradient problems in RNNs (e.g. Pascanu et al. (2013)) in section B.3. In section B.4, we highlight open research questions and potential future work.

## 2 Exploding gradients defined - the gradient scale coefficient

### 2.1 Notation and terminology

For the purpose of this paper, we define a neural network $f$ as a succession of layers $f_l$, $0 \leq l \leq L$, where each layer is a vector-to-vector transformation. We assume a prediction framework, where the 'prediction layer' $f_1$ is considered to output the prediction of the network and the goal is to minimize the value of the error layer $f_0$ over the network's prediction and the true label $y$, summed over some dataset $D$.

$$\arg \min_{\theta} E, \text{ where } E = \frac{1}{|D|} \sum_{(x,y) \in D} f_0(y, f_1(\theta_1, f_2(\theta_2, f_3(..f_L(\theta_L, x)..)))) \tag{1}$$

Note that in contrast to standard notation, we denote by $f_L$ the lowest layer and by $f_0$ the highest layer of the network as we are primarily interested in the direction of gradient flow. Let the dimensionality / width of layer $l$ be $d_l$ with $d_0 = 1$ and the dimensionality of the data input $x$ be $d$.

Each layer except $f_0$ is associated with a parameter sub-vector $\theta_l$ that collectively make up the parameter vector $\theta = (\theta_1, .., \theta_L)$. This vector represents the trainable elements of the network. Depending on the type of the layer, the sub-vector might be empty. For example, a layer composed of tanh nonlinearities has no trainable elements, so its parameter sub-vector is empty. We call these

layers 'unparametrized'. In contrast, a fully-connected linear layer has trainable weights, which are encompassed in the parameter sub-vector. We call these layers 'parametrized'.

We say a network that has layers $f_0$ through $f_L$ has 'nominal depth' $L$. In contrast, we say the 'compositional depth' is equal to the number of parametrized layers in the network, which is the quantity that is commonly referred to as "depth". For example, a network composed of three linear layers, two tanh layers and a softmax layer has nominal depth 6, but compositional depth 3.

Let the 'quadratic expectation' $\mathbb{Q}$ of a random variable $X$ be defined as $\mathbb{Q}[X] = \mathbb{E}[X^2]^{\frac{1}{2}}$, i.e. the generalization of the quadratic mean to random variables. Similarly, let the 'inverse quadratic expectation' $\mathbb{Q}^{-1}$ of a random variable $X$ be defined as $\mathbb{Q}[X] = \mathbb{E}[X^{-2}]^{-\frac{1}{2}}$. Further terminology, notation and conventions used only in the appendix are given in section C.

## 2.2 Colloquial notion of exploding gradients

Colloquially, the exploding gradient problem is understood approximately as follows:

*When the error is backpropagated through a neural network, it may increase exponentially from layer to layer. In those cases, the gradient with respect to the parameters in lower layers may be exponentially greater than the gradient with respect to parameters in higher layers. This makes the network hard to train if it is sufficiently deep.*

Let $\mathcal{J}_k^l(\theta, x, y)$ be the Jacobian of the $l$'th layer $f_l$ with respect to the $k$'th layer $f_k$ evaluated with parameter $\theta$ at $(x, y)$, where $0 \leq l \leq k \leq L$. Similarly, let $\mathcal{T}_k^l(\theta, x, y)$ be the Jacobian of the $l$'th layer $f_l$ with respect to the parameter sub-vector of the $k$'th layer $\theta_k$. Then we might take this colloquial notion to mean that if $||\mathcal{J}_k^l||$ and / or $||\mathcal{T}_k^l||$ grow exponentially in $k - l$, according to some to-be-determined norm $||.||$, the network is hard to train if it is sufficiently deep.

However, this notion is insufficient because we can construct networks that can be trained successfully yet have Jacobians that grow exponentially at arbitrary rates. In a nutshell, all we have to do to construct such a network is to take an arbitrary network of desired depth that can be trained successfully and scale each layer function $f_l$ and each parameter sub-vector $\theta_l$ by $R^{-l}$ for some constant $R > 1$. During training, all we have to do to correct for this change is to scale the gradient sub-vector corresponding to each layer by $R^{-2l}$.

**Proposition 1.** *Consider any $r > 1$ and any neural network $f$ which can be trained to some error level in a certain number of steps by some gradient-based algorithm. There exists a network $f'$ that can also be trained to the same error level as $f$ and to make the same predictions as $f$ in the same number of steps by the same algorithm, and has exponentially growing Jacobians with rate $r$. (See section E.1 for details.)*

Therefore, we need a definition of 'exploding gradients' different from 'exponentially growing Jacobians' if we hope to derive from it that training is intrinsically hard and not just a numerical issue to be overcome by gradient rescaling.

Note that all propositions and theorems are stated informally in the main body of the paper, for the purpose of readability and brevity. In the appendix in sections E and F respectively, they are re-stated in rigorous terms, proofs are provided and the practicality of conditions is discussed.

## 2.3 The gradient scale coefficient

In this section, we outline our definition of 'exploding gradients' which can be used to show hardness of training. It does not suffer from the confounding effect outlined in the previous section.

**Definition 1.** Let the 'quadratic mean norm' or 'qm norm' of an $m \times n$ matrix $A$ be the quadratic mean of its singular values where the sum of squares is divided by its right dimension $n$. If $s_1$, $s_2$, .., $s_{\min(m,n)}$ are the singular values of $A$, we write:

$$||A||_{qm} = \sqrt{\frac{s_1^2 + s_2^2 + .. + s_{\min(m,n)}^2}{n}}$$

An equivalent definition would be $||A||_{qm} = \mathbb{Q}_u ||Au||_2$, where $u$ is a uniformly random unit length vector. In plain language, it measures the expected impact the matrix has on the length of a vector with uniformly random orientation. The qm norm is closely related to the $L_2$ norm via $\sqrt{n}||A||_{qm} = ||A||_2$. We use $||.||_2$ to denote the $L_2$ norm of both vectors and matrices.

**Definition 2.** Let the *'gradient scale coefficient (GSC)'* for $0 \le l \le k \le L$ be as follows:

$$GSC(k, l, f, \theta, x, y) = \frac{||\mathcal{J}_k^l||_{qm}^2 ||f_k||_2^2}{||f_l||_2^2}$$

**Definition 3.** We say that the network $f(\theta)$ has 'exploding gradients with rate $r$ and intercept $c$' at some point $(x, y)$ if for all $k$ and $l$ we have $GSC(k, l, f, \theta, x, y) \ge cr^{k-l}$, and in particular $GSC(l, 0, f, \theta, x, y) \ge cr^l$.

Of course, under this definition, any network of finite depth has exploding gradients for sufficiently small $c$ and $r$. There is no objective threshold for $c$ and $r$ beyond which exploding gradients become pathological. Informally, we will say that a network has 'exploding gradients' if the GSC can be well-approximated by an exponential function.

The GSC combines the norm of the Jacobian with the ratio of the lengths of the forward activation vectors. In plain language, it measures the size of the gradient flowing backward relative to the size of the activations flowing forward. Equivalently, it measures the relative sensitivity of layer $l$ with respect to small random changes in layer $k$.

**Proposition 2.** $GSC(k, l)$ *measures the quadratic expectation of the relative size of the change in the value of $f_l$ in response to a small random change in $f_k$. (See section E.2 for details.)*

What about the sensitivity of layers with respect to the parameter? For fully-connected linear layers, we obtain a similar relationship.

**Proposition 3.** *When $f_k$ is a fully-connected linear layer without trainable bias parameters and $\theta_k$ contains the entries of the weight matrix, $GSC(k, l) \frac{||\theta_k||_2 ||f_{k+1}||_2}{||f_k||_2 \sqrt{d_{k+1}}}$ measures the quadratic expectation of the relative size of the change in the value of $f_l$ in response to a small random change in $\theta_k$. Further, if the weight matrix is randomly initialized, $\mathbb{Q}_{\theta_k}^{-1} \frac{||\theta_k||_2 ||f_{k+1}||_2}{||f_k||_2 \sqrt{d_{k+1}}} = 1$. (See section E.3 for details.)*

For reasons of space and mathematical simplicity, we focus our analysis for now on multi-layer perceptrons (MLPs) which are comprised only of fully-connected linear layers with no trainable bias parameters, and unparametrized layers. Therefore we also do not use trainable bias and variance parameters in the normalization layers. Note that using very deep MLPs with some architectural limitations as a testbed to advance the study of exploding gradients and related problems is a well-established practice (e.g. Balduzzi et al. (2017); Yang & Schoenholz (2017); Raghu et al. (2017)). As Schoenholz et al. (2017), we focus on training error rather than test error in our analysis as we do not consider the issue of generalization. While exploding gradients have important implications for generalization, this goes beyond the scope of this paper.

In section 2.2, we showed that we can construct trainable networks with exponentially growing Jacobians by simple multiplicative rescaling of layers, parameters and gradients. Crucially, the GSC is invariant to this rescaling as it affects both the forward activations and the Jacobian equally, so the effects cancel out.

**Proposition 4.** $GSC(k, l)$ *is invariant under multiplicative rescalings of the network that do not change the predictions or error values of the network. (See section E.4 for details.)*

## 3 GRADIENTS EXPLODE - DESPITE BOUNDED ACTIVATIONS

In this section, we show that exploding gradients exist in a range of popular MLP architectures. Consider the decomposability of the GSC.

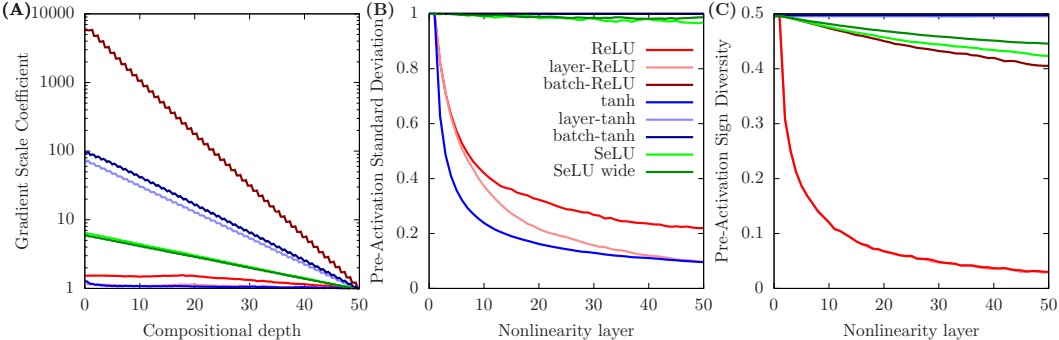

Figure 1: Key metrics for architectures in their randomly initialized state evaluated on Gaussian noise. The x axis in the left graph shows depth in terms of the number of linear layers counted from the input. Note: The curve for ReLU-layer is shadowed by tanh in the figure A and by ReLU in figure C.

**Proposition 5.** *Assuming the approximate decomposability of the norm of the product of Jacobians, i.e.* $||\mathcal{J}_{l+1}^l \mathcal{J}_{l+2}^{l+1}..\mathcal{J}_k^{k-1}||_{qm} \approx ||\mathcal{J}_{l+1}^l||_{qm}||\mathcal{J}_{l+2}^{l+1}||_{qm}..||\mathcal{J}_k^{k-1}||_{qm}$, *we have* $GSC(k,l) \approx GSC(k,k-1)GSC(k-1,k-2)..GSC(l+1,l)$. *(See section E.5 for the proof.)*

This suggests that as long as the GSC of individual layers is approximately $r > 1$, we may have an exponential growth of $GSC(k,l)$ in $k - l$. In figure 1A, we show $GSC(l, 0)$ for seven MLP architectures. A linear layer is followed by (i) a ReLU nonlinearity ('ReLU'), (ii) layer normalization (Ba et al., 2016) followed by a ReLU nonlinearity ('layer-ReLU'), (iii) batch normalization plus ReLU ('batch-ReLU'), (iv) tanh, (v) layer norm plus tanh ('layer-tanh'), (vi) batch norm plus tanh ('batch-tanh'), (vii) SeLU Klambauer et al. (2017). All networks have compositional depth 50 (i.e. 50 linear layers) and each layer has 100 neurons. Both data input and labels are Gaussian noise and the error layer computes the dot product between the label and the prediction. The entries of weight matrices are dawn from independent Gaussian distributions with mean zero. Weight matrix entries for ReLU architectures are initialized with variance $\frac{2}{100}$ as suggested by He et al. (2015), weight matrix entries for tanh architectures with variance $\frac{1}{100}$ as suggested by Saxe et al. (2014) and Glorot & Bengio (2015), and weight matrix entries for SeLU architectures with variance $\frac{1}{100}$ as suggested by Klambauer et al. (2017). For further experimental details, see section I.

We find that in four architectures (batch-ReLU, layer-tanh, batch-tanh and SeLU), $GSC(l, 0)$ grows almost perfectly linearly in log-space. This corresponds to gradient explosion. We call those architectures 'exploding architectures'. Among these architectures, a range of techniques that supposedly reduce or eliminate exploding gradients are used: careful initial scaling of weights, normalization layers, SeLU nonlinearities. Adding normalization layers may even bring about or exacerbate exploding gradients. The exploding architectures have all been designed to have stable forward activations and they exhibit gradient explosion under any reasonable metric.

In light of proposition 4, it is not surprising that these techniques are not effective in general at combating exploding gradients as defined by the GSC, as this metric is invariant under multiplicative rescaling. Normalization layers are used to scale the activations. Carefully choosing the initial scale of weights corresponds to a multiplicative scaling of weights. SeLU nonlinearities, again, act to scale down large activations and scale up small activations. While these techniques may of course impact the GSC by changing the fundamental mathematical properties of the network (as can be seen, for example, when comparing ReLU and batch-ReLU), they do not reduce it simply by virtue of controlling the size of forward activations.

In contrast, the other three architectures (ReLU, layer-ReLU and tanh) do not exhibit exploding gradients. However, this apparent advantage comes at a cost, as we further explain in section 5.

All curves in figure 1A exhibit small jitters. This is because we plotted the value of the GSC at every linear layer, every normalization layer and every nonlinearity layer in the graph and then connected the points corresponding to these values. Layers were placed equispaced on the x axis in the order they occurred in the network. Not every type of layer affects the GSC equally. In

fact, we find that as gradients pass through linear layers, they tend to shrink relative to forward activations. In the exploding architectures, this is more than counterbalanced by the relative increase the gradient experience as it passes through e.g. normalization layers. Despite these layer-dependent differences, it is worth noting that each individual layer used in the architectures studied has only a small impact on the GSC. This would not be true for either the forward activations or gradients taken by themselves. For example, passing through a ReLU layer reduces the length of both activation and gradient vector by $\approx \sqrt{2}$. This relative invariance to individual layers suggests that the GSC measures not just a superficial quantity, but a deep property of the network. This hypothesis is confirmed in the following sections.

Finally, we note that the GSC is also robust to changes in width and depth. Changing the depth has no impact on the rate of explosion of the four exploding architectures as the layer-wise GSC, i.e. $GSC(l+1, l)$, is itself independent of depth. In figure 1A, we also show the results for the SeLU architecture where each layer contains 200 neurons instead of 100 ('SeLU wide'). We found that the rate of gradient explosion decreases slightly when width increases. We also studied networks with exploding architectures where the width oscillated from layer to layer. $GSC(k, 0)$ still increased approximately exponentially and at a similar rate to corresponding networks with constant width.

A summary of results can be found in table 1.

## 4 EXPLODING GRADIENTS LIMIT DEPTH - THE RESIDUAL TRICK

### 4.1 BACKGROUND: EFFECTIVE DEPTH

In this section, we introduce the concept of 'effective depth' as defined for the ResNet architecture by Veit et al. (2016). We denote a residual network by writing each layer $f_l$ (except $f_0$) as the sum of a fixed initial function $i_l$ and a residual function $r_l$. We define the optimization problem for a residual net analogously to equation 1.

$$\arg \min_{\theta} E, \text{ where } E = \frac{1}{|D|} \sum_{(x,y) \in D} f_0(y, (i_1 + r_1(\theta_1)) \circ (i_2 + r_2(\theta_2)) \circ .. \circ (i_L + r_L(\theta_L)) \circ x) \quad (2)$$

Let's assume for the sake of simplicity that the dimension of each layer is identical. In that case, the initial function for ResNet is generally chosen to be the identity function. Then, writing the identity matrix as $I$, we have

$$\frac{df_0}{dx} = \frac{df_0}{df_1}(I + \frac{dr_1}{df_2})(I + \frac{dr_2}{df_3})..(I + \frac{dr_{L-1}}{df_L})(I + \frac{dr_L}{dx})$$

Multiplying out, this becomes the sum of $2^L$ terms. Almost all of those terms are the product of approximately $\frac{L}{2}$ identity matrices and $\frac{L}{2}$ residual Jacobians. However, if the operator norm of the residual Jacobians is less than $p$ for some $p < 1$, the norm of terms decreases exponentially in the number of residual Jacobians they contain. Let the terms in $\frac{df_0}{dx}$ containing $\lambda$ or more residual Jacobians be called '$\lambda$-residual' and let $res^\lambda$ be the sum of all $\lambda$-residual terms. Then:

$$||res^\lambda||_2 \leq ||\frac{df_0}{df_1}||_2 \sum_{l=\lambda}^{L} p^l \binom{L}{l}$$

Again, if $p < 1$, the right hand side decreases exponentially in $\lambda$ for sufficiently large $\lambda$, for example when $\lambda > \frac{L}{2}$, so the combined size of $\lambda$-residual terms is exponentially small. Therefore, Veit et al. (2016) argue, the full set of network layers does not jointly co-adapt during training because the information necessary for such co-adaption is contained in terms that contain many or all residual Jacobians. Only sets of layers of size at most $\lambda$ where $res^\lambda$ is not negligably small co-adapt. The largest such $\lambda$ is called the 'effective depth' of the network. Veit et al. (2016) argue that if the effective depth is less than the compositional depth of a residual network, the network is not really as deep as it appears, but rather behaves as an ensemble of relatively shallow networks. This argument

is bolstered by the success of the stochastic depth (Huang et al., 2016) training technique, where random sets of residual functions are deleted for each mini-batch update.

Veit et al. (2016) introduced the concept of effective depth somewhat informally. We give our formal definition in section D. There, we also provide a more detailed discussion of the concept and point out limitations.

## 4.2 THE RESIDUAL TRICK

Now we make a crucial observation. Any neural network can be expressed as a residual network as defined in 2. We can simply choose arbitrary initial functions $i_l$ and define $r_l(\theta_l) := f_l(\theta_l) - i_l$. Specifically, if we train a network $f$ from some fixed initial parameter $\theta^{(0)}$, we can set $i_l := f_l(\theta_l^{(0)})$ and thus $r_l(\theta_l) := f_l(\theta_l) - f_l(\theta_l^{(0)})$. Then training begins with all residual functions being zero functions. Therefore, all analysis devised for ResNet that relies on the small size of the residual Jacobians can then be brought to bear on arbitrary networks. We term this the *'residual trick'*. Indeed, the analysis by Veit et al. (2016) does not rely on the network having skip connections in the computational sense, but only on the mathematical framework of equation 2. Therefore, as long as the operator norms of $\frac{df_l(\theta_l)}{df_{l+1}} - \frac{df_l(\theta_l^{(0)})}{df_{l+1}}$ are small, $f$ is effectively shallow.

**Terminology** From now on, we will make a distinction between the terms 'ResNet' and 'residual network'. The former will be used to refer to networks that have an architecture as in He et al. (2016b) that uses skip connections. The latter will be used to refer to arbitrary networks expressed in the framework of equation 2. Networks without skip connections will be referred to as 'vanilla networks'.

## 4.3 THEORETICAL ANALYSIS

In this section, we will show that an exploding gradient as defined by the GSC causes the effective training time of deep MLPs to be exponential in depth and thus limits the effective depth that can be achieved.

The proof is based on the insight that the relative size of a gradient-based update $\Delta\theta_l$ on $\theta_l$ is bounded by the inverse of the GSC if that update is to be useful. The basic assumption underlying gradient-based optimization is that the function optimized is locally approximated by a linear function as indicated by the gradient. Any update made based on a local gradient computation must be small enough so that the updated value lies in the region around the original value where the linear approximation is sufficiently accurate. Let's assume we apply a random update to $\theta_l$ with relative magnitude $\frac{1}{GSC(0,l)}$, i.e. $\frac{||\Delta\theta_l||_2}{||\theta_l||_2} = \frac{1}{GSC(0,l)}$. Then under the local linear approximation, according to proposition 3, this would change the output $f_0$ approximately by a value with quadratic expectation $f_0$. Hence, with significant probability, the error would become negative. This is not reflective of the true behavior of the function $f_0$ in response to changes in $\theta_l$ of this magnitude. Since gradient-based updates impact the function value even more than random updates, useful gradient-based updates are even more likely to be bounded in relative magnitude by $\frac{1}{GSC(0,l)}$.

In a nutshell, if $\frac{1}{GSC(0,l)}$ decreases exponentially in $l$, so must the relative size of updates. So for a residual function to reach a certain size relative to the corresponding initial function, an exponential number of updates is required. But to reach a certain effective depth, a certain magnitude of $\lambda$-residual terms is required and thus a certain magnitude of residual functions relative to corresponding initial functions is required, and thus exponentially many updates.

**Theorem 1.** *Under certain conditions, if an MLP has exploding gradients with explosion rate $r$ and intercept $c$ on some dataset, then there exists a constant $c'$ such that training this neural network with a gradient-based algorithm to have effective depth $\lambda$ takes at least $c'cr^{\frac{\lambda}{4}}$ updates. (See section F.1 for details.)*

Importantly, the lower bound on the number of updates required to reach a certain effective depth stated by theorem 1 is independent of the nominal depth of the network. While the constant $c'$ depends on some constants that arise in the conditions of the theorem, as long as those constants do not change when depth is increased, neither does the lower bound.

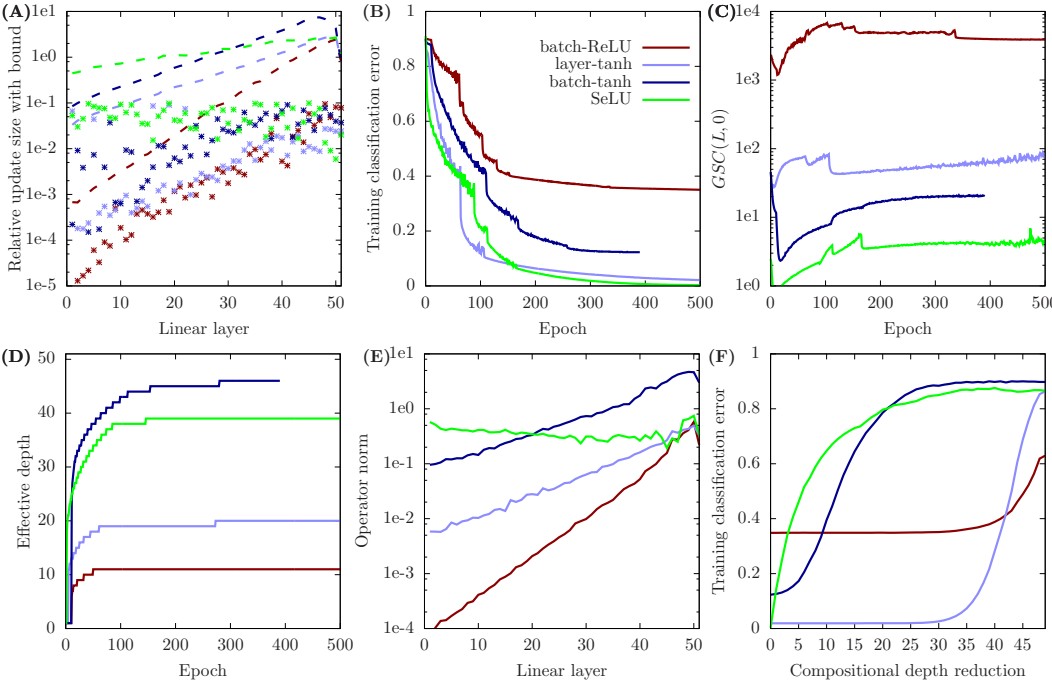

Figure 2: Key metrics for architectures trained on CIFAR10. See main text for explanation.

**Corollary 1.** *In the scenario of theorem 1, if the number of updates to convergence is bounded, so is effective depth.*

Here we simply state that if we reach convergence after a certain number of updates, but theorem 1 indicates that more would be required to attain a greater effective depth, then that greater effective depth is unreachable with that algorithm.

### 4.4 EXPERIMENTS

To practically validate our theory of limited effective depth, we train our four exploding architectures (batch-ReLU, layer-tanh, batch-tanh and SeLU) on CIFAR10. All networks studied have a compositional depth of 51 (i.e. 51 linear layers) and 100 neurons in each layer except for the input, prediction and error layers. Full experimental details can be found in section I.

First, we determined the approximate best step size for SGD for each individual linear layer. We started by pre-training the highest layers of each network with a small uniform step size until the training classification error was below 85%, but at most for 10 epochs. Then, for each linear layer, we trained only that layer for 1 epoch with various step sizes while freezing the other layers. The step size that achieved the lowest training classification error after that epoch was selected. Note that we only considered step sizes that induce relative update sizes of 0.1 or less, because larger updates can cause weight instability. The full algorithm for step size selection and a justification is given in section I.4.

In figure 2A, we show the relative update size induced on each linear layer by what was selected to be the best step size as well as $\frac{1}{GSC(l,0)}$ as a dashed line. In section 4.3, we argued that $\frac{1}{GSC(l,0)}$ is an upper bound for the relative size of a useful update. We find that this bound holds and is conservative except for a small number of outliers. Even though our algorithm for determining the best step size for each layer gives noisy results, there is a clear trend that lower layers require relatively smaller updates, and that this effect is more pronounced if the gradient explodes with a larger rate. Therefore the foundational assumption underlying theorem 1 holds.

We then smoothed these best step size estimates and trained each network for 500 epochs with those smoothed estimates. Periodically, we scaled all step sizes jointly by $\frac{1}{3}$. In figure 2B, we show the

training classification error of each architecture. There is a trend that architectures with less gradient explosion attain a lower final error. Note that, of course, all these error values are still much higher than the state of the art on CIFAR10. This is not a drawback however, as the goal of this section is to study and understand pathological architectures rather than find optimal ones. Those architectures, by definition, attain high errors.

In figure 2C, we show the GSC across the entire network, i.e. $GSC(L, 0)$, as training progresses. During the initial pre-training phase, this value drops significantly but later regains or even exceeds its original value. In figure 2A, the dashed line indicates the inverse of $GSC(l, 0)$ for each $l$ after pre-training. We find that the GSC actually falls below 1 as the gradient passes through the pre-trained layers, but then resumes explosion once it reached the layers that were not pre-trained. We find this behavior surprising and unexpected. We conclude that nonstandard training procedures can have a significant impact on the GSC but that there is no evidence that when all layers are trained jointly, which is the norm, the GSC either significantly increases or decreases during training.

We then went on to measure the effective depth of each network. We devised a conservative, computationally tractable estimate of the cumulative size of updates that stem from $\lambda$-residual terms. See section D.2 for details. The effective depth depicted in figure 2D is the largest value of $\lambda$ such that this estimate has a length exceeding $10^{-6}$. As expected, none of the architectures reach an effective depth equal to their compositional depth, and there is a trend that architectures that use relatively smaller updates achieve a lower effective depth. It is worth noting that the effective depth increases most sharply at the beginning of training. Once all step sizes have been multiplied by $\frac{1}{3}$ several times, effective depth no longer changes significantly while the error, on the other hand, is still going down. This suggests that, somewhat surprisingly, high-order co-adaption of layers takes place towards the beginning of training and that as the step size is reduced, layers are fine-tuned relatively independently of each other.

SeLU and especially tanh-batch reach an effective depth close to their compositional depth according to our estimate. In figure 2E, we show the operator norm of the residual weight matrices after training. All architectures except SeLU, which has a $GSC(L, 0)$ close to 1 after pre-training, show a clear downward trend in the direction away from the error layer. If this trend were to continue for networks that have a much greater compositional depth, then those networks would not achieve an effective depth significantly greater than our 51-linear layer networks.

Veit et al. (2016) argue that a limited effective depth indicates a lack of high-order co-adaptation. We wanted to verify that our networks, especially layer-tanh and batch-ReLU, indeed lack these high-order co-adaptations by using a strategy independent of the concept of effective depth to measure this effect. We used Taylor expansions to do this. Specifically, we replaced the bottom $k$ layers of the fully-trained networks by their first-order Taylor expansion around the initial functions. See section G for how this is done. This reduces the compositional depth of the network by $k - 2$. In figure 2F, we show the training classification error in response to compositional depth reduction. We find that the compositional depth of layer-tanh and batch-ReLU can be reduced enormously without suffering a significant increase in error. In fact, the resulting layer-tanh network of compositional depth 15 greatly outperforms the original batch-tanh and batch-ReLU networks. This confirms that these networks lack high-order co-adaptations. Note that cutting the depth by using the Taylor expansion not only eliminates high-order co-adaptions among layers, but also co-adaptions of groups of 3 or more layers among the bottom $k$ layers. Hence, we expect the increase in error induced by removing only high-order co-adaptions to be even lower than what is shown in figure 2F. Unfortunately, this cannot be tractably computed.

Finally, we trained each of the exploding architectures by using only a single step size for each layer that was determined by grid search, instead of custom layer-wise step sizes. As expected, the final error was higher. The results are found in table 2.

**Summary**  For the first time, we established a direct link between exploding gradients and severe training difficulties that cannot be overcome by gradient rescaling. These difficulties arise in MLPs composed of the most popular layer types, even those that utilize techniques that stabilize forward activations which are believed to combat exploding gradients. The gradient scale coefficient not only underpins this analysis, but is largely invariant to the confounders of network scaling (proposition 4), layer width and individual layers (section 3). Therefore we argue the GSC is the best metric for the study of exploding gradients in general.

4.5   A NOTE ON BATCH NORMALIZATION

We used minibatches of size 1000 to train all architectures except batch-ReLU, for which we conducted full-batch training. When minibatches were used on batch-ReLU, the training classification error stayed above 89% throughout training. (Random guessing achieves a 90% error.) In essence, no learning took place. This is because of the pathological interplay between exploding gradients and the noise inherent in batch normalization. Under batch normalization, the activations at a neuron are normalized by their mean and standard deviation. These values are estimated using the current batch. Hence, if a minibatch has size $b$, we expect the noise induced by this process to have relative size $\approx \frac{1}{\sqrt{b}}$. But we know that according to proposition 2, under the local linear approximation, this noise leads to a change in the error layer of relative size $\approx \frac{GSC}{\sqrt{b}}$. Hence, if the GSC between the error layer and the first batch normalization layer is larger than $\sqrt{b}$, learning should be seriously impaired. For the batch-ReLU architecture, this condition was satisfied and consequently, the architecture was untrainable using minibatches. Ironically, the gradient explosion that renders the noise pathological was introduced in the first place by adding batch normalization layers. Note that techniques exist to reduce the dependence of batch normalization on the current minibatch, such as using running averages (Ioffe, 2017). Other prominent techniques that induce noise and thus can cause problems in conjunction with large gradients are dropout Srivastava et al. (2014), stochastic nonlinearities (e.g. Gulcehre et al. (2016)) and network quantization (e.g. (Anonymous, 2018c)).

5   WHY GRADIENTS EXPLODE - DETERMINANTS VS QM NORM

Why do exploding gradients occur? As mentioned in section 3, gradients explode with rate $r > 1$ as long as we have (A) $GSC(k,l) \approx GSC(l+1,1)GSC(l+2,l+1)..GSC(k,k-1)$ and (B) $GSC(l+1,1) \approx r$ for all $k$ and $l$. It turns out that we can show both of these hold in expectation under fairly realistic conditions if we view the network parameter as a random variable.

**Theorem 2.** *Under certain conditions, for any neural network $f$ with random parameter $\theta$ composed of layer functions $f_l$ that are surjective endomorphisms on the hypersphere, where the absolute singular values of the Jacobian of each layer are IID and differ by at least $\epsilon$ with probability $\delta$, gradients explode in expectation with rate $r(\delta, \epsilon)$. (See section F.2 for details.)*

*Proof Summary.* Consider a surjective endomorphism $f_l$ on the hypersphere and a random input $x$ distributed uniformly on that hypersphere. Surjectivity implies that the absolute determinant of the Jacobian, in expectation over the input, is at least 1. The absolute determinant is the product of the absolute singular values. If those absolute singular values are IID and their expected product is at least 1, the expectation of each absolute singular value is also at least 1. So if these singular values are sufficently different from each other with sufficient probability, the expected quadratic mean of the singular values is at least $r > 1$. Since both input and output of $f_l$ have length 1, the expected GSC will also be at least $r$.

□

While the conditions of theorem 2 cannot be fulfilled exactly, it nevertheless reveals an important insight. Exploding gradients tend to arise in practice even when forward activations are stable because in order to preserve the domain of the forward activations from layer to layer, Jacobians of individual layers need to have unit absolute determinants in expectation, and this tends to cause their qm norm values to be greater than 1, and then these values tend to compound exponentially.[1] The theorem is stated for layers that are surjective endomorphisms on the hypersphere. Let 'length-only layer normalization' (LOlayer) be a function that divides its input vector by its length. Then a sequence of a tanh / SeLU layer, a linear layer and a LOlayer layer, all of the same width, when viewed as a single macro-layer, is a surjective endomorphism on the hypersphere, as shown in proposition 6. Consequently, both LOlayer-tanh and LOlayer-SeLU exhibit exploding gradients (table 1). Layer-tanh and SeLU explode at very similar rates to LOlayer-tanh and LOlayer-SeLU respectively.

---

[1]It is possible to define probability distributions over matrices where the expected determinant is 1 and the expected qm norm is less than 1, though these examples are contrived and tend not to occur in practice.

**Proposition 6.** *Any endomorphism on the hypersphere composed of (i) a strictly monotonic, continuous nonlinearity $\sigma$ that has $\sigma(0) = 0$, (ii) multiplication with a full-rank matrix and (iii) length-only layer normalization is bijective. (See section E.6 for the proof.)*

Theorem 2 presents two clear avenues for avoiding exploding gradients: (i) use non-surjective layer functions, (ii) ensure that Jacobians get progressively closer to multiples of orthogonal matrices as we go deeper. It turns out that these are exactly the strategies employed by ReLU and ResNet respectively to avoid exploding gradients, and we will discuss these in the next two sections.

## 6 THE COLLAPSING DOMAIN PROBLEM - HOW NOT TO REDUCE GRADIENTS

In the previous section, we showed how surjective endomorphisms can exhibit exploding gradients. This suggests that we can avoid exploding gradients by non-surjectivity, i.e. if we reduce the domain of the forward activations from layer to layer. Informally, this can be understood as follows. Consider some layer function $f_l$. If we shrink its co-domain by a factor $c$, we reduce the eigenvalues of the Jacobian and hence its qm norm by $c$. If we also ensure that the length of the output stays the same, the $GSC$ is also reduced by $c$. Similarly, inflating the co-domain would cause the qm norm to increase.

This suggests that in neural network design, we can actively trade off exploding gradients and shrinkage of the domain and that eliminating one effect may exacerbate the other. This is precisely what we find in practice. Returning to figure 1, we now turn our attention to the middle graph (1B). Here, we plot the standard deviation of the activation values at each neuron across datapoints in the layers before each nonlinearity layer ('pre-activations'), averaged over all neurons in the same layer. The four exploding architectures exhibit a near constant standard deviation, whereas the other three architectures (ReLU, layer-ReLU and tanh) exhibit a rapidly collapsing standard deviation, which shows that the activations corresponding to different datapoints become more and more similar with depth. We term a layer-to-layer shrinkage of the domain the *'collapsing domain problem'*. But why is this effect a problem? Two reasons.

**Collapsing depth causes pseudo-linearity**  If the pre-activations that are fed into a nonlinearity are highly similar, the nonlinearity can be well-approximated by a linear function. In the tanh architecture we studied, for example, activation values become smaller and smaller as they are propagated forward. If the pre-activations of a tanh nonlinearity have small magnitude, the tanh nonlinearity can be approximated by the identity function. But if a tanh layer is approximately equal to an identity layer, the entire network becomes equivalent to a linear network. We say the network becomes 'pseudo-linear'. Of course, linear networks of any depth have the representational capacity of a linear network of depth 1 and are unable to model nonlinear functions. Hence, a tanh network that is pseudo-linear beyond compositional depth $k$ approximately has the representational capacity of a compositional depth $k + 1$ tanh network. Based on the decrease in pre-activation standard deviation exhibited by the tanh architecture in figure 1B, a reasonable estimate is that the network of compositional depth 50 has the representational capacity of a network of compositional depth 10.

Similarly, for a ReLU nonlinearity, if either all or most pre-activations are positive or all or most pre-activations are negative, the nonlinearity can be approximated by a linear function. If all or most pre-activations are positive, ReLU can be approximated by the identity function. If all or most pre-activations are negative, ReLU can be approximated by the zero function. In figure 1C, we plot the proportion of pre-activations for each neuron in a nonlinearity layer that are positive or negative, whichever is smaller, averaged over each layer. We call this metric 'sign diversity'. For both ReLU and layer-ReLU, sign diversity decreases rapidly to cause pseudo-linearity from at least, say, the 20th linear layer onwards. None of the four exploding architectures suffers a significant loss in sign diversity.

**Collapsing depth is exploding gradient in disguise**  In theorem 1, we used the fact that the output of the error layer of the network was positive to bound the size of a useful gradient-based update. In other words, we used the fact that the domain of the error layer was bounded. However, the collapsing domain problem causes not just a reduction of the size of the domain of the error layer, but of all intermediate layers. Hence, we expect the largest useful update to shrink in proportion with the reduction of the size of the domain. Therefore, we suspect that a collapsing domain will

ultimately have the same effect on the largest useful update size of each layer as exploding gradients, that is to reduce them and thus cause a low effective depth.

In table 2, we show the final error values achieved by training ReLU, layer-ReLU and tanh on CIFAR10. The errors are substantially higher than those achieved by the exploding architectures, except for batch-ReLU. Also, training with layer-wise step sizes did not help compared to training with a single step size. In figure 4A, we show the estimated best relative update sizes for each layer. This time, there is no downward trend towards lower layers, which is likely why training with a single step size is "sufficient". Interestingly, we note that the difference between the $\frac{1}{GSC}$ bound and the empirically optimal relative update sizes is now much larger than it is for the exploding architectures (see figure 2A). This suggests that indeed, collapsing domains may similarly reduce the optimal relative update size, just as exploding gradients. In figure 4D, we find that again, the effective depth reached is significantly lower than the compositional depth of the network and is comparable to that of architectures with exploding gradients.

In figure 4G and H, we plot the pre-activation standard deviation and sign diversity at the highest nonlinearity layer throughout training. Interestingly, pseudo-linearity declines significantly early in training. The networks become less linear through training.

**Summary**   In neural network design, there is an inherent tension between avoiding exploding gradients and preserving the domain of forward activations. Avoiding one effect can bring about or exacerbate the other. Both effects are capable of severely hampering training. This tension is brought about by the discrepancy of determinant and qm norm of layer-wise Jacobians and is a foundational reason for the difficulty in constructing very deep trainable network.

In this paper, we do not give a rigorous definition of the collapsing domain problem, because it is hard to assess and measure. A number of metrics exist which all apply is somewhat different situations. We already discussed two metrics: pre-activation standard deviation and pre-activation sign diversity. In mean field theory, activation correlation plays a prominent role (section B.1.1). In fact, mean field theory is a formidable tool for statically estimating the properties of specific architectures, such as explosion rate and pre-activation standard deviation. See e.g. Schoenholz et al. (2017) for an anlysis of tanh and Arpit et al. (2016) for an analysis of batch-ReLU. We discuss collapsing domains further in the future work section B.4.

## 7    RESNET REDUCES THE GRADIENT - AND RESNET EXPLAINED

ResNet and related architectures that utilize skip connections have been very successful recently. One reason for this is that they can be successfully trained to much greater depths than vanilla networks. In this section, we show how skip connections are able to greatly reduce the GSC and thus largely circumvent the exploding gradient problem.

Let $s_b$, $1 \leq b \leq B$ be the function computed by the $b$'th skip connection. Let $\rho_b$ be the function computed by the $b$'th residual block. $b = B$ corresponds to the lowest block and $b = 1$ corresponds to the highest block. Let $f_b = s_b + \rho_b$.

**Definition 4.**   We say a function $f_b$ is '$k$-diluted' with respect to a random vector $v$ if there exists a matrix $S_b$ and a function $\rho_b$ such that $f_b(v) = S_b v + \rho_b(v)$ and $\frac{\mathbb{Q}_v ||S_b v||_2}{\mathbb{Q}_v ||\rho_b(v)||_2} = k$.

$k$-dilution expresses the idea that the kinds of functions that $f_b$ represents are of a certain form if $s_b$ is restricted to matrix multiplication. The larger the value of $k$, the more $\rho_b$ is "diluted" by a linear function, bringing $f_b$ itself closer and closer to a linear function. Note that the identity function can be viewed as matrix multiplication with the identity matrix.

**Theorem 3.**   *Under certain conditions, if a function $\rho_b$ would cause the $GSC$ to grow with expected rate $r$, $k$-diluting $\rho_b$ with an uncorrelated linear transformation reduces this rate to $1 + \frac{r-1}{k^2+1} + O((r-1)^2)$. (See section F.3 for details.)*

This reveals the reason why ResNet circumvents the exploding gradient problem. Diluting the block function by a factor $k$ does not just reduce the growth of the GSC by factor $k$, but by $k^2+1$. Therefore what appears to be a relatively mild reduction in representational capacity achieves, surprisingly, a

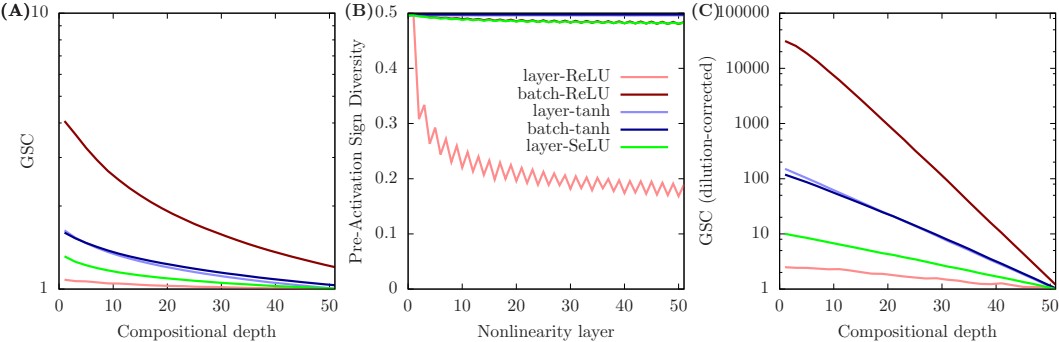

Figure 3: Key metrics for ResNet architectures at various depths. Left and right graph only show values between skeip connections. The x axis shows depth in terms of the number of linear layers counted from the input. Note: The curve for layer-tanh and batch-tanh is shadowed by SeLU in the right graph.

relatively large amount of gradient reduction, and therefore ResNet can be trained successfully to "unreasonably" great depths for general architectures.

To validate our theory, we repeated the experiments in figure 1 with 5 ResNet architectures: layer-ReLU, batch-ReLU, layer-tanh, batch-tanh and layer-SeLU. Each residual block is bypassed by an identity skip connection and composed of 2 sub-blocks of 3 layers each: first a normalization layer, then a nonlinearity layer, and then a fully-connected layer, similar to He et al. (2016a); Zaguroyko & Komodakis (2016). For further architectural details, see section I.1. Comparing figure 3A to figure 1A, we find the gradient growth is indeed much lower for ResNet compared to vanilla networks, with much of it taking place in the lower layers. In figure 3B we find that the rate of domain collapse for layer-ReLU, as measured by pre-activation sign diversity, is also significantly slowed.

We then went on to check whether the gradient reduction experienced is in line with theorem 3. We measured the dilution level $k_b$ and growth rate $r_b$ at each residual block $b$ and then replaced the growth rate with $1 + (k_b^2 + 1)(r_b - 1)$. The result of this post-processing is found in figure 3C. Indeed, the GSC of the exploding architectures now again grows almost linearly in log space, with the exception of batch-ReLU in the lowest few layers. The explosion rates closely track those in figure 1A, though they are slightly higher. This confirms that the estimate of the magnitude of gradient reduction from theorem 3 is quite accurate.

We then repeated the CIFAR10 experiments depicted in figure 2 with our 5 ResNet architectures. The results are shown in figure 5. As expected, in general, ResNet enables higher relative update sizes, achieves lower error, a higher effective depth and is less "robust" to taylor approximation than vanilla networks. The only exception to this trend is the layer-SeLU ResNet when compared to the SeLU vanilla network, which already has a relatively slowly exploding gradient to begin with. Note that the severe reduction of the GSC persists throughout training (figure 5C). Also see table 2 to compare final error values. Note that in order to make the effective depth results in figure 5D comparable to those in figure 2D, we applied the residual trick to ResNet. We let the initial function $i$ encompass not just the skip function $s$, but also the initial block function $\rho$. Hence, we set $i_b = s_b + \rho_b(\theta^{(0)})$ and $r_b = \rho_b(\theta) - \rho_b(\theta^{(0)})$. Note that our effective depth values for ResNet are much higher than those of Veit et al. (2016). This is because we use a much more conservative estimate of this intractable quantity for both ResNet and vanilla networks.

Gradient reduction is achieved not just by identity skip connections but, as theorem 3 suggests, also by skip connections that multiply the incoming value with a Gaussian random matrix. Results for those skip connections can be found in table 1.

Veit et al. (2016) argues that deep ResNets behave like an ensemble of relatively shallow networks. We argue that comparable vanilla networks often behave like ensembles of even shallower networks. Jastrzebski et al. (2018) argues that deep ResNets are robust to lesioning. Additionally, we argue that comparable vanilla networks are often even more robust to depth reduction when considering the first order Taylor expansion.

## 7.1 THE LIMITS OF DILUTION

$k$-dilution has its limits. Any $k$-diluted function with large $k$ is close to a linear function. Hence, we can view $k$-dilution as another form of pseudo-linearity that can damage representational capacity. It also turns out that under similar conditions to those used in theorem 3, dilution only disappears slowly as diluted functions are composed. If the diluting linear functions $s_b$ are the identity functions, this corresponds to feature refinement as postulated by Jastrzebski et al. (2018).

**Proposition 7.** *Under certain conditions, the composition of $B$ random functions that are $k_b$-diluted in expectation respectively is $\left( \left( \prod_l (1 + \frac{1}{k_b^2}) \right) - 1 \right)^{-\frac{1}{2}}$-diluted in expectation. (See section E.7 for details.)*

More simply, assume all the $k_b$ are equal to some $k$. Ignoring higher-order terms, the composition is $\frac{1}{\sqrt{B}} k$-diluted. The flipside of an $O(k^2)$ reduction in gradient via dilution is thus the requirement of $O(k^2)$ layers to eliminate that dilution. This indicates that the overall amount of gradient reduction achievable through dilution without incurring catastrophic pseudo-linearity is limited.

## 7.2 CHOOSING DILUTION LEVELS

The power of our theory lies in exposing the GSC-reducing effect of skip connections for general neural network architectures. As far as we know, all comparable previous works (e.g. Yang & Schoenholz (2017); Balduzzi et al. (2017)) demonstrated similar effects only for specific architectures. Our argument is not that certain ResNet's achieve a certain level of GSC reduction, but that ResNet users have the power to choose the level of GSC reduction by controlling the amount of dilution. For example, while the level of dilution increases as we go deeper in the style of ResNet architecture we used for experiments in this section, this need not be so.

The skip function $s$ and block function $\rho$ can be scaled with constants to achieve arbitrary, desired levels of dilution (Szegedy et al., 2016). Alternatively, instead of putting all normalization layers in the residual blocks, we can insert them between blocks / skip connections. This would keep the dilution level constant and hence cause gradients to explode again, though at a lower rate compared to vanilla networks.

## 7.3 THE ORTHOGONAL INITIAL STATE

Applying the residual trick to ResNet reveals several insights. The difference between ResNet and vanilla networks in terms of skip connections is superficial, because both ResNet and vanilla networks are residual networks in the framework of equation 2. Also, both ResNet and vanilla networks have nonlinear initial functions, because $\rho_b(\theta_b^{(0)})$ is initially nonzero and nonlinear. However, there is one key difference. The initial functions of ResNet are closer to a linear transformation and indeed closer to an orthogonal transformation because they are composed of a nonlinear function $\rho_b(\theta_b^{(0)})$ that is significantly diluted by an orthogonal transformation $s_b$. Therefore, ResNet, while being conceptually more complex, is mathematically simpler.

We have shown how ResNets achieve a reduced gradient via $k$-dilution. And just as with effective depth, the residual trick allows us to generalize this notion to arbitrary networks.

**Definition 5.** We say a residual network $f(\theta)$ has an *'orthogonal initial state'* if each initial function $i_l$ is multiplication with an orthogonal matrix or a slice / multiple thereof and $r_l(\theta_l)$ is the zero function.

Any network that is trained from an (approximate) orthogonal initial state can benefit from reduced gradients via dilution to the extent to which initial and residual function are uncorrelated. (See section F.3 for more information.). ResNet is a style of architecture that achieves this, but it is far from being the only one. Balduzzi et al. (2017) introduced the 'looks-linear initialization' (LLI) for ReLU networks, which achieves not only an approximate orthogonal initial state, but outperformed ResNet in their experiments. We detail this initialization scheme in section H. In table 2, we show that a simple ReLU network with LLI can achieve an ever lower training error than ResNet on CIFAR10. In figure 6C, we find that indeed LLI reduces the gradient growth of batch-ReLU drastically not just

in the initialized state, but throughout training even as the residual functions grow beyond the size achieved under Gaussian initialization (compare figure 6E to 2E and 4E). DiracNet (Zagoruyko & Komodakis, 2017) achieves an approximate orthogonal initial state in a very similar way to LLI. An even simpler but much less powerful strategy is to initialize weight matrices as orthogonal matrices instead of Gaussian matrices. This reduces the gradient growth in the initialized state somewhat (table 1).

Using the ensemble view of very deep networks reveals another significant disadvantage of non-orthogonal initial functions. The output computed by an ensemble member must pass through the initial functions of the layers not contained in that ensemble member to reach the prediction layer. Therefore, having non-orthogonal initial functions is akin to taking a shallow network and adding additional, *untrainable* non-orthogonal layers to it. This has obvious downsides such as a collapsing domain and / or exploding gradient, and an increasingly unfavorable eigenspectrum of the Jacobian (Saxe et al., 2014). One would ordinarily not make the choice to insert such untrainable layers. While there has been some success with convolutional networks where lower layers are not trained (e.g. Saxe et al. (2011); He et al. (2016c)), it is not clear whether such networks are capable of out-performing other networks where such layers are trained. While skip connections do not resolve the tension between exploding gradients and collapsing domains, they reduce the pathology by avoiding unnecessary non-orthogonality contained in the initial function.

The big question is now: What is the purpose of *not* training a network from an orthogonal initial state? We are not aware of such a purpose. Since networks with orthogonal initial functions are mathematically simpler than other networks, we argue they should be the default choice. Using non-orthogonality in the initial function, we believe, is what requires explicit justification.

Balduzzi et al. (2017) asks in the title: *If ResNet is the answer, then what is the question?* We argue that a better question would be: *Is there a question to which vanilla networks are the answer?*

## 8    CONCLUSION

**Summary**    In this paper, we demonstrate that contrary to popular belief, many MLP architectures composed of popular layer types exhibit exploding gradients, and those that do not exhibit collapsing domains (section 3). This tradeoff is caused by the discrepancy between absolute determinants and qm norms of layer-wise Jacobians (section 5). Both sides of this tradeoff cause pathologies. Exploding gradients, when defined by the GSC (section 2) cause low effective depth (section 4). Collapsing domains cause pseudo-linearity and can also cause low effective depth (section 6). However, both pathologies are caused to a surprisingly large degree by untrainable, and thus potentially unnecessary non-orthogonality contained in the initial functions. Making the initial functions more orthogonal via e.g. skip connections leads to improved outcomes (section 7).

**Practical Recommendations**

- **Train from an orthogonal initial state,** i.e. initialize the network such that it is a series of orthogonal linear transformations. This can greatly reduce the growth of the GSC and domain collapse not just in the initial state, but also as training progresses. It can prevent the forward activations from having to pass through unnecessary non-orthogonal transformations. Even if a perfectly orthogonal initial state is not achievable, an architecture that approximates this such as ResNet can still confer significant benefit.

- When not training from an orthogonal initial state, **avoid low effective depth**. A low effective depth signifies that the network is composed of an ensemble of networks significantly shallower than the full network. If the initial functions are not orthogonal, the values computed by these ensemble members have to pass through what may be unnecessary and harmful non-orthogonal transformations. Low effective depth may be caused by, for example, exploding gradients or a collapsing domain.

- **Avoid pseudo-linearity.** For the representational capacity of a network to grow with depth, linear layers must be separated by nonlinearities. If those nonlinearities can be approximated by linear functions, they are ineffective. Pseudo-linearity can be caused by, for example, a collapsing domain.

- **Keep in mind that skip connections help in general, but other techniques do not** Diluting a nonlinear function with an uncorrelated linear function can greatly help with the pathologies described above. Techniques such as normalization layers, careful initialization of weights or SeLU nonlinearities can prevent the explosion or vanishing of forward activations. Adam, RMSprop or vSGD can improve performance even if forward activations explode or vanish. While those are important functionalities, these techniques in general neither help address gradient explosion relative to forward activations as indicated by the GSC nor collapsing domains.

- **As the GSC grows, adjust the step size.** If it turns out that some amount of growth of the GSC is unavoidable or desirable, weights in lower layers could benefit from experiencing a lower relative change during each update. Optimization algorithms such as RMSprop or Adam may partially address this.

- **Control dilution level to control network properties.** Skip connections, normalization layers and scaling constants can be placed in a network to trade off gradient growth and representational capacity. Theorem 3 can be used for a static estimate of the amount of gradient reduction achieved. Similarly, proposition 7 can be used for a static estimate of the overall dilution of the network.

- **Great compositional depth may not be optimal.** Networks with more than 1000 layers have recently been trained (He et al., 2016b). Haber & Ruthotto (2017) gave a formalism for training arbitrarily deep networks. However, ever larger amounts of dilution are required to prevent gradient explosion (Szegedy et al., 2016). This may ultimately lead to an effective depth much lower than the compositional depth and individual layers that have a very small impact on learning outcomes, because functions they represent are very close to linear functions. If there is a fixed parameter budget, it may be better spent on width than extreme depth (Zaguroyko & Komodakis, 2016).

**Implications for deep learning research**

- **Exploding gradients matter.** They are not just a numerical quirk to be overcome by rescaling but are indicative of an inherently difficult optimization problem that cannot be solved by a simple modification to a stock algorithm.

- **Use GSC as a benchmark for gradient explosion.** For the first time, we established a rigorous link between a metric for exploding gradients and hardness of training. The GSC is also robust to network rescaling, layer width and individual layers.

- **Any neural network is a residual network.** The residual trick allows the application of ResNet-specific tools such as the popular theory of effective depth to arbitrary networks.

- **Step size matters when studying the behavior of networks.** We found that using different step sizes for different layers had a profound impact on the training success of various architectures. Many studies that investigate fundamental properties of deep networks either do not consider layerwise step sizes (e.g. Schoenholz et al. (2017)) or do not even consider different global step sizes (e.g. Keskar et al. (2017)). This can lead to inaccurate conclusions.

We provide continued discussion in section B.

| Nonlinearity | Normalization | Matrix type | Skip type | Width | $GSC(L,0)$ | St. Dev. | Sign Div. |
|---|---|---|---|---|---|---|---|
| ReLU | none | Gaussian | none | 100 | 1.52 | 0.22 | 0.030 |
| ReLU | layer | Gaussian | none | 100 | 1.16 | 0.096 | 0.029 |
| ReLU | LOlayer | Gaussian | none | 100 | 1.30 | 0.10 | 0.030 |
| ReLU | batch | Gaussian | none | 100 | 5728 | 1.00 | 0.41 |
| tanh | none | Gaussian | none | 100 | 1.26 | 0.096 | 0.50 |
| tanh | layer | Gaussian | none | 100 | 72.2 | 1.00 | 0.50 |
| tanh | LOlayer | Gaussian | none | 100 | 71.9 | 1.00 | 0.50 |
| tanh | batch | Gaussian | none | 100 | 93.6 | 1.00 | 0.50 |
| SeLU | none | Gaussian | none | 100 | 6.36 | 0.97 | 0.42 |
| SeLU | LOlayer | Gaussian | none | 100 | 7.00 | 0.98 | 0.42 |
| ReLU | batch | Gaussian | none | 200 | 5556 | 1.00 | 0.42 |
| ReLU | batch | Gaussian | none | 100/200 | 5527 | 1.00 | 0.41 |
| SeLU | none | Gaussian | none | 200 | 5.86 | 0.99 | 0.45 |
| SeLU | none | Gaussian | none | 100/200 | 6.09 | 0.98 | 0.43 |
| ReLU | none | orthogonal | none | 100 | 1.29 | 0.20 | 0.03 |
| ReLU | layer | orthogonal | none | 100 | 1.00 | 0.10 | 0.03 |
| ReLU | batch | orthogonal | none | 100 | 5014 | 1.00 | 0.42 |
| tanh | none | orthogonal | none | 100 | 1.18 | 0.10 | 0.50 |
| tanh | layer | orthogonal | none | 100 | 56.3 | 1.00 | 0.50 |
| tanh | batch | orthogonal | none | 100 | 54.6 | 1.00 | 0.50 |
| SeLU | none | orthogonal | none | 100 | 5.47 | 1.00 | 0.49 |
| ReLU | none | looks-linear | none | 100 | 1.00 | 1.00 | 0.50 |
| ReLU | layer | looks-linear | none | 100 | 1.00 | 1.00 | 0.50 |
| ReLU | batch | looks-linear | none | 100 | 1.00 | 1.00 | 0.50 |
| ReLU | layer | Gaussian | identity | 100 | 1.08 | 0.56 | 0.19 |
| ReLU | batch | Gaussian | identity | 100 | 4.00 | 1.00 | 0.48 |
| tanh | layer | Gaussian | identity | 100 | 1.63 | 1.00 | 0.50 |
| tanh | batch | Gaussian | identity | 100 | 1.57 | 1.00 | 0.50 |
| SeLU | layer | Gaussian | identity | 100 | 1.31 | 0.99 | 0.48 |
| ReLU | layer | Gaussian | Gaussian | 100 | 1.17 | 0.56 | 0.18 |
| ReLU | batch | Gaussian | Gaussian | 100 | 4.50 | 1.00 | 0.48 |
| tanh | layer | Gaussian | Gaussian | 100 | 1.97 | 1.00 | 0.50 |
| tanh | batch | Gaussian | Gaussian | 100 | 1.71 | 1.00 | 0.50 |
| SeLU | layer | Gaussian | Gaussian | 100 | 1.53 | 9.97 | 0.48 |

Table 1: Key metrics for architectures in their randomly initialized state evaluated on Gaussian noise. In the 'Normalization' column, 'layer' refers to layer normalization, 'batch' refers to batch normalization, 'LOlayer' refers to length-only layer normalization and 'none' refers to an absence of a normalization layer. In the 'Matrix type' column, 'Gaussian' refers to matrices where each entry is drawn from an independent Gaussian distribution with mean zero and a standard deviation that is constant across all entries. 'orthogonal' refers to a uniformly random orthogonal matrix and 'looks-linear' refers to the initialization scheme proposed by Balduzzi et al. (2017) and expounded in section H. In the 'Skip type' column, 'identity' refers to identity skip connections and 'Gaussian' refers to skip connections that multiply the incoming value with a matrix where each entry is drawn from an independent Gaussian distribution with mean zero and a standard deviation that is constant across all entries. 'none' refers to an absence of skip connections. In the 'Width' column, '100/200' refers to linear layers having widths alternating between 100 and 200. 'St. Dev.' refers to pre-activation standard deviation. 'Sign Div.' refers to pre-activation sign diversity. For details and definitions, see section I. Red values indicate gradient explosion or pseudo-linearity.

| Nonlinearity | Normalization | Matrix type | Skip type | Error (custom step size) | Error (single step size) |
|---|---|---|---|---|---|
| ReLU | none | Gaussian | none | 31.48% | **19.24%** |
| ReLU | layer | Gaussian | none | 42.48% | **21.23%** |
| ReLU | batch | Gaussian | none | **34.83%** | 76.65% |
| tanh | none | Gaussian | none | 23.42% | **16.22%** |
| tanh | layer | Gaussian | none | **1.92%** | 17.5% |
| tanh | batch | Gaussian | none | **12.31%** | 23.8% |
| SeLU | none | Gaussian | none | **0.24%** | 1.78% |
| ReLU | none | looks-linear | none | **0.002%** | 0.008% |
| ReLU | layer | looks-linear | none | **0.77%** | 1.2% |
| ReLU | batch | looks-linear | none | 0.38% | **0.19%** |
| tanh | layer | Gaussian | id | 0.35% | **0.27%** |
| tanh | batch | Gaussian | id | **0.13%** | 0.24% |
| ReLU | layer | Gaussian | id | 2.09% | **1.49%** |
| ReLU | batch | Gaussian | id | **0.06%** | 0.096% |
| SeLU | layer | Gaussian | id | **1.55%** | **1.55%** |

Table 2: Training classificaion error for architectures trained on CIFAR10. In the 'Normalization' column, 'layer' refers to layer normalization, 'batch' refers to batch normalization and 'none' refers to an absence of a normalization layer. In the 'Matrix type' column, 'Gaussian' refers to matrices where each entry is drawn from an independent Gaussian distribution with mean zero and a standard deviation that is constant across all entries. 'looks-linear' refers to the looks-linear initialization scheme proposed by Balduzzi et al. (2017) and expounded in section H. In the 'Skip type' column, 'identity' refers to identity skip connections and 'none' refers to an absence of skip connections. In the two rightmost columns, we show the training classification error achieved when using a single step size and when using a custom step size for each layer. Whichever error value is lower is shown in bold. For further methodological details, see section I. For a detailed breakdown of these results, see figures 2, 4, 5 and 6.

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

# A    ADDITIONAL EXPERIMENTAL RESULTS

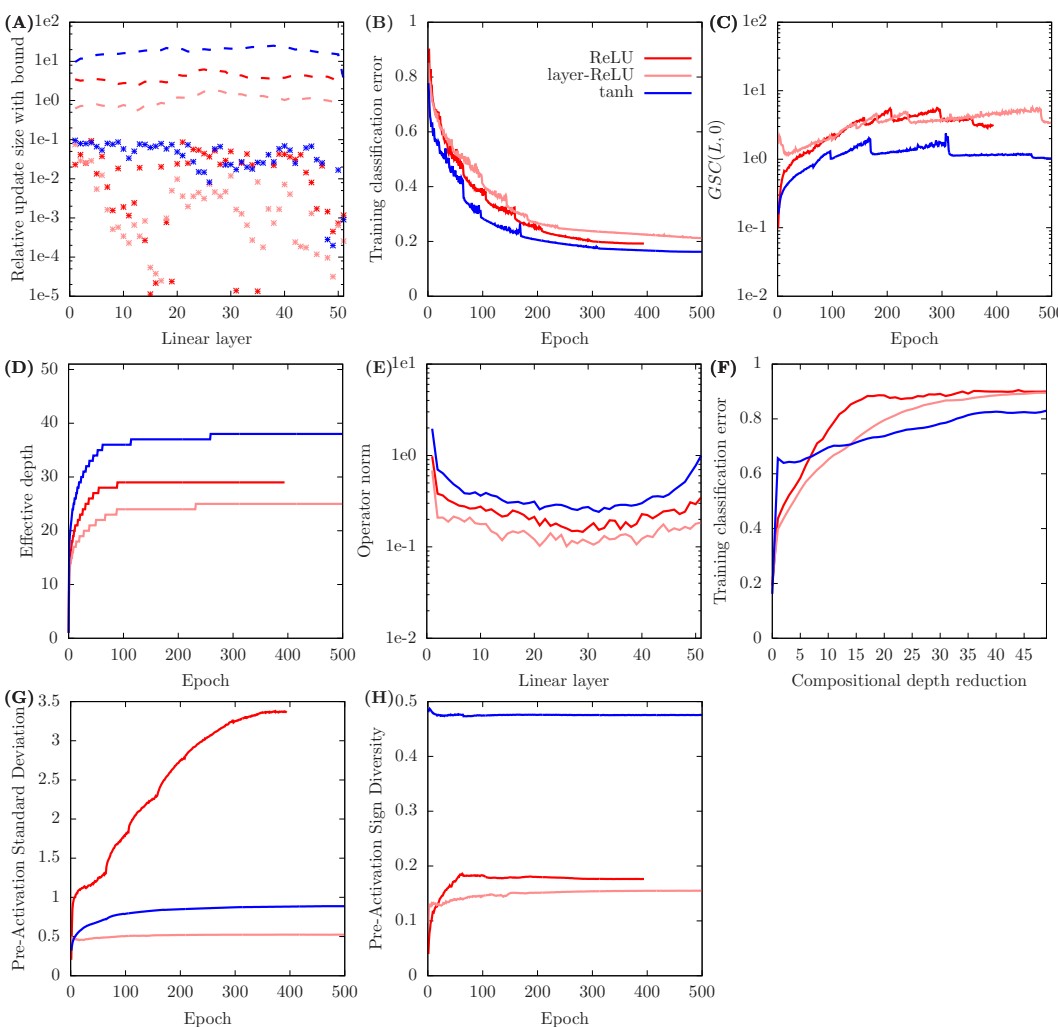

Figure 4: Key metrics for architectures with collapsing domain trained on CIFAR10. The top left graph shows the estimated optimal relative update size in each layer according to the algorithm described in section I.3. Remaining graphs show results obtained from training with a single step as this achieved lower error than training with layer-wise step sizes (see table 2). The top two rows are equivalent to graphs in figure 2. The bottom row shows pre-activation standard deviation and pre-activation sign diversity (see section I.2 for definition) of the highest nonlinearity layer as training progresses.

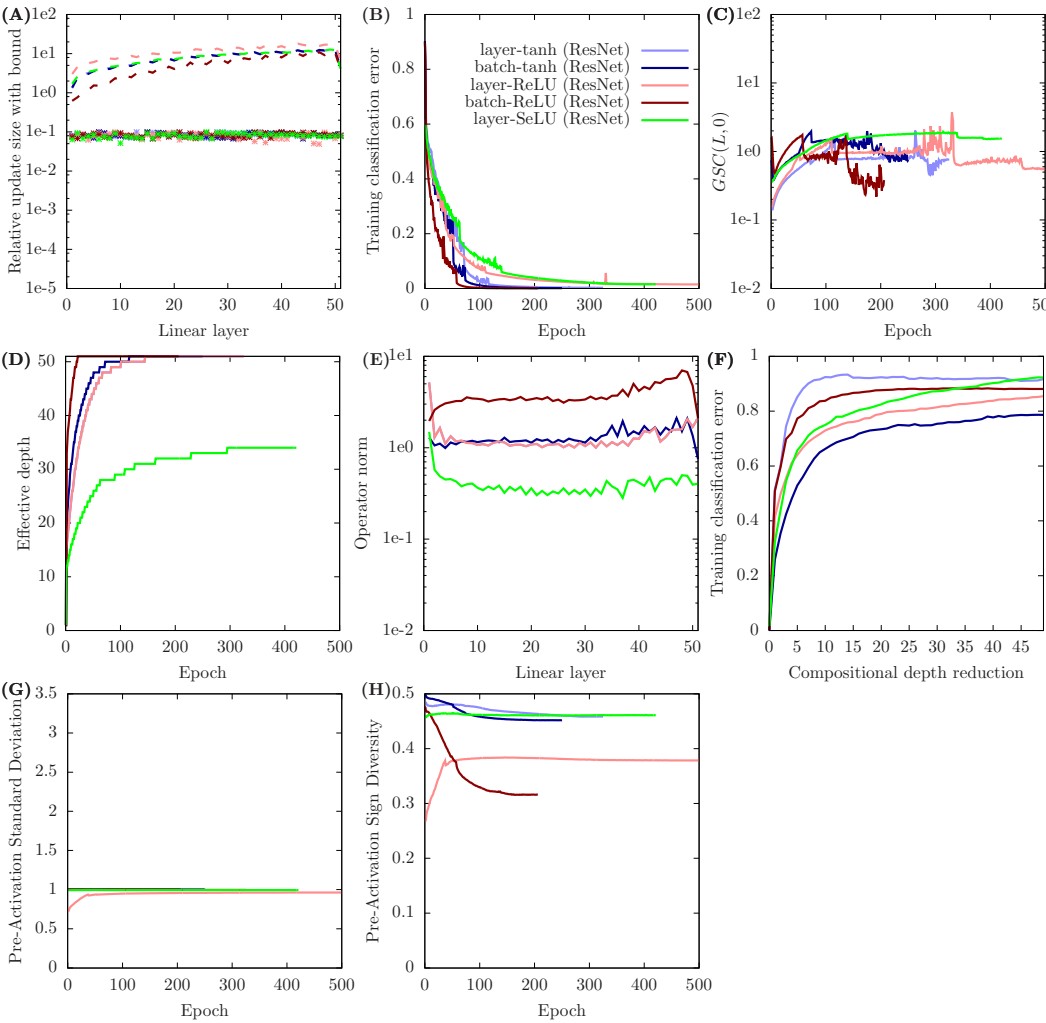

Figure 5: Key metrics for ResNet architectures trained on CIFAR10. The top left graph shows the estimated optimal relative update size in each layer according to the algorithm described in section I.3. Remaining graphs show results obtained from training with either those step sizes or a single step size, whichever achieved a lower error (see table 2). The top two rows are equivalent to graphs in figure 2. The bottom row shows pre-activation standard deviation and pre-activation sign diversity (see section I.2 for definition) of the highest nonlinearity layer as training progresses.

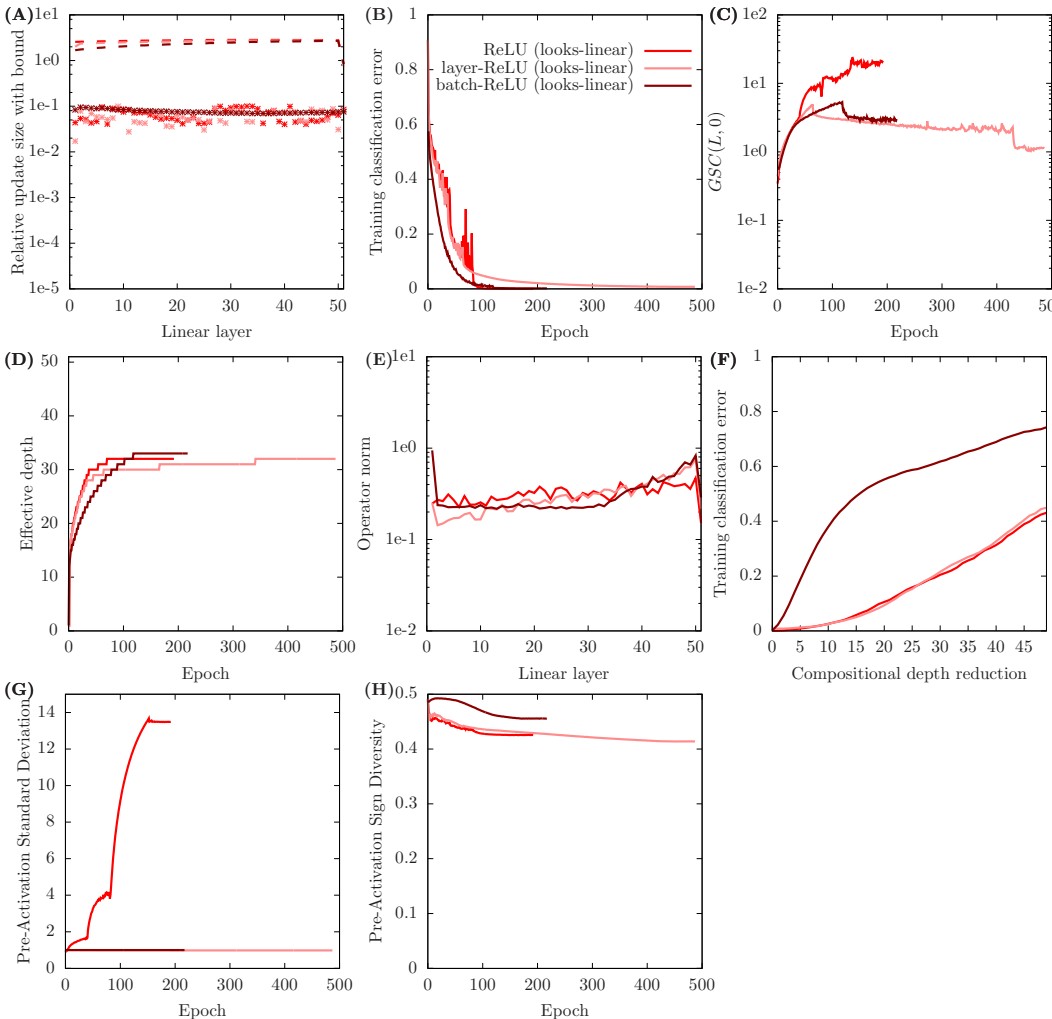

Figure 6: Key metrics for ReLU-based architectures with looks-linear initialization trained on CIFAR10. The top left graph shows the estimated optimal relative update size in each layer according to the algorithm described in section I.3. Remaining graphs show results obtained from training with either those step sizes or a single step size, whichever achieved a lower error (see table 2). The top two rows are equivalent to graphs in figure 2. The bottom row shows pre-activation standard deviation and pre-activation sign diversity (see section I.2 for definition) of the highest nonlinearity layer as training progresses.

# B  DISCUSSION

## B.1  RELATED WORK

So far, we have discussed exploding gradients and collapsing domains. In this section, we review related metrics and concepts from literature.

We build on the work of Balduzzi et al. (2017), who introduced the concept of gradient shattering. This states that in deep networks, gradients with respect to nearby points become more and more uncorrelated with depth. This is very similar to saying that the gradient is only informative in a smaller and smaller region around the point at which it is taken. This is precisely what happens when gradients explode and also, as we argue in section 6, when the domain collapses. Therefore, the exploding gradient problem and collapsing domain problem can be viewed as a further specification of the shattering gradient problem rather than as a counter-theory or independent phenomenon.

We extend the work of Balduzzi et al. (2017) in several important ways. First, they claim that the exploding gradient problem "has been largely overcome". We show that this is not true, especially in the context of very deep batch-ReLU MLPs, which are central to their paper. Second, by using effective depth we make a rigorous argument as to why exploding gradients cause hardness of training. While Balduzzi et al. (2017) point out that shattering gradients interfere with theoretical guarantees that exist for various optimization algorithms, they do not provide a definitive argument as to why shattering gradients are in fact a problem. Third, our analysis extends beyond ReLU networks.

We also build on the work of Raghu et al. (2017). They showed that both trajectories and small perturbations, when propagated forward, can increase exponentially in size. However, they do not distinguish too important cases: (i) an explosion that is simply due to an increase in the scale of forward activations and (ii) an explosion that is due to an increase in the gradient relative to forward activations. We are careful to make this distinction and focus only on case (ii). Since this is arguably the more interesting case, we believe the insights generated in our paper are more robust.

Saxe et al. (2014) investigated another important pathology of very deep networks: the divergence of singular values in multi-layer Jacobians. As layer-wise Jacobians are multiplied, the variances of their singular values compound. This leads to the direction of the gradient being determined by the dominant eigenvectors of the multi-layer Jacobian rather than the label, which slows down training considerably.

In their seminal paper, Ioffe & Szegedy (2015) motivated batch normalization with the argument that changes to the distribution of intermediate representations, which they term 'covariate shift', are pathological and need to be combated. This argument was then picked up by e.g. Salimans & Kingma (2016) and Chunjie et al. (2017) to motivate similar normalization schemes. We are not aware of any rigorous definition of the 'covariate shift' concept nor do we understand why it is undesirable. After all, isn't the very point of training deep networks to have each layer change the function it computes, to which other layers co-adapt, to which then other layers co-adapt and so on? Having each layer fine-tune its weights in response to shifts in other layers seems to be the very mechanism by which deep networks achieve high accuracy.

A classical notion of trainability in optimization theory is the conditioning of the Hessian. This can also deteriorate with depth. Recently, Luo (2017) introduced an architecture that combats this pathology in an effective and computationally tractable way via iterative numerical methods and matrix decomposition. Matrix decomposition has also been used by e.g. Arjovsky et al. (2016); Anonymous (2018b) to maintain orthogonality of recurrent weight matrices. Maybe such techniques could also be used to reduce the divergence of singular values of the layer-wise Jacobian during training.

### B.1.1  MEAN FIELD THEORY - EXPLODING GRADIENTS / COLLAPSING DOMAIN VS ORDER / CHAOS

Our work bears similarity to a recent line of research studying deep networks using mean field theory (Poole et al., 2016; Schoenholz et al., 2017; Yang & Schoenholz, 2017; Anonymous, 2018d). Those papers use infinitely wide networks to statically analyze the expected behavior of forward activations and gradients in the initialized state. They identify two distinct regimes, order and chaos, based on whether an infinitesimal perturbation shrinks or grows in expectation respectively as it is propagated

forward. This corresponds to the expected qm norm of the layer-wise Jacobian being smaller or larger than 1 respectively. They show that in the chaotic regime, gradient vector length explodes whereas in the ordered regime, gradient vector length vanishes. Further, they show that for tanh MLPs the correlation between forward activation vectors corresponding to two different data inputs converges to 1 ('unit limit correlation') in the ordered regime as activations are propagated forward and to some value less than 1 in the chaotic regime. Specifically, in a tanh MLP without biases, in the chaotic regime, the correlation converges to 0 ('zero limit correlation'). They show how to use mean field theory as a powerful tool for the static analysis of individual network architectures.

As in mean field theory, some of our analysis relies on the expected behavior of networks in their randomly initialized state (theorem 2). Further, it is clear that the order / chaos dichotomy bears similarity to the exploding gradient problem / collapsing domain problem dichotomy as presented in this paper. However, there are also important differences.

One difference is that we argue that the GSC is a better metric for determining the presence of pathological exploding or vanishing gradients than gradient vector length and thus more meaningful than order / chaos. Using the GSC, we obtain very different regions of explosion, vanishing and stability for popular architectures compared to gradient vector length. For a tanh MLP with no biases, using gradient vector length, vanishing is achieved for $\sigma_w < 1$, stability for $\sigma_w = 1$ and explosion for $\sigma_w > 1$. ($\sigma_w$ denotes the standard deviation of weight matrix entries times the square root of the right dimsion of the weight matrix, as defined in Poole et al. (2016).) For a tanh MLP with no biases, using the GSC, vanishing is impossible, stability is achieved for $\sigma_w \leq 1$ and explosion for $\sigma_w > 1$. For a ReLU MLP with no biases, using gradient vector length, vanishing is achieved for $\sigma_w < \sqrt{2}$, stability for $\sigma_w = \sqrt{2}$ and explosion for $\sigma_w > \sqrt{2}$. For a ReLU MLP with no biases, using the GSC, stability is inevitable.

The advantage of considering GSC can be seen in the case of the ReLU network. For tanh, Schoenholz et al. (2017) showed that order corresponds to an exponential convergence towards a unit limit correlation and chaos corresponds to an exponential convergence towards a zero limit correlation. For a ReLU MLP with no biases and $\sigma_w > \sqrt{2}$, infinitesimal noise grows (chaos), yet the correlation converges sub-exponentially to zero, which is a behavior we would expect from the edge of chaos. As we saw above, using the GSC to define order / chaos that is precisely where we are: the edge of chaos.

A second difference is that the concepts of unit limit correlation and the collapsing domain problem are not the same. In fact, the former can be seen as a special case of the latter. In a tanh MLP with no bias and $\sigma_w$ slightly larger than 1, correlation converges to 0 and eventually, gradients explode. Yet the domain can still collapse dramatically in the short term as shown in figure 1B to cause pseudo-linearity. In a tanh MLP with no bias and $\sigma_w$ very large, again, correlation converges to 0 and gradients explode. However, the tanh layer maps all points close to the corners of the hypercube, which corresponds to domain collapse.

We do not use the assumption of infinite width in our analysis. The only possible exception is that the SSD assumption in proposition 7 can be viewed as implying infinite width.

While Schoenholz et al. (2017) conjectures that the edge of chaos is necessary for training very deep networks, our paper provides somewhat contrary evidence. Our two best performing vanilla architectures, SeLU and layer-tanh, are both inside the chaotic regime whereas ReLU, layer-ReLU and tanh, which are all on the edge of chaos, exhibit a higher training classification error. Clearly, chaotic architectures avoid pseudo-linearity. The difference between our experiments and those in Schoenholz et al. (2017) is that we allowed the step size to vary between layers. This had a large impact, as can be seen in table 2. We believe that our results underscore the importance of choosing appropriate step sizes when comparing the behavior of different neural architectures or training algorithms in general.

In section 4, we present a rigorous argument for the harmful nature of exploding gradients, and thus of chaos as defined by the GSC, at high depth. No comparable argument exists in mean field literature.

While Schoenholz et al. (2017) obtained low accuracies for networks exhibiting unit limit correlation, it is not clear a priori that this effect is harmful for accuracy. After all, correlation information is a rather small part of the information present in the data, so the remaining information might be

sufficient for learning. As a simple example, consider k-means. Performing k-meafns on an arbitrary dataset yields the same result as first adding a large constant to the data and then performing k-means, even though the addition can easily destroy correlation information. In contrast, in section 6 , we show how collapsing domains can directly harm expressivity and trainability.

Yang & Schoenholz (2017) shows that pathologies such as gradient explosion that arise in vanilla networks are reduced in specific ResNet architectures. We extend this finding to general ResNet architectures.

Anonymous (2018d) proposes to combat gradient growth by downscaling the weights in the residual block of a ResNet. This corresponds to increased dilution, which indeed reduces gradient growth as shown in section 7. However, we also show in proposition 7 that the reduction achievable in this way may without suffering catastrophic pseudo-linearity be limited. Anonymous (2018d) also proposes to combat the exploding gradient problem by changing the width of intermediate layers. Our analysis in section 4.4 suggests that this is not effective in reducing the growth of the GSC. Anonymous (2018d) concludes that changing the width combats the exploding gradient problem because they implicitly assume that the pathology of exploding gradients is determined by the scale of individual components of the gradient vector rather than the length of the entire vector or the GSC. They do not justify this assumption. We propose the GSC as a standard for assessing pathological exploding gradients to avoid such ambiguity.

### B.1.2 ResNet from a dynamical systems view

Recently, Haber et al. (2017); Haber & Ruthotto (2017); Chang et al. (2017); Anonymous (2018a) proposed ResNet architectures inspired by dynamical systems and numerical methods for ordinary differential equations. The central claim is that these architectures are stable at arbitrary depth, i.e. both forward activations and gradients (and hence GSC) are bounded as depth goes to infinity. They propose four practical strategies for building and training ResNets: (a) ensuring that residual and skip functions compute vectors orthogonal to each other by using e.g. skew-symmetric weight matrices (b) ensuring that the Jacobian of the skip function has eigenvalues with negative real part by using e.g. weight matrices factorized as $-C^T C$ (c) scaling each residual function by $1/B$ where B is the number of residual blocks in the network and (d) regularizing weights in successive blocks to be similar via a fusion penalty.

| Architecture | $GSC(L, 0)$ (base 10 log) | $GSC(L, 0)$ dilution-corrected (base 10 log) |
|---|---|---|
| batch-ReLU (i) | 0.337 | 4.23 |
| batch-ReLU (ii) | 0.329 | 4.06 |
| batch-ReLU (iii) | 6.164 | 68.37 |
| batch-ReLU (iv) | 0.313 | 7.22 |
| layer-tanh (i) | 0.136 | 2.17 |
| layer-tanh (ii) | 0.114 | 1.91 |
| layer-tanh (iii) | 3.325 | 5.46 |
| layer-tanh (iv) | 0.143 | 2.31 |

Table 3: Key metrics for architectures derived from dynamical systems theory.

We evaluated those strategies empirically. In table 3, we show the value of the GSC across the network for 8 different architectures in their initialized state applied to Gaussian noise. (See section B.1.2 for details.) All architectures use residual blocks containing a single normalization layer, a single nonlinearity layer and a single linear layer. We initialize the linear layer in four different ways: (i) Gaussian initialization, (ii) skew-symmetric initialization, (iii) initialization as $-C^T C$ where C is Gaussian initialized and (iv) Gaussian initialization where weight matrices in successive blocks have correlation 0.5. Initializations (ii), (iii) and (iv) mimic strategies (a), (b) and (d) respectively. To enable the comparison of the four initialization styles, we normalize each weight matrix to have a unit qm norm. We study all four initializations for both batch-ReLU and layer-tanh.

Initialization (ii) improves slightly over initialization (i). This is expected given theorem 3. One of the key assumptions is that skip and residual function be orthogonal in expectation. While initialization (i) achieves this, under (ii), the two functions are orthogonal not just in expectation, but with probability 1.

Initialization (iii) has gradients that grow much faster than initialization (i). On the one hand, this is surprising as Haber & Ruthotto (2017) states that eigenvalues with negative real parts in the residual Jacobian supposedly slow gradient growth. On the other hand, it is not surprising because introducing correlation between the residual and skip path breaks the conditions of theorem 3.

Initialization (iv) performs comparably to initialization (i) in reducing gradient growth, but requires a larger amount of dilution to achieve this result. Again, introducing correlation between successive blocks and thus between skip and residual function breaks the conditions of theorem 3 and weakens the power of dilution.

While we did not investigate the exact architectures proposed in Haber & Ruthotto (2017); Chang et al. (2017), our results show that more theoretical and empirical evaluation is necessary to determine whether architectures based on (a), (b) and (d) are indeed capable of increasing stability. Of course, those architectures might still confer benefits in terms of e.g. inductive bias or regularization.

Finally, strategy (c), the scaling of either residual and/or skip function with constants is a technique already widely used in regular ResNets. In fact, our study suggests that in order to bound the GSC at arbitrary depth in a regular ResNet, it is sufficient to downscale each residual function by only $\frac{1}{\sqrt{B}}$ instead of $\frac{1}{B}$ as papers in this line of work suggest.

## B.2 VANISHING GRADIENTS IN FEEDFORWARD NETWORKS

We have not experienced vanishing gradients as defined by the GSC in our experiments. Our analysis suggests that strong domain collapse is necessary to not only overcome the gradient growth implied by theorem 2, but reverse it. We conjecture that such domain collapse could actually occur in e.g. ReLU and tanh architecture if a non-zero additive bias was introduced, though this goes beyond the scope of this paper.

## B.3 EXPLODING AND VANISHING GRADIENTS IN RNNs

Exploding gradients and their counterpart, vanishing gradients, have been studied more extensively in the context of on RNNs (e.g. Pascanu et al. (2013); Bengio et al. (1994)). It is important to note that the problem as it arises in RNNs is similar but also different from the exploding gradient problem in feedforward networks. The goal in RNNs is often to absorb information early on and store that information through many time steps and sometimes indefinitely. In the classical RNN architecture, signals acquired early would be subjected to a non-orthogonal transformation at every time step which leads to all the negative consequences described in this paper. LSTMs (Hochreiter & Schmidhuber, 1997) and GRUs (Cho et al., 2014), which are the most popular solutions to exploding / vanishing gradients in RNNs, are capable of simply leaving each neuron that is considered part of the latent state completely unmodified from time step to time step unless new information is received that is pertinent to that specific neuron. This solution does not apply in feedforward networks, because it is the very goal of each layer to modify the signal productively. Hence, managing exploding gradients in feedforward networks is arguably more difficult.

Nevertheless, there is similarity between LSTM and the orthogonal initial state because both eliminate non-orthogonality "as much as possible". LSTM can eliminate non-orthogonality completely from time step to time step whereas in the orthogonal initial state, non-orthogonality is eliminated only from the initial function. Again, viewing feedforward networks as ensembles of shallower networks, orthogonal initial functions ensure that information extracted from each ensemble member does not have to pass through non-orthogonal transformations needlessly. This is precisely what LSTM attempts to achieve.

## B.4 OPEN RESEARCH QUESTIONS AND FUTURE WORK

**Biases, convolutional and recurrent layers**    In this paper, we focus our analysis on MLPs without trainable bias and variance parameters. Theorem 1, in its formulation, applies only to such MLPs. Theorems 2, 3 and proposition 7 use assumptions that are potentially harder to achieve in non-MLP architectures. Our experimental evaluation is limited to MLPs.

We think that results very similar to those presented in this paper are acheivable for other types of neural networks, such as those containing trainable biases, convolutional layers or recurrent layers.

The fundamental behavior of those architectures should be the same, though additional nuance and heavier mathematical bookkeeping might come into play.

Analysis of deep gradients has so far focused on MLPs (e.g. Balduzzi et al. (2017); Schoenholz et al. (2017); Yang & Schoenholz (2017); Saxe et al. (2014)), so a principled extension of these results to other network types would break new and important ground.

**Understanding collapsing domains**   It is difficult to assess or measure the degree to which the domain collapses in a given network. There is no single correct metric to measure this effect and depending on the metric chosen, the set of networks exhibiting collapsing domains may look very different. So far, we discussed pre-activation standard deviation (section 6), pre-activation sign diversity (section 6) and activation correlation (section B.1.1). The volume of the domain and the entropy of the distribution of activations may also be of interest.

Not all domains collapse in the same way. In the tanh architecture we studied in this paper, the domain collapses onto the origin. In linear MLPs, the domain collapses onto the line through the dominant eigenvector of the product of weight matrices, but never collapses onto a single point. In ReLU, the domain collapses onto a ray from the origin. In tanh with very large weights, activation vectors are mapped approximately onto the corners of the hypercube by the tanh layer.

**What gradient scale is best?**   $GSC(1, L)$ indicates the relative responsiveness of the prediction layer with respect to changes in the input layer. Of course, the goal in deep learning, at least within a prediction framework, is to model some ground truth function $t$ that maps data inputs to true labels. That function has itself a GSC at each input location $x$ that measures the relative responsiveness of $t(x)$ to changes in $x$. If the network is to perfectly represent the ground truth function, the GSCs would also have to match up. If, on the other hand, the GSC of the network differs significantly from that of $t$, the network is not fitting $t$ well. This suggests that in fact, the "best" value of the GSC is one that matches that of the ground truth. If the GSC of the network is too low, we may experience underfitting. If the GSC of the network is too high, we may experience overfitting.

**How to achieve the "right" gradient?**   To model the ground truth function, we may not just want to consider the overall magnitude of the GSC across the dataset, but to enable the network to have gradients of different magnitudes from one data input to the next; or to learn highly structured gradients. For example, given an image of a dog standing in a meadow, we might desire a high gradient with respect to pixels signifying e.g. facial features of the dog but a low gradient with respect to pixels that make up the meadow, and a uniformly low gradient given an image of a meadow. Such gradients would be very valuable not just in modelling real world functions more accurately and improving generalization, but in making the output of neural networks more explainable and avoiding susceptibility to attacks with adversarial inputs.

**Understanding representational capacity and pseudo-linearity**   In section 7, we explained how dilution reduces gradient growth but may harm representation capacity. While removing untrainable non-orthogonality from the initial functions may not be harmful, to achieve large levels of dilution and thus large amounts of GSC reduction, many ResNet architectures also suppress the size of the residual function relative to the initial function. This may happen naturally when the size of the skip path grows via repeated addition as we go deeper, or by deliberately scaling down residual blocks (Szegedy et al., 2016). Clearly, if a neural network can only represent functions that are very close to linear functions, it may not be able to model the ground truth function well. However, there exist no mechanisms to determine how much dilution is harmful for a given ground truth function or dataset.

Dilution is not only present in ResNet and special constructs such as looks-linear initialized ReLU networks, but even in vanilla, Gaussian initialized MLPs. For example, a SeLU nonlinearity can be more easily approximated by a linear function in terms of mean square error over a unit Gaussian input than a ReLU nonlinearity. We suspect that this is related to gradients in a SeLU MLP exploding more slowly than in a batch-ReLU MLP. Assessing the total amount of "linearity" present in a network is an open question. Therefore, we also cannot make blanket statements such as "SeLU is superior to batch-ReLU because gradients explode more slowly", because a batch-ReLU MLP with fewer layers might in some sense have as much representational power as a SeLU MLP with more layers.

Finally, the impact of dilution on the representational power conferred by depth is an open question.

**How far does the orthogonal initial state take us?**    An orthogonal initial state reduces gradients via dilution, which allows for relatively larger updates, which enables increased growth of residual functions, which allows for greater effective depth. However, as residual functions grow, dilution decreases, so the gradient increases, so updates must shrink, so the growth of residual functions slows, so the growth of effective depth slows.

In other words, for the network to become deeper, it needs to be shallow.

Therefore, while training from an orthogonal initial state can increase effective depth, we expect this effect to be limited. Additional techniques could be required to learn functions which require a compositional representation beyond this limit.

## C    FURTHER TERMINOLOGY, NOTATION AND CONVENTIONS

- $x$ and $y$ are generally used to refer to the components of a datapoint. Then, we have $(x, y) \in D$.
- $X$ refers to a vector of dimension $d$, i.e. the same dimension as the $x$ component of datapoints. Similarly, $Y$ refers to an element of the domain of possible labels. We call $X$ a 'data input' and $Y$ a 'label input'.
- $F_l$ refers to a vector of dimension $d_l$, i.e. the same dimension as $f_l$.
- We write $f_l(\theta, x)$ as a short form of $f_l(\theta_l, f_{l+1}(..(f_L(\theta_L, x))..))$. Sometimes, we omit $x$ and / or $\theta$. In that case, $x$ and / or $\theta$ remains implicit. $f_l(\theta, X)$ is an analogous short form.
- We write $f_l(\theta, f_k)$ as a short form of $f_l(\theta_l, f_{l+1}(..f_{k-1}(\theta_{k-1}, f_k)..))$. Sometimes, we omit $f_k$ and / or $\theta$. In that case, $f_k$ and / or $\theta$ remain implicit. $f_l(\theta, F_k)$ is an analogous short form.
- We use $f_{L+1}$, $i_{L+1}$ and $F_{L+1}$ interchangeably with $x$ or $X$.
- We say a random vector is 'radially symmetric' if its length is independent of its orientation and its orientation is uniformly distributed.
- We say a random matrix is 'Gaussian initialized' if its entries are independent Gaussian random variables with mean zero and the standard deviation of all entries is the same.
- We say an $m * n$ random matrix is 'orthogonally initialized' if it is a fixed multiple of an $m * n$ submatrix of a $\max(m, n) * \max(m, n)$ uniformly random orthogonal matrix.
- We use parentheses () to denote vector and matrix elements, i.e. $A(3, 4)$ is the fourth element in the third row of the matrix $A$.
- Throughout sections E and F, we assume implicitly that the GSC is defined and thus that neural networks are differentiable. All results can be trivially extended to cover networks that are almost surely differentiable and directionally differentiable everywhere, which includes SeLU and ReLU networks.
- All theoretical results apply to arbitrary networks, not just MLPs, unless otherwise stated. However, some assumptions arising in proposition 7 and theorems 2 and 3 may be less easy to achieve in general architectures. We focus our discussion of these assumptions exclusively on the MLPs within the scope of this paper as outlined at the end of section 2.

## D    EFFECTIVE DEPTH: DETAILS

### D.1    FORMAL DEFINITION

Let a 'gradient-based algorithm' for training a mutable parameter vector $\theta$ from an initial value $\theta^{(0)}$ for a network $f$ be defined as a black box that is able to query the gradient $\frac{df(\theta, X, Y)}{d\theta}$ at arbitrary query points $(X, Y)$ but only at the current value of the mutable parameter vector $\theta$. It is able to generate updates $\Delta\theta$ which are added to the mutable parameter vector $\theta$. Let the sequence of updates be

denoted as $\Delta\theta^{(1)}, \Delta\theta^{(2)}, ...$ We define the successive states of $\theta$ recursively as $\theta^{(t)} = \theta^{(t-1)} + \Delta\theta^{(t)}$. For simplicity, assume the algorithm is deterministic.

In a residual network defined according to equation 2, we can write the gradient with respect to a parameter sub-vector as $\frac{df(\theta,X,Y)}{d\theta_l} = \frac{df_0}{df_1}\frac{df_1}{df_2}..\frac{df_{l-1}}{df_l}\frac{df_l}{d\theta_l} = \frac{df_0}{df_1}\left(\frac{di_1}{df_2} + \frac{dr_1}{df_2}\right)..\left(\frac{di_{l-1}}{df_l} + \frac{dr_{l-1}}{df_l}\right)\frac{df_l}{d\theta_l}$. Multiplying this out, we obtain $2^{l-1}$ terms. We call a term '$\lambda$-residual' if it contains $\lambda$ or more Jacobians of residual functions, as opposed to Jacobians of initial functions. Let $res_l^\lambda(f,\theta,X,Y)$ be the sum of all $\lambda$-residual terms in $\frac{df(\theta,X,Y)}{d\theta_l}$.

Now consider two scenarios. In scenario (1), when the algorithm queries the gradient, it receives $\{\frac{df(\theta,X,Y)}{d\theta_1}, \frac{df(\theta,X,Y)}{d\theta_2}, .., \frac{df(\theta,X,Y)}{d\theta_L}\}$ i.e. the "regular" gradient. In scenario (2), it receives $\{\frac{df(\theta,X,Y)}{d\theta_1} - res_1^\lambda(f,\theta,X,Y), \frac{df(\theta,X,Y)}{d\theta_2} - res_2^\lambda(f,\theta,X,Y), .., \frac{df(\theta,X,Y)}{d\theta_L} - res_L^\lambda(f,\theta,X,Y)\}$, i.e. a version of the gradient where all $\lambda$-residual terms are removed. Let the parameter vector attain states $\theta^{(1)}, \theta^{(2)}, ..$ in scenario (1) and $\theta^{(1,\lambda)}, \theta^{(2,\lambda)}, ..$ in scenario (2). Then we say the '$\lambda$-contribution' at time $t$ is $\theta^{(t)} - \theta^{(t,\lambda)}$. Finally, we say the 'effective depth at time $t$ with threshold $h$' is the largest $\lambda$ such that there exists an $l$ with $||\theta_l^{(t)} - \theta_l^{(t,\lambda)}||_2 \geq h$.

There is no objectively correct value for the threshold $h$. In practice, we find that the $\lambda$-contribution decreases quickly when $\lambda$ is increased beyond a certain point. Hence, the exact value of $h$ is not important when comparing different networks by effective depth.

The impact that the shift $\theta_l^{(t)} - \theta_l^{(t,\lambda)}$ has on the output of the network is influenced by the scale of $\theta_l^{(t)}$ as well as $GSC(l,0)$. If those values vary enormously between layers, it may be advisable to set different thresholds for different layers.

### D.2 Computational estimate

Unfortunately, computing the effective depth measure is intractable as it would require computing exponentially many gradient terms. In this section, we explain how we estimate effective depth in our experiments.

In this paper, we train networks only by stochastic gradient descent with either a single step size for all layers or a custom step size for each layer. Our algorithm for computing effective depth assumes this training algorithm.

**Vanilla networks** Assume that the network is expressed as a residual network as in equation 2. Let $B$ be the batch size, let $c_l^{(t)}$ be the step size used at layer $l$ for the $t$'th update and let $((X^{(t,1)}, Y^{(t,1)}), (X^{(t,2)}, Y^{(t,2)}), .., (X^{(t,B)}, Y^{(t,B)}))$ be the batch of query points used to compute the $t$'th update. Then SGD computes

$$\Delta\theta_l^{(t)} = c_l^{(t)}\sum_b \frac{df_0(\theta^{(t-1)}, X^{(t,b)}, Y^{(t,b)})}{d\theta_l^{(t-1)}}$$
$$\theta_l^{(t)} = \theta_l^{(t-1)} + \Delta\theta_l^{(t)}$$

For any update $t$ and query point $b$, we estimate its $\lambda$-contribution at layer $l$ as follows.

```
1   arr := [1];
2   for k = 0 to l − 1 do
3       size = size(arr);
4       arr.push_back(arr[size − 1] * ||r_k||_op);
5       for i = size − 1 to 1 do
6           arr[i] = arr[i] * (||df_0/df_{k+1}||_2) / (||df_0/df_k||_2) + arr[i − 1] * ||r_k||_op;
7       end
8       arr[0] = arr[0] * (||df_0/df_{k+1}||_2) / (||df_0/df_k||_2);
9   end
10  out = 0;
11  for i = λ to size(arr) − 1 do
12      out = out + arr[λ];
13  end
14  return out * c_l^{(t)} * ||f_{l+1}||_2;
```

For unparametrized layers, $||r_k||_{op}$ is set to zero. For linear layers, it is the operator norm of the residual weight matrix. The final estimate of the length of the $\lambda$-contribution at layer $l$ for the entire training period is then simply the sum of the lengths of the estimated $\lambda$-contributions over all time points and query points.

The core assumption here is that applying the Jacobian of the initial function of a given layer will increase the lengths of all terms approximately equally, no matter how many residual Jacobians they contain. In other words, we assume that in $\lambda$-residual terms, the large singular values of layer-wise Jacobians do not compound disproportionately compared to other terms. This is similar to the core assumption in theorem 1 in section F.1.

We conservatively bound the impact of the Jacobian of the initial function with the impact of the Jacobian of the entire layer, i.e. $\frac{||\frac{df_0}{df_{k+1}}||_2}{||\frac{df_0}{df_k}||_2}$.

We use $||r_k||_{op}$ as a conservative estimate on how a residual Jacobian will increase the length of a term.

We use the sum of the lengths of all $\lambda$-residual terms in a batch as a conservative bound on the length of the $\lambda$-contribution of the batch. In essence, we assume that all $\lambda$-residual terms have the same orientation.

Finally, we use the sum of the lengths of the $\lambda$-contributions within each update as an estimate of the length of the total $\lambda$-contribution of the entire training period. On the one hand, this is conservative as we implicitly assume that the $\lambda$-contributions of each batch have the same orientation. On the other hand, we ignore indirect effects that $\lambda$-contributions in early batches have on the trajectory of the parameter value and hence on $\lambda$-contributions of later batches. Since we are ultimately interested in effective depth, we can ignore these second-order effects as they are negligible when the total $\lambda$-contribution is close to a small threshold $h$.

Overall, we expect that our estimate of the effective depth (e.g. figure 2D) is larger than its actual value. This is bolstered by the robustness of some of our trained networks to Taylor expansion (see figure 2F).

**ResNet** For ResNet architectures, we need to tweak our estimate of effective depth to take into account skip connections. Below, we detail how the variable $arr$ is modified as it crosses a skip connection / residual block. We write $f_n(f_m) = s_n(f_m) + \rho_n(f_m)$, where $f_n$ is the layer at which the skip connection terminates, $f_m$ is the layer at which the skip connection begins, $s_n$ is the function computed by the skip connection and $\rho_n(f_m) = \rho_n(f_{n+1}(..f_{m-1}(f_m)..))$ is the function computed

by the residual block. We write $f_k = i_k + r_k$ for $n + 1 \leq k \leq m - 1$ and $\rho_n = i_n + r_n$, i.e. we break down each layer in the residual block into an initial function and a residual function.

```
1   arr_copy = arr;
2   for k = n to m − 1 do
3       size = size(arr);
4       arr.push_back(arr[size − 1] * ||r_k||_op);
5       for i = size − 1 to 1 do
6           arr[i] = arr[i] * ||(df_0/dρ_n)(dρ_n/df_{k+1})||_2 / ||df_0/df_k||_2 + arr[i − 1] * ||r_k||_op;
7       end
8       arr[0] = arr[0] * ||(df_0/dρ_n)(dρ_n/df_{k+1})||_2 / ||df_0/df_k||_2;
9   end
10  for i = 0 to size(arr_copy) − 1 do
11      arr[i] = arr[i] + arr_copy[i] * (||df_0/df_m||_2 − ||(df_0/dρ_n)(dρ_n/df_m)||_2) / ||df_0/df_n||_2;
12  end
```

In line 11, the combined effect of the skip connection and the initial functions of the residual block is approximated by the effect of the entire block, i.e. $\frac{||\frac{df_0}{df_m}||_2}{||\frac{df_0}{df_n}||_2}$. In the same line, we must subtract the impact of the initial functions accumulated while passing through the residual block, i.e. $\frac{-||\frac{df_0}{d\rho_n}\frac{d\rho_n}{df_m}||_2}{||\frac{df_0}{df_n}||_2}$. The impact of the residual functions in the block is, correctly, unaffected by the skip connection and bounded by the operator norm, as before.

### D.3 DISCUSSION

The effective depth measure has several limitations.

One can train a linear MLP to have effective depth much larger than 1, but the result will still be equivalent to a depth 1 network.

Consider the following training algorithm: first randomly re-sample the weights, then apply gradient descent. Clearly, this algorithm is equivalent to just running gradient descent in any meaningful sense. The re-sampling step nonetheless blows up the residual functions so as to significantly increase effective depth.

The effective depth measure is very susceptible to the initial step size. In our experiments, we found that starting off with unnecessarily large step sizes, even if those step sizes were later reduced, lead to worse outcomes. However, because of the inflating impact on the residual function, the effective depth would be much higher nonetheless.

Effective depth may change depending on how layers are defined. In a ReLU MLP, for example, instead of considering a linear transformation and the following ReLU operation as different layers, we may define them to be part of the same layer. While the function computed by the network and the course of gradient-based training do not depend on such redefinition, effective depth can be susceptible to such changes.

## E PROPOSITIONS AND PROOFS

### E.1 PROPOSITION 1

**Proposition 1.** *Given:*

- *a neural network $f$ of nominal depth $L$*

- *an initial parameter value $\theta^{(0)}$*

- *a mutable parameter value $\theta$ that can take values in some closed, bounded domain $\Theta$*

- *a dataset $D$ of datapoints $(x, y)$*

- *a closed, bounded domain $\mathcal{D}$ of possible query points $(X, Y)$*

- *a function $||.||$ from matrices to the reals that has $c||.|| = ||c.||$ and $||.|| \geq 0$*

- *some deterministic algorithm that is able to query gradients of $f$ at the current parameter value and at query points in $\mathcal{D}$ and that is able to apply updates $\Delta\theta$ to the parameter value*

- *constant $r'$*

*Assume:*

- *Running the algorithm on $f$ with $\theta$ initialized to $\theta^{(0)}$ for a certain number of updates $T$ causes $\theta$ to attain a value $\hat{\theta}$ at which $f$ attains some error value $E_{final}$ on $D$.*

- *At every triplet $(\theta, X, Y) \in \Theta \times \mathcal{D}$, we have $||\mathcal{J}_k^l|| \neq 0$ and $||\mathcal{T}_k^l|| \neq 0$ for all $0 \leq l \leq k \leq L$.*

*Then we can specify some other neural network $f'$ and some other initial parameter value $\theta'^{(0)}$ such that the following claims hold:*

1. *$f'$ has nominal depth $L$ and the same compositional depth as $f$.*

2. *The algorithm can be used to compute $T$ updates by querying gradients of $f'$ at the current parameter value and at query points in $\mathcal{D}$ which cause $\theta$ to attain a value $\hat{\theta}'$ where $f'$ attains error $E_{final}$ on $D$ and makes the same predictions as $f(\hat{\theta})$ on $D$.*

3. *At every triplet $(\theta, X, Y) \in \Theta \times \mathcal{D}$, we have $||\mathcal{T'}_k^l|| \geq r'^{k-l}$ for all $0 \leq l \leq k \leq L$ and $||\mathcal{J'}_k^l|| \geq r'^{k-l}$ for all $0 \leq l \leq k \leq L$ except $(k, l) = (1, 0)$.*

*Proof.* Since $\Theta$ and $\mathcal{D}$ are closed and bounded, so is $\Theta \times \mathcal{D}$. Therefore for all $0 \leq l \leq k \leq L$, both $||\mathcal{J}_k^l||$ and $||\mathcal{T}_k^l||$ attain their infimum on that domain if it exists. $||.||$ is non-negative, so the infimum exists. $||.||$ is non-zero on the domain, so the infimum, and therefore the minimum, is positive. Since $f$ has finite depth, there is an $r$ such that for all tuplets $(\theta, X, Y, k, l)$, we have $||\mathcal{J}_k^l|| \geq r^{k-l}$ and $||\mathcal{T}_k^l|| \geq r^{k-l}$. Let $R = \frac{r'}{r}$.

Now, we define $f'$ via its layer functions.

$$
\begin{aligned}
f'_0 &= f_0 \\
f'_1(\theta_1, F_2) &= f_1(R\theta_1, R^2 F_2) \\
f'_l(\theta_l, F_{l+1}) &= R^{-l} f_l(R^l \theta_l, R^{l+1} F_{l+1}) \text{ for } 2 \leq l \leq L-1 \\
f'_L(\theta_L, X) &= R^{-L} f_L(R^L \theta_L, X)
\end{aligned}
$$

$f$ and $f'$ clearly have the same nominal and compositional depth, so claim (1) holds. Given any vector $v$ with $L$ sub-vectors, define the transformation $R(v)$ as $R(v)_l = R^l v_l$. Finally, we set $\theta'^{(0)} := R^{-1}(\theta^{(0)})$.

We use the algorithm to train $f'$ as follows. Whenever the algorithm queries some gradient value $\frac{df'}{d\theta}$, we instead submit to it the value $R^{-1}(\frac{df'}{d\theta})$. Whenever the algorithm wants to apply an update $\Delta\theta$ to the parameter, we instead apply $R^{-1}(\Delta\theta)$. Let $S'^{(t)}$, $0 \leq t \leq T$ be the state of the system after applying $t$ updates to $\theta$ under this training procedure. Let $S^{(t)}$, $0 \leq t \leq T$ be the state of the system after applying $t$ updates to $\theta$ when the algorithm is run on $f$. Then the following invariances hold.

A $\theta'^{(t)} = R^{-1}(\theta^{(t)})$, where $\theta'^{(t)}$ is the value of $\theta$ under $S'^{(t)}$ and $\theta^{(t)}$ is the value of $\theta$ under $S^{(t)}$.

B $f$ makes the same predictions and attains the same error on $D$ under $S^{(t)}$ as $f'$ under $S'^{(t)}$.

C Any state the algorithm maintains is equal under both $S^{(t)}$ and $S'^{(t)}$.

We will show these by induction. At time $t = 0$, we have $\theta'^{(0)} = R^{-1}(\theta^{(0)})$ as chosen, so (A) holds. It is easy to check that (B) follows from (A). Since the algorithm has thus far not received any inputs, (C) also holds.

Now for the induction step. Assuming that $\theta'^{(t)} = R^{-1}(\theta^{(t)})$, it is easy to check that $\frac{df'(\theta'^{(t)})}{d\theta} = R(\frac{df(\theta^{(t)})}{d\theta})$. Therefore, whenever the algorithm queries a gradient of $f'$, it will receive $R^{-1}(\frac{df'(\theta'^{(t)})}{d\theta}) = R^{-1}(R(\frac{df(\theta^{(t)})}{d\theta})) = \frac{df(\theta^{(t)})}{d\theta}$. Therefore, the algorithm receives the same inputs under both $S^{(t)}$ and $S'^{(t)}$. Since the internal state of the algorithm is also the same, and the algorithm is deterministic, the update returned by the algorithm is also the same and so is the internal state after the update is returned, which completes the induction step for (C). Because the algorithm returns the same update in both cases, after the prescribed post-processing of the update under $f'$, we have $\Delta\theta'^{(t)} = R^{-1}(\Delta\theta^{(t)})$. Therefore $\theta'^{(t+1)} = \theta'^{(t)} + \Delta\theta'^{(t)} = R^{-1}(\theta^{(t)}) + R^{-1}(\Delta\theta^{(t)}) = R^{-1}(\theta^{(t)} + \Delta\theta^{(t)}) = R^{-1}(\theta^{(t+1)})$. This completes the induction step for (A) and again, (B) follows easily from (A).

(B) implies directly that claim (2) holds. Finally, for any tuplet $(\theta, X, Y, k, l)$, we have $||\mathcal{T}'^l_k|| = ||\frac{df'_l(\theta'^{(t)})}{d\theta'^{(t)}_k}|| = ||R^{k-l}\frac{df_l(\theta^{(t)})}{d\theta^{(t)}_k}|| \geq R^{k-l}r^{k-l} = r'^{k-l}$ and unless $(k, l) = (1, 0)$ we have $||\mathcal{J}'^l_k|| = ||\frac{df'_l(\theta'^{(t)})}{df'_k}|| = ||R^{k-l}\frac{df_l(\theta^{(t)})}{df_k}|| \geq R^{k-l}r^{k-l} = r'^{k-l}$. Therefore, claim (3) also holds, which completes the proof.

$\square$

Notes:

- The condition that the Jacobians of $f$ always have non-zero norms may be unrealistic. For practical purposes, it should be enough to have Jacobians that mostly have non-zero norms. This leads to a network $f'$ that has exploding Jacobians wherever $f$ has Jacobians of size above some threshold, where that threshold can be arbitrarily chosen.

- Claim (3) of the proposition does not include the case $(k, l) = (1, 0)$ and it does not include Jacobians with respect to the input $X$. These Jacobians have to be the same between $f$ and $f'$ if we require $f'$ to have the same error and predictions as $f$. However, if we are ok with multiplicatively scaled errors and predictions, claim (3) can be extended to cover those two cases. Scaled training errors and predictions are generally not a problem in e.g. classification.

- Note that not only does the algorithm achieve the same predictions in the same number of updates for both $f$ and $f'$, but the computation conducted by the algorithm is also identical, so $f'$ is as "easy to train" as $f$ no matter how we choose to quantify this as long as we know to apply the scaling transformation.

- There are no constraints on the explosion rate $r'$. If we can successfully train a network with some explosion rate, we can successfully train an equivalent network with an arbitrary explosion rate.

- $f'$ is very similar to $f$, so this proposition can be used to construct trainable networks with exploding Jacobians of any shape and depth as long as there exists some trainable network of that shape and depth.

- The proposition can be easily be extended to non-deterministic algorithms by using distributions and expectations.

- The proposition can be easily extended to use directional derivatives instead of total derivatives to cover e.g. ReLU and SeLU nonlinearities.

### E.2 PROPOSITION 2

**Proposition 2.** *Let $U$ be the uniform distribution over the hypersphere. Then $GSC(k, l)$ measures the quadratic expectation $\mathbb{Q}$ of the relative size of the change in the value of $f_l$ in response to a change in $f_k$ that is a small multiple of a random variable drawn from $U$.*

*Equivalently, $GSC(k, l) = \lim_{\epsilon \to 0} \mathbb{Q}_{u \sim U} \frac{\frac{||f_l(f_k + \epsilon u) - f_l(f_k)||_2}{||f_l(f_k)||_2}}{\frac{||f_k + \epsilon u - f_k||_2}{||f_k||_2}}$*

*Proof.* We use $L\Sigma R^T$ to denote the singular value decomposition and $s_i$ to denote singular values.

$$\lim_{\epsilon \to 0} \mathbb{Q}_u \frac{\frac{||f_l(f_k + \epsilon u) - f_l(f_k)||_2}{||f_l(f_k)||_2}}{\frac{||f_k + \epsilon u - f_k||_2}{||f_k||_2}}$$

$$= \lim_{\epsilon \to 0} \mathbb{Q}_u \frac{||f_l(f_k + \epsilon u) - f_l(f_k)||_2 ||f_k||_2}{\epsilon ||f_l(f_k)||_2}$$

$$= \lim_{\epsilon \to 0} \frac{||f_k||_2}{\epsilon ||f_l(f_k)||_2} \mathbb{Q}_u ||f_l(f_k + \epsilon u) - f_l(f_k)||_2$$

$$= \lim_{\epsilon \to 0} \frac{||f_k||_2}{\epsilon ||f_l(f_k)||_2} \mathbb{Q}_u ||f_l(f_k) + \epsilon \mathcal{J}_k^l u + O(\epsilon^2) - f_l(f_k)||_2$$

$$= \lim_{\epsilon \to 0} \frac{||f_k||_2}{||f_l(f_k)||_2} \mathbb{Q}_u ||\mathcal{J}_k^l u + \frac{O(\epsilon^2)}{e}||_2$$

$$= \frac{||f_k||_2}{||f_l(f_k)||_2} \mathbb{Q}_u ||\mathcal{J}_k^l u||_2$$

$$= \frac{||f_k||_2}{||f_l(f_k)||_2} \mathbb{Q}_u ||L\Sigma R^T u||_2$$

$$= \frac{||f_k||_2}{||f_l(f_k)||_2} \mathbb{Q}_u ||\Sigma u||_2$$

$$= \frac{||f_k||_2}{||f_l(f_k)||_2} \mathbb{Q}_u \sqrt{\sum_{i=1}^{\min(d_k, d_l)} s_i^2 u(i)^2}$$

$$= \frac{||f_k||_2}{||f_l(f_k)||_2} \sqrt{\mathbb{E}_u \sum_{i=1}^{\min(d_k, d_l)} s_i^2 u(i)^2}$$

$$= \frac{||f_k||_2}{||f_l(f_k)||_2} \sqrt{\sum_{i=1}^{\min(d_k, d_l)} s_i^2 \mathbb{E}_u u(i)^2}$$

$$= \frac{||f_k||_2}{||f_l(f_k)||_2} \sqrt{\frac{1}{d_k} \sum_{i=1}^{\min(d_k, d_l)} s_i^2}$$

$$= \frac{||f_k||_2}{||f_l(f_k)||_2} ||\Sigma||_{qm}$$

$$= \frac{||f_k||_2}{||f_l(f_k)||_2} ||\mathcal{J}_k^l||_{qm}$$

$$= GSC(k, l)$$

$\square$

### E.3 PROPOSITION 3

**Proposition 3.** *Let $U$ be the uniform distribution over the hypersphere. Assume $f_k$ is a fully-connected linear layer without trainable bias parameters and $\theta_k$ contains the entries of the weight matrix. Then $GSC(k, l) \frac{||\theta_k||_2 ||f_{k+1}||_2}{||f_k||_2 \sqrt{d_{k+1}}}$ measures the quadratic expectation $\mathbb{Q}$ of the relative size of the change in the value of $f_l$ in response to a change in $\theta_k$ that is a small multiple of a random variable drawn from $U$.*

*Equivalently, $GSC(k, l) \frac{||\theta_k||_2 ||f_{k+1}||_2}{||f_k||_2 \sqrt{d_{k+1}}} = \lim_{\epsilon \to 0} \mathbb{Q}_{u \sim U} \frac{\frac{||f_l(f_k(\theta_k + \epsilon u, f_{k+1})) - f_l(f_k(\theta_k, f_{k+1}))||_2}{||f_l||_2}}{\frac{||\theta_k + \epsilon u - \theta_k||_2}{||\theta_k||_2}}$*

*Further, if $\theta_k$ is random and*

- *all entries of $\theta_k$ have the same quadratic expectation*
- *all products of two different entries of $\theta_k$ have an expectation of 0*
- *the orientation of $\theta_k$ is independent of its length*

*we have $\mathbb{Q}_{\theta_k}^{-1} \frac{||\theta_k||_2 ||f_{k+1}||_2}{||f_k||_2 \sqrt{d_{k+1}}} = 1$.*

*Proof.* Throughout this derivation, we will use $\theta_k$ to refer to both the parameter sub-vector and the weight matrix. Similarly, we will use $\epsilon u$ to refer to both a perturbation of the parameter sub-vector and of the weight matrix. We use $L \Sigma R^T$ to denote the singular value decomposition and $s_i$ to denote singular values.

$$
\lim_{\epsilon \to 0} \mathbb{Q}_u \frac{\frac{||f_l(f_k(\theta_k + \epsilon u, f_{k+1})) - f_l(f_k(\theta_k, f_{k+1}))||_2}{||f_l||_2}}{\frac{||\theta_k + \epsilon u - \theta_k||_2}{||\theta_k||_2}}
$$

$$
= \lim_{\epsilon \to 0} \frac{||\theta_k||_2}{\epsilon ||f_l||_2} \mathbb{Q}_u ||f_l(f_k(\theta_k + \epsilon u, f_{k+1})) - f_l(f_k(\theta_k, f_{k+1}))||_2
$$

$$
= \lim_{\epsilon \to 0} \frac{||\theta_k||_2}{\epsilon ||f_l||_2} \mathbb{Q}_u ||f_l((\theta_k + \epsilon u) f_{k+1}) - f_l(\theta_k f_{k+1})||_2
$$

$$
= \lim_{\epsilon \to 0} \frac{||\theta_k||_2}{\epsilon ||f_l||_2} \mathbb{Q}_u ||f_l(\theta_k f_{k+1} + \epsilon u f_{k+1}) - f_l(\theta_k f_{k+1})||_2
$$

$$
= \lim_{\epsilon \to 0} \frac{||\theta_k||_2}{\epsilon ||f_l||_2} \mathbb{Q}_u ||f_l(\theta_k f_{k+1}) + \mathcal{J}_k^l \epsilon u f_{k+1} + O(\epsilon^2) - f_l(\theta_k f_{k+1})||_2
$$

$$
= \lim_{\epsilon \to 0} \frac{||\theta_k||_2}{||f_l||_2} \mathbb{Q}_u ||\mathcal{J}_k^l u f_{k+1} + \frac{O(\epsilon^2)}{\epsilon}||_2
$$

$$
= \frac{||\theta_k||_2}{||f_l||_2} \mathbb{Q}_u ||\mathcal{J}_k^l u f_{k+1}||_2
$$

$$
= \frac{||\theta_k||_2}{||f_l||_2} \mathbb{Q}_u ||L \Sigma R^T u f_{k+1}||_2
$$

$$
= \frac{||\theta_k||_2}{||f_l||_2} \mathbb{Q}_u ||\Sigma u f_{k+1}||_2
$$

$$
= \frac{||\theta_k||_2}{||f_l||_2} \mathbb{Q}_u \sqrt{\sum_{i=1}^{\min(d_k, d_l)} (\sum_{j=1}^{d_{k+1}} s_i u(i, j) f_{k+1}(j))^2}
$$

$$
= \frac{||\theta_k||_2}{||f_l||_2} \sqrt{\mathbb{E}_u \sum_{i=1}^{\min(d_k, d_l)} (\sum_{j=1}^{d_{k+1}} s_i u(i, j) f_{k+1}(j))^2}
$$

$$= \frac{||\theta_k||_2}{||f_l||_2} \sqrt{\mathbb{E}_u \sum_{i=1}^{\min(d_k,d_l)} \sum_{j=1}^{d_{k+1}} \sum_{m=1}^{d_{k+1}} s_i^2 u(i,j) f_{k+1}(j) u(i,m) f_{k+1}(m)}$$

$$= \frac{||\theta_k||_2}{||f_l||_2} \sqrt{\sum_{i=1}^{\min(d_k,d_l)} \sum_{j=1}^{d_{k+1}} \sum_{m=1}^{d_{k+1}} s_i^2 f_{k+1}(j) f_{k+1}(m) \mathbb{E}_u u(i,j) u(i,m)}$$

$$= \frac{||\theta_k||_2}{||f_l||_2} \sqrt{\sum_{i=1}^{\min(d_k,d_l)} \sum_{j=1}^{d_{k+1}} \sum_{m=1}^{d_{k+1}} s_i^2 f_{k+1}(j) f_{k+1}(m) \delta_{jm} \mathbb{E}_u u(i,j)^2}$$

$$= \frac{||\theta_k||_2}{||f_l||_2} \sqrt{\sum_{i=1}^{\min(d_k,d_l)} \sum_{j=1}^{d_{k+1}} s_i^2 f_{k+1}(j)^2 \mathbb{E}_u u(i,j)^2}$$

$$= \frac{||\theta_k||_2}{||f_l||_2} \sqrt{\frac{1}{d_k d_{k+1}} \sum_{i=1}^{\min(d_k,d_l)} \sum_{j=1}^{d_{k+1}} s_i^2 f_{k+1}(j)^2}$$

$$= \frac{||\theta_k||_2}{||f_l||_2 \sqrt{d_{k+1}}} \sqrt{\frac{1}{d_k} \sum_{i=1}^{\min(d_k,d_l)} s_i^2 \sum_{j=1}^{d_{k+1}} f_{k+1}(j)^2}$$

$$= \frac{||\theta_k||_2}{||f_l||_2 \sqrt{d_{k+1}}} ||\Sigma||_{qm} ||f_{k+1}||_2$$

$$= \frac{||\theta_k||_2}{||f_l||_2 \sqrt{d_{k+1}}} ||\mathcal{J}_k^l||_{qm} ||f_{k+1}||_2$$

$$= GSC(k,l) \frac{||\theta_k||_2 ||f_{k+1}||_2}{||f_k||_2 \sqrt{d_{k+1}}}$$

Further, assume that $\theta_k$ is random and fulfills the conditions stated. Under those conditions, $\theta_k$ is the product of a random scalar length variable $\ell$ and an independent random vector orientation variable $\theta_k'$ of unit length. Then for all $1 \leq i \leq d_k$ and $1 \leq j \leq d_{k+1}$, we have $\mathbb{E}\theta_k(i,j)^2 = \mathbb{E}\ell^2 \theta_k'(i,j)^2 = \mathbb{E}_\ell \ell^2 \mathbb{E}_{\theta'} \theta_k'(i,j)^2$ and so $\mathbb{Q}\theta_k'(i,j) = \frac{\mathbb{Q}\theta_k(i,j)}{\mathbb{Q}\ell}$. Since all entries of $\theta_k$ have the same quadratic expectation, all entries in $\theta_k'$ have the same quadratic expectation. Further, $1 = ||\theta_k'||_2 = \mathbb{Q}||\theta_k'||_2 = \sqrt{\mathbb{E}\sum_{i,j} \theta_k'(i,j)^2} = \sqrt{d_k d_{k+1} \mathbb{E}\theta_k'(1,1)^2} = \sqrt{d_k d_{k+1}} \mathbb{Q}\theta_k'(1,1)$, so the quadratic expectation of each entry of $\theta_k'$ is $\frac{1}{\sqrt{d_k d_{k+1}}}$. Further, for all $1 \leq i_1, i_2 \leq d_k$ and $1 \leq j_1, j_2 \leq d_{k+1}$ with $(i_1, j_1) \neq (i_2, j_2)$, we have $0 = \mathbb{E}\theta_k(i_1, j_1)\theta_k(i_2, j_2) = \mathbb{E}\theta_k'(i_1, j_1)\ell\theta_k'(i_2, j_2)\ell = \mathbb{E}_\ell \ell^2 \mathbb{E}_{\theta'} \theta_k'(i_1, j_1)\theta_k'(i_2, j_2)$, so the expectation of the product of two different entries of $\theta_k'$ is 0.

Then, we have:

$$\mathbb{Q}_{\theta_k}^{-1} \frac{||\theta_k||_2 ||f_{k+1}||_2}{||f_k||_2 \sqrt{d_{k+1}}}$$

$$= \frac{||f_{k+1}||_2}{\sqrt{d_{k+1}}} \mathbb{Q}^{-1} \frac{||\theta_k||_2}{||f_k||_2}$$

$$= \frac{||f_{k+1}||_2}{\sqrt{d_{k+1}}} \mathbb{Q}^{-1} \sqrt{\frac{||\theta_k||_2^2}{||f_k||_2^2}}$$

$$= \frac{||f_{k+1}||_2}{\sqrt{d_{k+1}}} \mathbb{Q}^{-1} \sqrt{\frac{||\theta_k||_2^2}{||\theta_k f_{k+1}||_2^2}}$$

$$= \frac{||f_{k+1}||_2}{\sqrt{d_{k+1}}} \sqrt{\mathbb{E} \frac{||\theta_k f_{k+1}||_2^2}{||\theta_k||_2^2}}^{-1}$$

$$
\begin{aligned}
&= \frac{||f_{k+1}||_2}{\sqrt{d_{k+1}}} \sqrt{\mathbb{E} \frac{||\ell\theta'_k f_{k+1}||_2^2}{||\ell\theta'_k||_2^2}}^{-1} \\
&= \frac{||f_{k+1}||_2}{\sqrt{d_{k+1}}} \sqrt{\mathbb{E} \frac{||\theta'_k f_{k+1}||_2^2}{||\theta'_k||_2^2}} \\
&= \frac{||f_{k+1}||_2}{\sqrt{d_{k+1}}} \sqrt{\mathbb{E}||\theta'_k f_{k+1}||_2^2}^{-1} \\
&= \frac{||f_{k+1}||_2}{\sqrt{d_{k+1}}} \sqrt{\mathbb{E} \sum_i (\sum_j \theta'_k(i,j) f_{k+1}(j))^2}^{-1} \\
&= \frac{||f_{k+1}||_2}{\sqrt{d_{k+1}}} \sqrt{\mathbb{E} \sum_{i,j,m} \theta'_k(i,j) f_{k+1}(j) \theta'_k(i,m) f_{k+1}(m)}^{-1} \\
&= \frac{||f_{k+1}||_2}{\sqrt{d_{k+1}}} \sqrt{\mathbb{E} \sum_{i,j} \theta'_k(i,j)^2 f_{k+1}(j)^2}^{-1} \\
&= \frac{||f_{k+1}||_2}{\sqrt{d_{k+1}}} \sqrt{\sum_j f_{k+1}(j)^2 \sum_i \mathbb{E}\theta'_k(i,j)^2}^{-1} \\
&= \frac{||f_{k+1}||_2}{\sqrt{d_{k+1}}} \sqrt{\sum_j f_{k+1}(j)^2 \sum_i (\mathbb{Q}\theta'_k(i,j))^2}^{-1} \\
&= \frac{||f_{k+1}||_2}{\sqrt{d_{k+1}}} \sqrt{\sum_j f_{k+1}(j)^2 \sum_i \frac{1}{d_k d_{k+1}}}^{-1} \\
&= \frac{||f_{k+1}||_2}{\sqrt{d_{k+1}}} \sqrt{||f_{k+1}||_2^2 d_k \frac{1}{d_k d_{k+1}}}^{-1} \\
&= 1
\end{aligned}
$$

$\square$

The conditions stated in the proposition for the random parameter sub-vector $\theta_k$ are fulfilled, for example, if the corresponding weight matrix is either Gaussian initialized or orthogonally initialized. Therefore, the most popular initialization strategies for weight matrices are covered by this proposition.

### E.4  PROPOSITION 4

**Proposition 4.** *Given:*

- *some network $f$ of nominal depth $L$*

- *some parameter value $\theta = (\theta_1, .., \theta_L)$*

- *constants $c_2, .., c_L$ and $\gamma_1, .., \gamma_L$*

- *a network $f'$ of nominal depth $L$ defined via its layer functions as follows.*

$$
\begin{aligned}
f'_0 &= f_0 \\
f'_1(\theta_1, F_2) &= f'_1(\gamma_1 \theta_1, c_2 F_2) \\
f'_l(\theta_l, F_{l+1}) &= c_l f_l(\gamma_l \theta_l, \frac{1}{c_{l+1}} F_{l+1}) \, for \, 2 \le l \le L - 1 \\
f'_L(\theta_L, X) &= c_L f_L(\gamma_L \theta_L, X)
\end{aligned}
$$

- *a parameter value $\theta' = (\theta'_1, .., \theta'_L)$ defined via $\theta'_l = \frac{1}{\gamma_l}\theta_l$*

*Then for all tuples $(\theta, \theta', X, Y)$, $GSC(k, l, f, \theta, X, Y) = GSC(k, l, f', \theta', X, Y)$.*

*Proof.* Let $c_0 = c_1 = 1$. Then we have $||f'_l||_2 = c_l||f_l||_2$ for $0 \leq l \leq L$ and we have $||\mathcal{J}'^l_k||_{qm} = ||\frac{c_l}{c_k}\mathcal{J}'^l_k||_{qm} = \frac{c_l}{c_k}||\mathcal{J}'^l_k||_{qm}$ for $0 \leq l \leq k \leq L$, so $GSC'(k, l) = \frac{||\mathcal{J}'^l_k||_{qm}||f'_k||_2}{||f'_l||_2} = \frac{\frac{c_l}{c_k}||\mathcal{J}^l_k||_{qm}c_k||f_k||_2}{c_l||f_l||_2} = GSC(k, l)$ for $0 \leq l \leq k \leq L$ as required.

$\square$

Here, we consider general multiplicative rescalings provided they do not change the predictions and error values of the network. To ensure this, each layer function must compensate for the factor introduced by the previous layer as well as for the rescaling of the parameter. Not all network transformations that are used in practice to control the scale of forward activations fall under this proposition. Changing the scale of weights in a tanh or SeLU network or adding normalization layers is not covered. These changes can have a drastic impact on the high-level properties of the network, as shown throughout the paper. On the other hand, changing the scale of weights in a ReLU network is covered by the proposition, as long as the error layer $f_0$ also compensates for this rescaling. Also, changing the scale of weights in any architecture where linear layers are followed by a normalization layer is covered by the proposition.

## E.5 PROPOSITION 5

**Proposition 5.** *Assuming the approximate decomposability of the norm of the product of Jacobians, i.e. $||\mathcal{J}^l_{l+1}\mathcal{J}^{l+1}_{l+2}..\mathcal{J}^{k-1}_k||_{qm} \approx ||\mathcal{J}^l_{l+1}||_{qm}||\mathcal{J}^{l+1}_{l+2}||_{qm}..||\mathcal{J}^{k-1}_k||_{qm}$, we have $GSC(k, l) \approx GSC(k, k-1)GSC(k-1, k-2)..GSC(l+1, l)$.*

*Proof.*

$$
\begin{aligned}
&\quad GSC(k, l) \\
&= \frac{||\mathcal{J}^l_k||_{qm}||f_k||_2}{||f_l||_2} \\
&= \frac{||\mathcal{J}^l_{l+1}\mathcal{J}^{l+1}_{l+2}..\mathcal{J}^{k-1}_k||_{qm}||f_k||_2}{||f_l||_2} \\
&\approx \frac{||\mathcal{J}^l_{l+1}||_{qm}||\mathcal{J}^{l+1}_{l+2}||_{qm}..||\mathcal{J}^{k-1}_k||_{qm}||f_k||_2}{||f_l||_2} \\
&= \frac{||\mathcal{J}^l_{l+1}||_{qm}||\mathcal{J}^{l+1}_{l+2}||_{qm}..||\mathcal{J}^{k-1}_k||_{qm}||f_k||_2}{||f_l||_2}\frac{||f_{l+1}||_2}{||f_{l+1}||_2}\frac{||f_{l+2}||_2}{||f_{l+2}||_2}..\frac{||f_{k-1}||_2}{||f_{k-1}||_2} \\
&= \frac{||\mathcal{J}^l_{l+1}||_{qm}||f_{l+1}||_2}{||f_l||_2}\frac{||\mathcal{J}^{l+1}_{l+2}||_{qm}||f_{l+2}||_2}{||f_{l+1}||_2}..\frac{||\mathcal{J}^{k-1}_k||_{qm}||f_k||_2}{||f_{k-1}||_2} \\
&= GSC(k, k-1)GSC(k-1, k-2)..GSC(l+1, l)
\end{aligned}
$$

$\square$

## E.6 PROPOSITION 6

**Proposition 6.** *Any endomorphism on the hypersphere composed of (i) a strictly monotonic, continuous nonlinearity $\sigma$ that has $\sigma(0) = 0$, (ii) multiplication with a full-rank matrix and (iii) length-only layer normalization is bijective.*

*Proof.* We will prove this by showing that the inverse image of any point under such an endomorphism is a single point. Take any point on the hypersphere. The inverse image under length-only layer normalization is a ray from the origin not including the origin. The inverse image of this ray under multiplication with a full-rank matrix is also a ray from the origin not including the origin.

What remains to be shown is that the inverse image of this ray under the nonlinearity layer, when intersected with the hypersphere, yields a single point. We will show this via a series of claims. Let the dimension of the hypersphere be $d$ and its radius be $r$.

*Claim 1*: If a point on this ray has an inverse image, that inverse image is a single point.

Let this point on the ray be $x$. Assume its inverse image contains two points $y$ and $z$. Then for $1 \leq i \leq d$, $\sigma(y(i)) = x(i)$ and $\sigma(z(i)) = x(i)$ and so $\sigma(y(i)) = \sigma(z(i))$. But $y \neq z$, so there exists an $i$ such that $y(i) \neq z(i)$. So there exist two different values $y(i)$ and $z(i)$ at which $\sigma$ returns the same result. But $\sigma$ is strictly monotonic. Contradiction.

*Claim 2*: If two points $x_1$ and $x_2$ on the ray have inverse images $y_1$ and $y_2$ and $x_1$ is closer to the origin than $x_2$, then for $1 \leq i \leq d$, we have $|y_1(i)| \leq |y_2(i)|$.

For $1 \leq i \leq d$, $\sigma$ attains $x_2(i)$ at $y_2(i)$ and $0$ at $0$. Since $\sigma$ is strictly monotonic, continuous and $0 \leq |x_1(i)| \leq |x_2(i)|$ and $x_1(i)$ and $x_2(i)$ have the same sign, $\sigma$ attains $x_1(i)$ at a point between $0$ and $y_2(i)$. Hence, $|y_1(i)| \leq |y_2(i)|$ as required.

*Claim 3*: The function $f$ that assigns to each point on the ray that has an inverse image the length of that inverse image is strictly increasing in the direction away from the origin.

Take any two points on the ray $x_1$ and $x_2$ that have inverse images $y_1$ and $y_2$ where $x_1$ is closer to the origin. By the previous claim, for $1 \leq i \leq d$, we have $|y_1(i)| \leq |y_2(i)|$ and therefore $||y_1||_2 \leq ||y_2||_2$ and therefore $f(x_1) \leq f(x_2)$. Assume $f(x_1) = f(x_2)$. Then we must have $|y_1(i)| = |y_2(i)|$ for $1 \leq i \leq d$. Since $\sigma$ is strictly monotonic and $\sigma(0) = 0$, $\sigma$ either preserves the sign of all inputs or reverses the sign of all inputs. Since $x_1(i)$ and $x_2(i)$ have the same sign, so do $y_1(i)$ and $y_2(i)$. So $y_1(i) = y_2(i)$, so $y_1 = y_2$, so the forward images of $y_1$ and $y_2$ are the same, so $x_1 = x_2$. Contradiction. So $f(x_1) \neq f(x_2)$, so $f(x_1) < f(x_2)$.

*Claim 4*: The function $f$ that assigns to each point on the ray that has an inverse image the length of that inverse image is continuous.

Since $f$ is only defined on a 1-dimensional space, it is enough to show left-continuity and right-continuity.

Part 1: left-continuity. Take a sequence of points on the ray with inverse images that approach some point $x_{\lim}$ on the ray from the left and assume that $x_{\lim}$ also has an inverse image $y_{\lim}$. Then we need to show that the length of $y_{\lim}$ is the limit of the lengths of the inverse images of the sequence. It is enough to show this for the monotonic re-ordering of the sequence. Let that monotonic re-ordering be $x_n$. Then we have $x_n \to x_{\lim}$ and $||x_n||$ increases. By claim 2, for $1 \leq i \leq d$, $|y_n(i)|$ is an increasing sequence. This means that $|y_n(i)|$ either converges or it is an unbounded sequence. If the latter is true, then it will exceed $|y_{\lim}(i)|$. But since $x_{\lim}$ is at least as large as all $x_n$, again by claim 2 we must have $|y_{\lim}(i)| \geq |y_n(i)|$. Contradiction. So $|y_n(i)|$ converges. Since $\sigma$ is strictly monotonic and $\sigma(0)$, $\sigma$ either preserves the sign of all values or it reverses the sign of all values. Since for each $1 \leq i \leq d$, the $x_n(i)$ all have the same sign because the $x_n$ are on a ray, the $y_n(i)$ all have the same sign and so since $|y_n(i)|$ converges, $y_n(i)$ converges. Let its limit be $y'_{\lim}(i)$. Because $\sigma$ is continuous, its value at $y'_{\lim}(i)$ is the limit of $\sigma(y_n(i))$. But that is $x_{\lim}(i)$. So if $y'_{\lim}$ is the vector made up of the $y'_{\lim}(i)$, it is the inverse image of $x_{\lim}$. i.e. $y'_{\lim} = y_{\lim}$. Since $||.||_2$ is also a continuous function, $||y_n||_2 \to ||y'_{\lim}||_2$ and so $||y_n||_2 \to ||y_{\lim}||_2$ and so $f(x_n) \to f(x)$ as required.

Part 2: right-continuity. This case is analogous. We have a decreasing sequence $x_n$ and so decreasing sequences $|y_n(i)|$ and so convergent sequences $y_n(i)$ with a limit $y'_{\lim}$ that is equal to $y_{\lim}$ and so $f(x_n) \to f(x)$ as required.

*Claim 5*: The co-domain of the function $f$ that assigns to each point on the ray that has an inverse image the length of that inverse image is the positive reals.

We argue by contradiction. Assume the co-domain of $f$ is not the positive reals. Then the set $S$ of positive reals not attained by $f$ is non-empty. Let $s$ be the infimum of $S$.

Case 1: $s = 0$. Since $\sigma$ is strictly monotonic with $\sigma(0) = 0$, there exists an $\epsilon > 0$ such that $\sigma$ attains all values in the interval $[-\epsilon, \epsilon]$. So all points on the ray for which all components have absolute value less than $\epsilon$ have an inverse image, so there exists an $\epsilon' > 0$ such that all points on the ray with length in the interval $(0, \epsilon']$ have an inverse image, so $f$ is defined there. Further, if we extend the domain of $f$ to include the origin, we have $f(0) = 0$. But by the same argument used for claim 4, $f$

is then right-continuous at 0. Since $f$ is also continuous and defined on $(0, \epsilon']$, it attains all values in the interval $[0, f(\epsilon')]$. Specifically, if $f$ is restricted to its original domain (the ray), it still attains all values in $(0, f(\epsilon')]$. So the infimum of $S$ is at least $f(\epsilon')$. Contradiction.

Case 2: $s > 0$ and $s \in S$. Then there exists a sequence $x_n$ of points on the ray such that $f(x_n) \to s$ and $f(x_n)$ is strictly increasing. By claim 3, $||x_n||_2$ is strictly increasing. Let the inverse images of the $x_n$ be $y_n$. By claim 2, $|y_n(i)|$ is increasing for $1 \le i \le d$. Since $|y_n(i)| \le ||y_n||_2 < s$, $|y_n(i)|$ is bounded from above, so it converges. As $\sigma$ is strictly monotonic and $\sigma(0) = 0$, $\sigma$ either preserves the sign or reverses it for all inputs. So since the $x_n(i)$ all have the same sign, so do the $y_n(i)$. So $y_n(i)$ converges. Let this limit be $y_{\lim}(i)$. Since for $1 \le i \le d$, $y_n(i) \to y_{\lim}(i)$, we have $y_n \to y_{\lim}$ where $y_{\lim}$ is the vector composed of the $y_{\lim}(i)$. Since $\sigma$ is continuous, the forward image of $y_{\lim}$ is the limit of forward images of the $y_n$. Since the forward images of the $y_n$ lie on the ray, so does their limit. Hence, the forward image of $y_{\lim}$ lies on the ray. Call it $x_{\lim}$. Since length is also continuous, $s = \lim_{n \to \inf} ||y_n||_2 = ||y_{\lim}||_2$. So $f(x_{\lim}) = s$ so $s \notin S$. Contradiction.

Case 3: $s > 0$ and $s \notin S$. Then there is a point $x$ on the ray for which $f(x) = s$. Let its inverse image be $y$. Let $I$ be the set of indeces $i$ with $1 \le i \le d$ and $x(i) \ne 0$. Since $\sigma$ has $\sigma(0) = 0$ and it is strictly monotonic, $y(i) \ne 0$. For $i \in I$, let $\sigma_{\max}(i) = \sigma(2y(i))$. Since $\sigma$ has $\sigma(0) = 0$ and it is strictly monotonic, we have $|\sigma(2y(i))| > |\sigma(y(i))| > 0$ and so $|\sigma_{\max}(i)| > |x(i)| > 0$ and also $\sigma_{\max}(i)$ and $x(i)$ have the same sign. Let $C = \min_{i \in I} \frac{\sigma_{\max}(i)}{x(i)}$. Take some vector $y'$ that can vary. Since $\sigma$ is continuous, as $y'(i)$ varies between $y(i)$ and $2y(i)$, it attains all values between $x(i)$ and $\sigma_{\max}(i)$. So for all $1 \le c \le C$, we can set $y'(i)$ to some value $y_c(i)$ such that $\sigma(y_c(i)) = cx(i)$. So the vector $y_c$ that has the aforementioned $y_c(i)$ components for $i \in I$ and has zero components for $i \notin I$ is the inverse image of $cx$. So $f$ is defined on $cx$ for $1 \le c \le C$. Let $s' := f(Cx)$. By claim 3, $f$ is strictly increasing so $s' > s$. By claim 4, $f$ is continuous. So between $x$ and $Cx$, $f$ takes all values between and including $s$ and $s'$. Also, by the original definition of $s$, $f$ attains all positive real values less than $s$. So $f$ attains all positive real values less than $s'$. But $s$ was defined to be the infimum of positive real values that $f$ does not attain. Contradiction.

*Claim 6*: The inverse image of the ray intersects the hypersphere in a single point.

By claim 5, there is a point on the ray that has an inverse image of length $r$. By claim 3, there is exactly one such point. Therefore, the inverse image of the ray contains exactly one point of length $r$, so it intersects the hypersphere in exactly one point, as required.

$\square$

The proposition also applies if each neuron in the nonlinearity layer uses a different nonlinearity $\sigma_i$ as long as it fulfills the stated conditions.

### E.7 PROPOSITION 7

We say a random function $f_b$ is '$k$-diluted in expectation' with respect to random vector $v$ if there exists a random matrix $S_b$ and a random function $\rho_b$ such that $f_b(v) = S_b v + \rho_b(v)$ and $\frac{\mathbb{Q}||S_b v||_2}{\mathbb{Q}||\rho_b(v)||_2} = k$.

We say a random function $\rho(v)$ is 'scale-symmetric decomposable' (SSD) if it can be written as $u_\rho \ell_\rho(\frac{v}{||v||_2})||v||_2$, where $\ell_\rho$ is a random scalar function and $u_\rho$ is uniformly distributed on the hypersphere and independent of both $\ell_\rho$ and $v$.

We say a random matrix $S$ is 'scale-symmetric decomposable' (SSD) if $Sv$, when viewed as a function of the vector $v$ is SSD.

**Proposition 7.** *Let $u$ be a uniformly distributed unit length vector. Given random functions $f_b$, $1 \le b \le B$, which are $k_b$-diluted in expectation with respect to $u$, a matrix $S_b$ that is either SSD or a multiple of the identity and an SSD random function $\rho_b := f_b - S_b$ where all the $S_b$ and $\rho_b$ are independent, $f_1 \circ f_2 \circ .. \circ f_B$ is $\left( \left( \prod_b (1 + \frac{1}{k_b^2}) \right) - 1 \right)^{-\frac{1}{2}}$-diluted in expectation with respect to $u$.*

*Proof.* Let $U$ be the uniform distribution over unit length vectors and $u \sim U$. We will procede by induction over $B$, where the induction hypothesis includes the following claims.

1. $f_1 \circ f_2 \circ .. \circ f_B$ is $\left( \left( \prod_b (1 + \frac{1}{k_b^2}) \right) - 1 \right)^{-\frac{1}{2}}$-diluted in expectation.

2. $\mathbb{Q}||f_1 \circ f_2 \circ .. \circ f_B(u)||_2 = \sqrt{(\mathbb{Q}||S_1 S_2 .. S_B u||_2)^2 + (\mathbb{Q}||f_1 \circ f_2 \circ .. \circ f_B(u) - S_1 S_2 .. S_B u||_2)^2}$

3. $S_1 S_2 .. S_B u$, $f_1 \circ f_2 \circ .. \circ f_B(u)$ and $f_1 \circ f_2 \circ .. \circ f_B(u) - S_1 S_2 .. S_B u$ are all radially symmetric

Let's start by looking at the case $B = 1$. Claim (1) follows directly from the conditions of the proposition. We have:

$$
\begin{aligned}
& \mathbb{Q}||f_1(u)||_2 \\
=\ & \mathbb{Q}||S_1 u + \rho_1(u)||_2 \\
=\ & \mathbb{Q}\sqrt{(S_1 u + \rho_1(u)).(S_1 u + \rho_1(u))} \\
=\ & \sqrt{\mathbb{E}(S_1 u + \rho_1(u)).(S_1 u + \rho_1(u))} \\
=\ & \sqrt{\mathbb{E}(S_1 u).(S_1 u) + 2\mathbb{E}(S_1 u).\rho_1(u) + \mathbb{E}\rho_1(u).\rho_1(u)} \\
=\ & \sqrt{\mathbb{E}... + 2\mathbb{E}(S_1 u).(u_{\rho_1}\ell_{\rho_1}(\frac{u}{||u||_2})||u||_2) + \mathbb{E}...} \\
=\ & \sqrt{\mathbb{E}... + 2\mathbb{E}_{u,S_1,\ell_{\rho_1}}(S_1 u).(\ell_{\rho_1}(\frac{u}{||u||_2})||u||_2 \mathbb{E}_{u_{\rho_1}} u_{\rho_1}) + \mathbb{E}...} \\
=\ & \sqrt{\mathbb{E}(S_1 u).(S_1 u) + 0 + \mathbb{E}\rho_1(u).\rho_1(u)} \\
=\ & \sqrt{(\mathbb{Q}||S_1 u||_2)^2 + (\mathbb{Q}||\rho_1(u)||_2)^2} \\
=\ & \sqrt{(\mathbb{Q}||S_1 u||_2)^2 + (\mathbb{Q}||f_1(u) - S_1 u||_2)^2}
\end{aligned}
$$

This is claim (2). For any $u$, $\rho_1(u)$ is radially symmetric because $\rho_1$ is SSD. If $S_1$ is SSD, $S_1 u$ is also radially symmetric for arbitrary $u$. If $S_1$ is a multiple of the identity, $S_1 u$ is radially symmetric because $u$ is radially symmetric. In either case, $S_1 u$ is radially symmetric. Because the orientation of $\rho_1(u)$ is governed only by $u_{\rho_1}$ which is independent of both $u$ and $S_1$, the orientations of $S_1 u$ and $\rho_1(u)$ are independent. But the sum of two radially symmetric random variables with independent orientations is itself radially symmetric, so $f_1(u)$ is also radially symmetric. This yields claim (3).

Now for the induction step. Set $B$ to some value and also define $k_\circ := \left( \left( \prod_{b=2}^B (1 + \frac{1}{k_b^2}) \right) - 1 \right)^{-\frac{1}{2}}$, $S_\circ := S_2 S_3 .. S_B$, $f_\circ := f_2 \circ .. \circ f_B$ and $\rho_\circ := f_\circ - S_\circ$. Then the induction hypothesis yields that $f_\circ$ is $k_\circ$-diluted in expectation, that $\mathbb{Q}||f_\circ(u)||_2 = \sqrt{\mathbb{Q}||S_\circ u||_2 + \mathbb{Q}||\rho_\circ(u)||_2}$ and that $f_\circ(u)$, $\rho_\circ(u)$ and $S_\circ u$ are radially symmetric. This implies that $\frac{f_\circ(u)}{||f_\circ(u)||_2}$, $\frac{\rho_\circ(u)}{||\rho_\circ(u)||_2}$ and $\frac{S_\circ u}{||S_\circ u||_2}$ are uniformly distributed unit length vectors. Define $c_\circ := \mathbb{Q}||S_\circ u||_2$. Then $k_\circ$-dilution in expectation implies $\mathbb{Q}||\rho_\circ(u)||_2 = \frac{c_\circ}{k_\circ}$ and hence $\mathbb{Q}||f_\circ(u)||_2 = \sqrt{1 + \frac{1}{k_\circ^2}} c_\circ$. Similarly, let $c_1 := \mathbb{Q}||S_1 u||_2$ and then $\mathbb{Q}||\rho_1(u)||_2 = \frac{c_1}{k_1}$ and analogously to the $B = 1$ case we have $\mathbb{Q}||f_1(u)||_2 = \sqrt{(\mathbb{Q}||S_1 u||_2)^2 + (\mathbb{Q}||f_1(u) - S_1 u||_2)^2} = \sqrt{c_1^2 + (\frac{c_1}{k_1})^2} = \sqrt{1 + \frac{1}{k_1^2}} c_1$. We have

$$
\begin{aligned}
& \mathbb{Q}||\rho_1(f_\circ(u))||_2 \\
=\ & \sqrt{\mathbb{E}\rho_1(f_\circ(u)).\rho_1(f_\circ(u))} \\
=\ & \sqrt{\mathbb{E}(u_{\rho_1}\ell_{\rho_1}(\frac{f_\circ(u)}{||f_\circ(u)||_2})||f_\circ(u)||_2).(u_{\rho_1}\ell_{\rho_1}(\frac{f_\circ(u)}{||f_\circ(u)||_2})||f_\circ(u)||_2)} \\
=\ & \sqrt{\mathbb{E}_{||f_\circ(u)||_2, \frac{f_\circ(u)}{||f_\circ(u)||_2}, \rho_1}||f_\circ(u)||_2^2 (u_{\rho_1}\ell_{\rho_1}(\frac{f_\circ(u)}{||f_\circ(u)||_2})).(u_{\rho_1}\ell_{\rho_1}(\frac{f_\circ(u)}{||f_\circ(u)||_2}))}
\end{aligned}
$$

$$= \sqrt{\mathbb{E}_{||f_\circ(u)||_2}||f_\circ(u)||_2^2 \mathbb{E}_{\frac{f_\circ(u)}{||f_\circ(u)||_2},\rho_1}(u_{\rho_1}\ell_{\rho_1}(\frac{f_\circ(u)}{||f_\circ(u)||_2})).(u_{\rho_1}\ell_{\rho_1}(\frac{f_\circ(u)}{||f_\circ(u)||_2}))}$$

$$= \sqrt{(1+\frac{1}{k_\circ^2})c_\circ^2 \mathbb{E}_{\frac{f_\circ(u)}{||f_\circ(u)||_2},\rho_1}\rho_1(\frac{f_\circ(u)}{||f_\circ(u)||_2}).\rho_1(\frac{f_\circ(u)}{||f_\circ(u)||_2})}$$

$$= \sqrt{(1+\frac{1}{k_\circ^2})c_\circ^2(\mathbb{Q}||\rho_1(\frac{f_\circ(u)}{||f_\circ(u)||_2})||_2)^2}$$

$$= \sqrt{1+\frac{1}{k_\circ^2}}c_\circ\frac{c_1}{k_1}$$

If $S_1$ is SSD, analogously, we have $\mathbb{Q}||S_1 S_\circ u||_2 = c_\circ c_1$, $\mathbb{Q}||S_1 \rho_\circ(u)||_2 = \frac{c_\circ}{k_\circ}c_1$ and $\mathbb{Q}||S_1 f_\circ(u)||_2 = \sqrt{1+\frac{1}{k_\circ^2}}c_\circ c_1$. If $S_1$ is a multiple of the identity, by the definition of $c_1$, it is $c_1$ times the identity. Therefore $\mathbb{Q}||S_1 S_\circ u||_2 = \mathbb{Q}||c_1 S_\circ u||_2 = c_\circ c_1$, $\mathbb{Q}||S_1 \rho_\circ(u)||_2 = \mathbb{Q}||c_1 \rho_\circ(u)||_2 = \frac{c_\circ}{k_\circ}c_1$ and $\mathbb{Q}||S_1 f_\circ(u)||_2 = \mathbb{Q}||c_1 f_\circ(u)||_2 = \sqrt{1+\frac{1}{k_\circ^2}}c_\circ c_1$ also hold. Then we have

$$\mathbb{Q}||f_1(f_\circ(u)) - S_1 S_\circ u||_2$$
$$\mathbb{Q}||S_1(f_\circ(u)) + \rho_1(f_\circ(u)) - S_1 S_\circ u||_2$$
$$\mathbb{Q}||S_1(S_\circ u + \rho_\circ(u)) + \rho_1(f_\circ(u)) - S_1 S_\circ u||_2$$
$$\mathbb{Q}||S_1 \rho_\circ(u) + \rho_1(f_\circ(u))||_2$$
$$= \sqrt{\mathbb{E}(S_1\rho_\circ(u)).(S_1\rho_\circ(u)) + \mathbb{E}(S_1\rho_\circ(u)).\rho_1(f_\circ(u)) + \mathbb{E}\rho_1(f_\circ(u)).\rho_1(f_\circ(u))}$$
$$= \sqrt{\mathbb{E}... + 2\mathbb{E}(S_1\rho_\circ(u)).(u_{\rho_1}\ell_{\rho_1}(\frac{f_\circ(u)}{||f_\circ(u)||_2})||f_\circ(u)||_2) + \mathbb{E}...}$$
$$= \sqrt{\mathbb{E}... + 2\mathbb{E}_{f_\circ(u),\rho_\circ(u),S_1,\ell_{\rho_1}}(S_1\rho_\circ(u)).(\ell_{\rho_1}(\frac{f_\circ(u)}{||f_\circ(u)||_2})||f_\circ(u)||_2 \mathbb{E}_{u_{\rho_1}}u_{\rho_1}) + \mathbb{E}...}$$
$$= \sqrt{\mathbb{E}(S_1\rho_\circ(u)).(S_1\rho_\circ(u)) + 0 + \mathbb{E}\rho_1(f_\circ(u)).\rho_1(f_\circ(u))}$$
$$= \sqrt{(\mathbb{Q}||S_1\rho_\circ(u)||_2)^2 + (\mathbb{Q}||\rho_1(f_\circ(u))||_2)^2}$$
$$= \sqrt{\left(\frac{c_\circ}{k_\circ}c_1\right)^2 + \left(\sqrt{1+\frac{1}{k_\circ^2}}c_\circ\frac{c_1}{k_1}\right)^2}$$
$$= c_\circ c_1 \sqrt{\frac{1}{k_\circ^2} + (1+\frac{1}{k_\circ^2})\frac{1}{k_1^2}}$$
$$= c_\circ c_1 \sqrt{-1 + (1+\frac{1}{k_\circ^2})(1+\frac{1}{k_1^2})}$$

And analogously we have

$$\mathbb{Q}||f_1(f_\circ(u))||_2$$
$$\mathbb{Q}||S_1(f_\circ(u)) + \rho_1(f_\circ(u))||_2$$
$$= \sqrt{(\mathbb{Q}||S_1 f_\circ(u)||_2)^2 + (\mathbb{Q}||\rho_1(f_\circ(u))||_2)^2}$$
$$= \sqrt{\left(\sqrt{1+\frac{1}{k_\circ^2}}c_\circ c_1\right)^2 + \left(\sqrt{1+\frac{1}{k_\circ^2}}c_\circ\frac{c_1}{k_1}\right)^2}$$

$$= c_\circ c_1 \sqrt{(1 + \frac{1}{k_\circ^2})(1 + \frac{1}{k_1^2})}$$

So $\frac{\mathbb{Q}||S_1 S_\circ u||_2}{\mathbb{Q}||f_1(f_\circ(u)) - S_1 S_\circ u||_2} = \frac{c_\circ c_1}{c_\circ c_1 \sqrt{-1 + (1 + \frac{1}{k_\circ^2})(1 + \frac{1}{k_1^2})}} = (-1 + (1 + \frac{1}{k_\circ^2})(1 + \frac{1}{k_1^2}))^{-\frac{1}{2}}$. Substituting

back $k_\circ$, $S_\circ$ and $f_\circ$, we obtain claim (1). Claim (2), when substituting in $k_\circ$, $S_\circ$ and $f_\circ$ becomes $\mathbb{Q}||f_1(f_\circ(u))||_2 = \sqrt{(\mathbb{Q}||S_1 S_\circ u||_2)^2 + (\mathbb{Q}||f_1(f_\circ(u)) - S_1 S_\circ u||_2)^2}$. Substituting the identities we obtained results in $c_\circ c_1 \sqrt{(1 + \frac{1}{k_\circ^2})(1 + \frac{1}{k_1^2})} = \sqrt{(c_\circ c_1)^2 + \left(c_\circ c_1 \sqrt{-1 + (1 + \frac{1}{k_\circ^2})(1 + \frac{1}{k_1^2})}\right)^2}$, which is true, so we have claim (2).

Consider $S_1 S_2 .. S_B u = S_1 S_\circ u$. We know $S_\circ u$ is radially symmetric by the induction hypothesis, so if $S_1$ is a multiple of the identity, so is $S_1 S_\circ u$. If $S_1$ is SSD, then $S_1 S_\circ u$ is radially symmetric for any value of $S_\circ u$. In either case, $S_1 S_\circ u$ is radially symmetric.

Consider $f_1 f_2 .. f_B u = f_1 f_\circ u$. We know $f_\circ u$ is radially symmetric by the induction hypothesis. We also have $f_1 f_\circ u = S_1 f_\circ u + \rho_1(f_\circ u)$. We just showed $S_1 f_\circ u$ is radially symmetric. Because $\rho_1$ is SSD, $\rho_1(f_\circ u)$ is radially symmetric with an orientation independent of that of $S_1 f_\circ u$ because it is governed only by $u_{\rho_1}$. The sum of two radially symmetric random variables with independent orientation is itself radially symmetric, so $f_1 f_\circ u$ is radially symmetric.

Finally, consider $f_1 f_2 .. f_B u - S_1 S_2 .. S_B u = S_1 \rho_\circ(u) + \rho_1(f_\circ(u))$. We know $\rho_\circ(u)$ is radially symmetric by the induction hypothesis so as before, $S_1 \rho_\circ(u)$ is radially symmetric. And again, $\rho_1(f_\circ(u))$ is radially symmetric with independent orientation, so the sum $f_1 f_2 .. f_B u - S_1 S_2 .. S_B u$ is radially symmetric.

So we also have claim (3). This completes the proof.

$\square$

A Gaussian initialized matrix and an orthogonally initialized matrix are both SSD. Therefore the condition that the $S_b$ are either the identity or SSD is fulfilled for all popular skip connection types.

Unfortunately, many ResNets do not quite fulfill the SSD condition on $\rho_b$, but they come close. If the last operation of each $\rho_b$ is multiplication with an SSD matrix, a popular choice, the orientation of $\rho_b$ is indeed governed by an independent, uniform unit length vector $u_\rho$ as required. However, many choices of $\rho_b$ do not preserve the length of the incoming vector $||v||_2$ as in an SSD function. However, because the inputs to each $\rho_b$ are random and high-dimensional, we do not expect their lengths to vary much, especially if the original inputs to the network have themselves been normalized. So the loss of the information of the length of the incoming vector should not cause the overall behavior to change significantly.

A block function $\rho_b$ that is SSD, for example, is any such function that is composed of only linear and ReLU layers, where the final layer is Gaussian or orthogonally initialized.

Finally, note that this proposition applies only in expectation over randomly initialized matrices. As long as those matrices are high-dimensional, we expect it to apply approximately to specific realizations of those matrices as well.

# F  THEOREMS AND PROOFS

## F.1  THEOREM 1 - EXPLODING GRADIENTS LIMIT DEPTH

See section D for the formal definition of effective depth and related concepts.

Consider some MLP $f$ with nominal depth $L$ and layers $f_l$, $1 \le l \le L$. Let its compositional depth be $\mathcal{N}$ and its linear layers be $f_{l_n}$, $1 \le n \le \mathcal{N}$ where $l_1 < l_2 < .. < l_\mathcal{N}$. Let each linear layer be the sum of an unparametrized initial function $i_{l_n}$ and a parametrized residual function $r_{l_n}(\theta_{l_n})$. $i_{l_n}$ represents multiplication with the initial weight matrix and is used interchangeably to denote that initial weight matrix. $r_{l_n}(\theta_{l_n})$ represents multiplication with the residual weight matrix and is used

interchangeably to denote that residual weight matrix. The parameter sub-vector $\theta_{l_n}$ contains the entries of the residual weight matrix.

Let an $N$-trace $\phi_N$ be a subset of $\{1, .., N\}$. Let $\Phi_N$ be the set of all possible $N$-traces and let $\Phi_N^\lambda$ be the set of all $N$-traces of size $\lambda$ or more. We define the 'gradient term' $G(\phi_N, f, \theta, X, Y) := J_0 J_1..J_{l_{N+1}-1}$ where $J_k = \mathcal{J}_{k+1}^k$ if layer $k$ is not a linear layer, $J_k = r_{l_n}(\theta_{l_n})$ if layer $k$ corresponds to linear layer $l_n$ and $n \in \phi_N$, and $J_k = i_{l_n}$ if layer $k$ corresponds to linear layer $l_n$ and $n \notin \phi_N$.

Let $res_N^\Lambda(f, \theta, X, Y) := \sum_{\lambda=\Lambda}^{N-1} \sum_{\phi_{N-1} \in \Phi_{N-1}^\lambda} G(\phi_{N-1}, f, \theta, X, Y)$.

**Theorem 1.** *Consider an MLP $f$ as defined above with all parameter sub-vectors initialized to $\theta_{l_n}^{(0)} = 0$. Taken some set of possible query points $\mathcal{D}$. Let each of the parameter sub-vectors be updated with a sequence of updates $\Delta\theta_{l_n}^{(t)}$ such that $\theta_{l_n}^{(t)} = \theta_{l_n}^{(t-1)} + \Delta\theta_{l_n}^{(t)}$ with $1 \leq t \leq T$. Let $alg$ be a fixed function and $\Lambda$ an integer. Further assume:*

1. $\Delta\theta_{l_n}^{(t)} = alg\left(\frac{df(\theta^{(t-1)}, X^{(t)}, Y^{(t)})}{d\theta_{l_n}^{(t-1)}}\right)$ *for some* $(X^{(t)}, Y^{(t)}) \in \mathcal{D}$

2. *there exist constants $r$ and $c$ such that* $\frac{||\Delta\theta_{l_n}^{(t)}||_2}{||i_{l_n}||_2} \leq \frac{1}{cr^n}$ *for all $n$ and $t$*

3. *there exists a constant $c' \geq 1$ such that*

$$\frac{||alg\left(\frac{df(\theta^{(t-1)}, X^{(t)}, Y^{(t)})}{d\theta_{l_N}^{(t-1)}}\right) - alg\left(\frac{df(\theta^{(t-1)}, X^{(t)}, Y^{(t)})}{d\theta_{l_N}^{(t-1)}} - res_N^\Lambda(f, \theta^{(t-1)}, X^{(t)}, Y^{(t)})\right)||_2}{||alg\left(\frac{df(\theta^{(t-1)}, X^{(t)}, Y^{(t)})}{d\theta_{l_N}^{(t-1)}}\right)||_2}$$

$$\leq c' \sum_{\lambda=\Lambda}^{N-1} \sum_{\phi_{N-1} \in \Phi_{N-1}^\lambda} \prod_{n \in \phi_{N-1}} \frac{||r_{l_n}(\theta_{l_n}^{(t)})||_2}{||i_{l_n}||_2}$$

*for all $N$ and $t$.*

4. $T \leq \frac{ch^{\frac{1}{\Lambda}}}{32rc'}(\ln r)^3 r^{\frac{\Lambda}{4}}$ *for some $h \leq 1$*

*Then for all $N$ we have*

$$\frac{1}{||i_{l_N}||_2}||\sum_{t=1}^{T}\left(alg\left(\frac{df(\theta^{(t-1)}, X^{(t)}, Y^{(t)})}{d\theta_{l_N}^{(t-1)}}\right) - alg\left(\frac{df(\theta^{(t-1)}, X^{(t)}, Y^{(t)})}{d\theta_{l_N}^{(t-1)}} - res^\Lambda(f, \theta^{(t-1)}, X^{(t)}, Y^{(t)})\right)\right)||_2 \leq h$$

*Proof.* For all $T' \leq T$, we have $||\theta_{l_n}^{(T')}||_2 = ||\sum_{t=1}^{T'} \Delta\theta_{l_n}^{(t)}||_2 \leq \sum_{t=1}^{T'} ||\Delta\theta_{l_n}^{(t)}||_2 \leq \frac{T'}{cr^n}||i_{l_n}||_2 \leq \frac{T}{cr^n}||i_{l_n}||_2$. Then we have

$$\frac{1}{||i_{l_N}||_2}||\sum_{t=1}^{T}\left(alg\left(\frac{df(\theta^{(t-1)}, X^{(t)}, Y^{(t)})}{d\theta_{l_N}^{(t-1)}}\right) - alg\left(\frac{df(\theta^{(t-1)}, X^{(t)}, Y^{(t)})}{d\theta_{l_N}^{(t-1)}} - res^\Lambda(f, \theta^{(t-1)}, X^{(t)}, Y^{(t)})\right)\right)||_2$$

$$\leq \frac{1}{||i_{l_N}||_2}\sum_{t=1}^{T}||alg\left(\frac{df(\theta^{(t-1)}, X^{(t)}, Y^{(t)})}{d\theta_{l_N}^{(t-1)}}\right) - alg\left(\frac{df(\theta^{(t-1)}, X^{(t)}, Y^{(t)})}{d\theta_{l_N}^{(t-1)}} - res^\Lambda(f, \theta^{(t-1)}, X^{(t)}, Y^{(t)})\right)||_2$$

$$\leq \frac{1}{||i_{l_N}||_2}\sum_{t=1}^{T}||alg\left(\frac{df(\theta^{(t-1)}, X^{(t)}, Y^{(t)})}{d\theta_{l_N}^{(t-1)}}\right)||_2 c' \sum_{\lambda=\Lambda}^{N-1} \sum_{\phi_{N-1} \in \Phi_{N-1}^\lambda} \prod_{n \in \phi_{N-1}} \frac{||r_{l_n}^{(t)}(\theta_{l_n})||_2}{||i_{l_n}||_2}$$

$$= c' \sum_{t=1}^{T} \frac{||\Delta\theta_{l_N}^{(t)}||_2}{||i_{l_N}||_2} \sum_{\lambda=\Lambda}^{N-1} \sum_{\phi_{N-1} \in \Phi_{N-1}^\lambda} \prod_{n \in \phi_{N-1}} \frac{||r_{l_n}(\theta_{l_n}^{(t)})||_2}{||i_{l_n}||_2}$$

$$\leq \quad \frac{c'T}{cr^N} \sum_{\lambda=\Lambda}^{N-1} \sum_{\phi_{N-1}\in\Phi_{N-1}^\lambda} \prod_{n\in\phi_{N-1}} \frac{||\theta_{l_n}^{(t)}||_2}{||i_{l_n}||_2}$$

$$\leq \quad \frac{Tc'}{cr^N} \sum_{\lambda=\Lambda}^{N} \sum_{\phi_{N-1}\in\Phi_{N-1}^\lambda} \prod_{n\in\phi_{N-1}} \frac{T}{cr^n}$$

$$\leq \quad \frac{Tc'}{cr^N} \sum_{\lambda=\Lambda}^{\infty} (\frac{T}{c})^\lambda \sum_{\phi_\infty\in\Phi_\infty^\lambda} \prod_{n\in\phi_\infty} \frac{1}{r^n}$$

Let $K(\lambda, n)$ be the number of ways to choose $\lambda$ distinct positive integers such that their sum is $n$. Clearly, $K(\lambda, n) = 0$ for $n < \frac{\lambda(\lambda+1)}{2}$. For $n \geq \frac{\lambda(\lambda+1)}{2}$, the largest number that can be chosen is $n - \frac{\lambda(\lambda+1)}{2} + \lambda$ and so $K(\lambda, n) \leq (n - \frac{\lambda(\lambda+1)}{2} + \lambda)^\lambda = (n - \frac{\lambda(\lambda-1)}{2})^\lambda$. So we have

$$\frac{Tc'}{cr^N} \sum_{\lambda=\Lambda}^{\infty} (\frac{T}{c})^\lambda \sum_{\phi_\infty\in\Phi_\infty^\lambda} \prod_{n\in\phi_\infty} \frac{1}{r^n}$$

$$= \quad \frac{Tc'}{cr^N} \sum_{\lambda=\Lambda}^{\infty} (\frac{T}{c})^\lambda \sum_{n} \frac{K(\lambda, n)}{r^n}$$

$$= \quad \frac{Tc'}{cr^N} \sum_{\lambda=\Lambda}^{\infty} (\frac{T}{c})^\lambda \sum_{n=\frac{\lambda(\lambda-1)}{2}+1}^{\infty} \frac{K(\lambda, n)}{r^n}$$

$$\leq \quad \frac{Tc'}{cr^N} \sum_{\lambda=\Lambda}^{\infty} (\frac{T}{c})^\lambda \sum_{n=\frac{\lambda(\lambda-1)}{2}+1}^{\infty} \frac{(n - \frac{\lambda(\lambda-1)}{2})^\lambda}{r^n}$$

$$= \quad \frac{Tc'}{cr^N} \sum_{\lambda=\Lambda}^{\infty} (\frac{T}{c})^\lambda r^{-\frac{\lambda(\lambda-1)}{2}} \sum_{n=1}^{\infty} \frac{n^\lambda}{r^n}$$

$$< \quad \frac{Tc'}{cr^N} \sum_{\lambda=\Lambda}^{\infty} (\frac{T}{c})^\lambda r^{-\frac{\lambda(\lambda-1)}{2}} \lambda^\lambda (\ln r)^{-\lambda}$$

$$\leq \quad c' \sum_{\lambda=\Lambda}^{\infty} (\frac{T}{c})^{\lambda+1} r^{-\frac{\lambda(\lambda-1)}{2}} \lambda^\lambda (\ln r)^{-\lambda}$$

$$\leq \quad c' \sum_{\lambda=\Lambda}^{\infty} (\frac{ch^{\frac{1}{\Lambda}}}{32rc'}(\ln r)^3 r^{\frac{\Lambda}{4}})^{\lambda+1} r^{-\frac{\lambda(\lambda-1)}{2}} \lambda^\lambda (\ln r)^{-\lambda}$$

$$< \quad \sum_{\lambda=\Lambda}^{\infty} h^{\frac{\lambda+1}{\Lambda}} (\frac{1}{2})^{\lambda+1} (\frac{1}{16})^{\lambda+1} (\ln r)^{2\lambda+3} r^{-\frac{1}{4}\lambda^2} \lambda^\lambda$$

$$< \quad \sum_{\lambda=\Lambda}^{\infty} h^{\frac{\lambda+1}{\Lambda}} (\frac{1}{2})^{\lambda+1}$$

$$< \quad h$$

$\square$

$alg$ represents the gradient-based algorithm and the quantity that is ultimately bounded by $h$ is the first-order approximation of the relative $\lambda$-contribution at layer $l_N$ until time $T$. To obtain that the network has effective depth $\Lambda$, all we need is to set $h$ to a small value. In that case, the first-order approximation is sufficient.

Now, we analyze the four conditions in turn.

Condition (1) states that the algorithm computes the update. For convenience, we write the algorithm as a deterministic function of the gradient of the layer for which the update is computed. The proof can be trivially extended to algorithms that use the gradients of other layers, past gradients and as well randomness if we add the same dependencies to condition (3). Also for convenience, we assume a batch size of 1. We can apply the result to larger batch sizes, for example, by having $alg$ use past gradients and setting the majority of updates to zero.

Condition (2) reflects the argument from section 4.3 that the area around the current parameter value in which the gradient is reflective of the function is bounded by a hypersphere of relative radius $\frac{1}{GSC(l_n,0)}$, and the assumption that gradients explode, i.e. $GSC(l_n, 0) \geq cr^{l_n}$. Note that for convenience, we divide the size of the update $||\Delta\theta_{l_n}^{(t)}||_2$ by the weight matrix in the initialized state $||i_{l_n}||_2$ instead of $||\theta_{l_n}^{(t-1)}||_2$. This is realistic given the general observation that the largest useful update size decreases in practice when training a deep network. Therefore, we can bound all updates by the largest useful update size in the initialized state.

The strongest condition is (3). It can be understood as making two distinct assertions.

Firstly, ignoring the $alg()$ function, it bounds the length of the sum of the $\Lambda$-residual terms. In essence, it requires that on average, the size of these terms is "what one would expect" given the $L_2$ norm of the initial and residual weight matrices up to some constant $c'$. In other words, we assume that in $\lambda$-residual terms, the large singular values of layer-wise Jacobians do not compound disproportionately compared to the full gradient. The bound is however also very conservative in the sense that it implicitly assumes that all $\lambda$-residual terms have the same orientation.

Secondly, it asserts that $alg()$ is "relatively Lipschitz" over the gradient. This is fulfilled e.g. for SGD and SGD with custom layer-wise step sizes as used in our experiments. It is fulfilled by SGD with momentum as long as size of the momentum term is bounded below. In theory, it is not fulfilled by RMSprop or Adam as gradients on individual weight matrix entries can be "scaled up" arbitrarily via the denominator. In practice, the $\epsilon$ regularization term used in the denominator prevents this, although this is rarely necessary.

Finally, condition (4) states that the training time is limited. Importantly, the bound on $T$ is exponential in $\Lambda$ and independent of both $L$ and $\mathcal{N}$. Note that we did not attempt to make the bound tight. As it stands, unfortunately, the bound too loose to have much practical value. It would indicate that networks can be trained to far greater depth than is possible in practice. The limitation of effective depth in practice is studied in section 4.4.

## F.2 THEOREM 2 - WHY GRADIENTS EXPLODE

**Theorem 2.** *Consider a neural network $f$ with random parameter $\theta$ composed of layer functions $f_l$ that are surjective endomorphisms on the $d$-dimensional hypersphere. Let $\mathcal{J}_l := \mathcal{J}_{l+1}^l$. Let $S$ be the hypersphere and let $\mathcal{S}$ be the uniform distribution on it. Let $F_l$, $1 \leq l \leq L$, be random vectors independent of $\theta$ where $F_l \sim \mathcal{S}$. Assume:*

1. *The $\theta_l$ are independent of each other.*

2. *Each Jacobian $\mathcal{J}_l(\theta_l, F_{l+1})$ has $d-1$ nonzero singular values which, as $F_{l+1}$ and $\theta_l$ vary, are independent and drawn from some distribution $P_l$.*

3. *There exist some $\epsilon > 0$ and $\delta > 0$ such that $\mathbb{P}_{p_l \sim P_l}(||p_l| - \mathbb{E}_{q_l \sim P_l}(|q_l|)| > \epsilon) > \delta$ for all l.*

4. *$f_l(\theta_l, F_{l+1})$ is a uniformly distributed vector on the hypersphere.*

5. *For any unit length vector $u$, $\mathcal{J}_l(\theta_l, F_{l+1})u$ can be written as $\ell_l(\theta_l, F_{l+1}, u)u_l$. Here, $\ell_l$ is random scalar independent of both $u_l$ and $f_l(\theta_l, F_{l+1})$. $u_l$, conditioned on $f_l(\theta_l, F_{l+1})$, is uniformly distributed in the space of unit length vectors orthogonal to $f_l(\theta_l, F_{l+1})$.*

*Let $X \sim \mathcal{S}$ be a random vector independent of $\theta$. Then setting $r(\delta, \epsilon) := \sqrt{1 + \delta\epsilon^2}$ we have $\mathbb{Q}_{\theta,X}GSC(k, l, f, \theta, X) \geq \sqrt{\frac{d-1}{d}}r(\delta, \epsilon)^{k-l}$.*

*Proof.* The hypersphere is a $d-1$-dimensional subspace in $\mathbb{R}^d$. Hence, any endomorphism $f_l$ on that subspace will have Jacobians with at least one zero singular value with right eigenvector equal to the normal of the subspace at the input and left eigenvector equal to the normal of the subspace at the output. Since the subspace is the unit hypersphere, the normal at the input is the input and the normal at the output is the output. By assumption, the Jacobian has no other zero singular values. Let the singular values of the Jacobian be $s_1, .., s_{d-1}, s_d$. WLOG we set $s_d = 0$.

Throughout this proof, we use $\det(\mathcal{J}_l)$ to denote the product of singular values of the Jacobian excluding the zero singular value, i.e. $\det(\mathcal{J}_l) = \prod_{i=1}^{d-1} s_i$. We use $||\mathcal{J}_l||_{qm}$ to denote the quadratic mean of the singular values excluding the zero singular value, i.e. $||\mathcal{J}_l||_{qm} = \sqrt{\frac{\sum_{i=1}^{d-1} s_i^2}{d-1}}$.

As $f_l$ is surjective, for fixed $\theta_l$, we have by integration by substitution $\int_S 1 \leq \int_S |\det(\mathcal{J}_l(\theta_l, F_{l+1}))| dF_{l+1}$. But we also have $\frac{\int_S |\det(\mathcal{J}_l(\theta_l, F_{l+1}))| dF_{l+1}}{\int_S 1} = \mathbb{E}_{F_{l+1}} |\det(\mathcal{J}_l(\theta_l, F_{l+1}))|$ and so $\mathbb{E}_{F_{l+1}} |\det(\mathcal{J}_l(\theta_l, F_{l+1}))| \geq 1$ and so $\mathbb{E}_{F_{l+1},\theta_l} |\det(\mathcal{J}_l(\theta_l, F_{l+1}))| \geq 1$ and so $\mathbb{E}_{F_{l+1},\theta_l} |\prod_{i=1}^{d-1} s_i| = \mathbb{E}_{F_{l+1},\theta_l} \prod_{i=1}^{d-1} |s_i| \geq 1$. But the nonzero singular values are assumed to be independent by condition (2). So $\mathbb{E}_{F_{l+1},\theta_l} \prod_{i=1}^{d-1} |s_i| = (\mathbb{E}_{F_{l+1},\theta_l} |s_i|)^{d-1}$ for $1 \leq i \leq d-1$. So $(\mathbb{E}_{F_{l+1},\theta_l} |s_i|)^{d-1} \geq 1$ and so $\mathbb{E}_{F_{l+1},\theta_l} |s_i| \geq 1$ for $1 \leq i \leq d-1$.

Similarly, we have

$$
\mathbb{Q}_{F_{l+1},\theta_l} ||\mathcal{J}_l(\theta_l, F_{l+1})||_{qm}
$$

$$
= \mathbb{Q}_{F_{l+1},\theta_l} \sqrt{\frac{\sum_{i=1}^{d-1} s_i^2}{d-1}}
$$

$$
= \sqrt{\mathbb{E}_{F_{l+1},\theta_l} \frac{\sum_{i=1}^{d-1} s_i^2}{d-1}}
$$

$$
= \sqrt{\mathbb{E}_{F_{l+1},\theta_l} s_1^2}
$$

$$
= \sqrt{(\mathbb{E}_{F_{l+1},\theta_l} |s_1|)^2 + \mathrm{Var}_{F_{l+1},\theta_l}(|s_1|)}
$$

$$
\geq \sqrt{1 + \delta \epsilon^2}
$$

The last identity uses condition (3). Let $\theta_k^l := (\theta_l, \theta_{l+1}, .., \theta_{k-1})$.

Now, we will prove the following claim by induction:

(A) $f_l(\theta_L^l, X)$ is a uniformly distributed vector on the hypersphere independent of $\theta_l^1$.

Since $X \sim S$, by condition (4), we have $f_L(\theta_L, X) \sim S$. Also, since $X$ is independent of $\theta_L^1$ and $\theta_L$ is independent of $\theta_L^1$, $f_L(\theta_L, X)$ is independent of $\theta_L^1$, as required.

Now for the induction step. Assume (A) is true for some $l$. Then by the induction hypothesis, $f_l(\theta_L^l, X) \sim S$ and $f_l(\theta_L^l, X)$ is independent of $\theta_l^1$ and thus independent of $\theta_{l-1}^1$. Then by condition (4), $f_{l-1}(\theta_L^{l-1}, X) = f_{l-1}(\theta_{l-1}, f_l(\theta_L^l, X)) \sim S$. Also, since both $\theta_{l-1}$ and $f_l(\theta_L^l, X)$ are independent of $\theta_{l-1}^1$, so is $f_{l-1}(\theta_L^{l-1}, X)$. This completes the induction step.

Analogously, we have claim (A2): $f_l(\theta_k^l, F_k)$ is a uniformly distributed vector on the hypersphere independent of $\theta_l^1$.

Now we will prove the following claim by induction on $k - l$.

(B) For any unit length vector $u$, $\mathcal{J}_k^l(\theta_k^l, F_k) u$ can be written as $\ell_k^l(\theta_k^l, F_k, u) u_l$. Here, $\ell_k^l$ is a random scalar independent of both $u_l$ and $f_l(\theta_k^l, F_k)$. $u_l$, conditioned on $f_l(\theta_k^l, F_k)$, is uniformly distributed in the space of unit length vectors orthogonal to $f_l(\theta_k^l, F_k)$.

The case $k = l + 1$ is equivalent to condition (5). For the induction step, consider

$$\mathcal{J}_k^l(\theta_k^l, F_k)u$$
$$= \mathcal{J}_{l+1}^l(\theta_l, f_{l+1}(\theta_k^{l+1}, F_k))\mathcal{J}_k^{l+1}(\theta_k^{l+1}, F_k)u$$
$$= \mathcal{J}_{l+1}^l(\theta_l, f_{l+1}(\theta_k^{l+1}, F_k))\ell_k^{l+1}(\theta_k^{l+1}, F_k, u)u_{l+1}$$
$$= \ell_k^{l+1}(\theta_k^{l+1}, F_k, u)\mathcal{J}_{l+1}^l(\theta_l, f_{l+1}(\theta_k^{l+1}, F_k))u_{l+1}$$
$$= \ell_k^{l+1}(\theta_k^{l+1}, F_k, u)\ell_{l+1}^l(\theta_l, f_{l+1}, u_{l+1})u_l$$

Here, we use claim (A2) and the induction hypothesis twice. Let $\ell_k^l(\theta_k^l, F_k, u) := \ell_k^{l+1}(\theta_k^{l+1}, F_k, u)\ell_{l+1}^l(\theta_l, f_{l+1}, u_{l+1})$. Then $\mathcal{J}_k^l(\theta_k^l, F_k)u$ has the required decomposition. From when we used the induction hypothesis the first time, we obtained that $\ell_k^{l+1}$ is independent of $f_{l+1}$ and $u_{l+1}$, and therefore of $f_l(\theta_l, f_{l+1})$ and $\mathcal{J}_{l+1}^l(\theta_l, f_{l+1})$, and therefore of $u_l$. From when we used the induction hypothesis the second time, we obtain that $\ell_{l+1}^l$ is independent of $u_l$ and $f_l$. Therefore, both $\ell_k^{l+1}$ and $\ell_{l+1}^l$ are independent of both $u_l$ and $f_l$, and then so is $\ell_k^l$. From using claim (B) the second time we also obtained that conditioned on $f_l$, $u_l$ is a uniformly distributed among unit vectors orthogonal to $f_l$. Therefore, the decomposition $\mathcal{J}_k^l(\theta_k^l, F_k)u = \ell_k^l u_l$ fulfills the required conditions, so the induction step is complete.

Now we will prove the main claim by induction on $k - l$.

We will begin with the case $k = l + 1$. We have $\mathbb{Q}_{\theta,X}GSC(l+1, l, f, \theta, X) = \mathbb{Q}_{\theta,X}\sqrt{\frac{d-1}{d}}\frac{||\mathcal{J}_{l+1}^l||_{qm}||f_{l+1}||_2}{||f_l||_2}$. (Note that the qm norm was redefined.) Because the domain of all layer functions is the hypersphere, we have $||f_{l+1}||_2 = ||f_l||_2$ and so $\mathbb{Q}_{\theta,X}GSC(l+1, l, f, \theta, X) = \sqrt{\frac{d-1}{d}}\mathbb{Q}_{\theta,X}||\mathcal{J}_{l+1}^l||_{qm} = \sqrt{\frac{d-1}{d}}\mathbb{Q}_{\theta,X}||\mathcal{J}_{l+1}^l(\theta_l, f_{l+1}(\theta_L^{l+1}, X))||_{qm}$. But by claim (A), $f_{l+1}(\theta_L^{l+1}, X) \sim \mathcal{S}$ and $f_{l+1}(\theta_L^{l+1}, X)$ is independent of $\theta_l$. So $\mathbb{Q}_{\theta,X}||\mathcal{J}_{l+1}^l||_{qm} \geq \sqrt{1 + \delta\epsilon^2}$, so $\mathbb{Q}_{\theta,X}GSC(l+1, l, f, \theta, X) \geq \sqrt{\frac{d-1}{d}}\sqrt{1 + \delta\epsilon^2}$.

Now for the induction step. Let $U$ be the uniform distribution over the unit hypersphere. As before, we have $\mathbb{Q}_{\theta,X}GSC(k, l, f, \theta, X) = \sqrt{\frac{d-1}{d}}\mathbb{Q}_{\theta,X}||\mathcal{J}_k^l||_{qm}$ and so

$$\mathbb{Q}_{\theta,X}GSC(k, l, f, \theta, X)$$
$$= \sqrt{\frac{d-1}{d}}\mathbb{Q}_{\theta,X}||\mathcal{J}_{l+1}^l\mathcal{J}_k^{l+1}||_{qm}$$
$$= \sqrt{\frac{d-1}{d}}\mathbb{Q}_{\theta,X,u\sim U}||\mathcal{J}_{l+1}^l\mathcal{J}_k^{l+1}u||_2$$
$$= \sqrt{\frac{d-1}{d}}\mathbb{Q}_{\theta,X,u\sim U}||\mathcal{J}_{l+1}^l(\theta_l, f_{l+1}(\theta_L^{l+1}, X))\mathcal{J}_k^{l+1}(\theta_k^{l+1}, f_k(\theta_L^k, X))u||_2$$

$f_k(\theta_L^k, X)$ is $\sim \mathcal{S}$ and independent of $\theta_k^{l+1}$ by claim (A). So by claim (B), we have that $\mathcal{J}_k^{l+1}u$ can be written as $\ell_k^{l+1}(\theta_k^{l+1}, f_k, u)u_{l+1}$ with the properties stated in claim (B). Using those properties, we obtain

$$\mathbb{Q}_{\theta,X}GSC(k, l, f, \theta, X)$$
$$= \mathbb{Q}_{\theta,X,u\sim U}||\mathcal{J}_{l+1}^l(\theta_l, f_{l+1}(\theta_L^{l+1}, X))\mathcal{J}_k^{l+1}(\theta_k^{l+1}, f_k(\theta_L^k, X))u||_2$$
$$= (\mathbb{Q}_{\ell_k^{l+1}}\ell_k^{l+1})(\mathbb{Q}_{\theta_l, f_{l+1}, u_{l+1}}||\mathcal{J}_{l+1}^l(\theta_l, f_{l+1})u_{l+1}||_2)$$

Let's look at the first term $\mathbb{Q}_{\ell_k^{l+1}}\ell_k^{l+1}$. We have

$$\mathbb{Q}_{\ell_k^{l+1}}\ell_k^{l+1}$$

$$
\begin{aligned}
&= \mathbb{Q}_{\theta, X, u} ||\mathcal{J}_k^{l+1} u||_2 \\
&= \mathbb{Q}_{\theta, X} GSC(k, l+1, f, \theta, X) \\
&\geq \sqrt{\frac{d-1}{d}} \sqrt{1 + \delta \epsilon^2}^{k-l-1}
\end{aligned}
$$

The last line comes from the induction hypothesis.

Now, let's look at the second term $\mathbb{Q}_{\theta_l, f_{l+1}, u_{l+1}} ||\mathcal{J}_{l+1}^l(\theta_l, f_{l+1}) u_{l+1}||_2$. $u_{l+1}$ if uniform among unit length vectors orthogonal to $f_{l+1}$. But this leads to $u_{l+1}$ being orthogonal to the normal of the hypersphere at $f_{l+1}$ and thus orthogonal to the right null space of $\mathcal{J}_{l+1}^l$. Since $u_{l+1}$ is also independent of $\theta_l$, we have $\mathbb{Q}_{\theta_l, f_{l+1}, u_{l+1}} ||\mathcal{J}_{l+1}^l(\theta_l, f_{l+1}) u_{l+1}||_2 = \mathbb{Q}_{\theta_l, f_{l+1}} ||\mathcal{J}_{l+1}^l(\theta_l, f_{l+1})||_{qm}$. By claim (A), $f_{l+1}$ is $\sim \mathcal{S}$ and independent of $\theta_l$, so $\mathbb{Q}_{\theta_l, f_{l+1}} ||\mathcal{J}_{l+1}^l(\theta_l, f_{l+1})||_{qm} \geq \sqrt{1 + \delta \epsilon^2}$.

Putting those results together, we obtain

$$
\begin{aligned}
&\mathbb{Q}_{\theta, X} GSC(k, l, f, \theta, X) \\
&= (\mathbb{Q}_{\ell_k^{l+1}} \ell_k^{l+1})(\mathbb{Q}_{\theta_l, f_{l+1}, u_{l+1}} ||\mathcal{J}_{l+1}^l(\theta_l, f_{l+1}) u_{l+1}||_2) \\
&\geq \sqrt{\frac{d-1}{d}} \sqrt{1 + \delta \epsilon^2}^{k-l-1} \sqrt{1 + \delta \epsilon^2} \\
&= \sqrt{\frac{d-1}{d}} \sqrt{1 + \delta \epsilon^2}^{k-l}
\end{aligned}
$$

This is the desired claim.

$\square$

Let's look at the conditions.

Condition (1) is standard for randomly initialized weight matrices.

Conditions (4) and (5) are both fulfilled if the last two operations of each layer function are multiplication with a weight matrix and length-only layer normalization and that weight matrix is Gaussian or orthogonally initialized.

If the weight matrix is orthogonally initialized, this is easy to see, because the linear transformation and the normalization operation commute. If we exchange those two operations then the last operation applied in both $f_l(\theta_l, F_{l+1})$ and $\mathcal{J}_l(\theta_l, F_{l+1}) u$ is the orthogonal transformation, which decouples the orientations of both terms from the length of $\mathcal{J}_l(\theta_l, F_{l+1}) u$ as well as decoupling the orientations of the terms from each other up to preserving their angle. Finally, note that $\mathcal{J}_l(\theta_l, F_{l+1}) u$ always lies in the left-null space of $\mathcal{J}_l(\theta_l, F_{l+1})$. But that space is orthogonal to $f_l(\theta_l, F_{l+1})$, and hence the two terms are orthogonal as required.

If the weight matrix is Gaussian initialized, note that the product of a Gaussian initialized matrix and an orthogonally initialized matrix is Gaussian initialized. Hence, we can insert an additional orthogonally initialized matrix and then proceed with the previous argument to show that conditions (4) and (5) are fulfilled.

After applying a linear transformation with one of the two initializations, conditions (4) and (5) hold except for the length of $f_l$ is not 1. Hence, even if length-only layer normalization is not used as part of the endomorphism, we expect (4) and (5) to hold approximately in practice.

As far as we can tell, conditions (2) and (3) are not fulfilled in practice. They are both used to derive from unit determinants a greater than unit $qm$ norm. As long this implications holds for practical layer functions, (2) and (3) are not necessary.

### F.3 THEOREM 3 - SKIP CONNECTIONS REDUCE THE GRADIENT

**Theorem 3.** *Let $g$ and $u$ be random vectors. Consider a function $f$ that is $k$-diluted with respect to $u$, a matrix $S$ and a function $\rho$. Let $\mathcal{R}(v)$ be the Jacobian of $\rho$ at input $v$. Let $r := \frac{\mathbb{Q} ||g \mathcal{R}(u)||_2 \mathbb{Q} ||u||_2}{\mathbb{Q} ||\rho(u)||_2 \mathbb{Q} ||g||_2}$.*

*Assume that* $\mathbb{E}(Su).(\rho(u)) = 0$ *and that* $\mathbb{E}(g\mathcal{R}(u)).(gS) = 0$. *Also assume that* $\frac{\mathbb{Q}||gS||_2\mathbb{Q}||u||_2}{\mathbb{Q}||Su||_2\mathbb{Q}||g||_2} = 1$.
*Then* $\frac{\mathbb{Q}||g\mathcal{R}(u)+gS||_2\mathbb{Q}||u||_2}{\mathbb{Q}||\rho(u)+Su||_2\mathbb{Q}||g||_2} = 1 + \frac{r-1}{k^2+1} + O((r-1)^2)$.

*Proof.* We have

$$\frac{\mathbb{Q}||g\mathcal{R}(u)+gS||_2\mathbb{Q}||u||_2}{\mathbb{Q}||\rho(u)+Su||_2\mathbb{Q}||g||_2}$$

$$= \frac{\mathbb{Q}\sqrt{||g\mathcal{R}(u)+gS||_2^2}\mathbb{Q}||u||_2}{\mathbb{Q}\sqrt{||\rho(u)+Su||_2^2}\mathbb{Q}||g||_2}$$

$$= \frac{\sqrt{\mathbb{E}||g\mathcal{R}(u)+gS||_2^2}\mathbb{Q}||u||_2}{\sqrt{\mathbb{E}||\rho(u)+Su||_2^2}\mathbb{Q}||g||_2}$$

$$= \frac{\sqrt{\mathbb{E}(g\mathcal{R}(u)+gS).(g\mathcal{R}(u)+gS)}\mathbb{Q}||u||_2}{\sqrt{\mathbb{E}(\rho(u)+Su).(\rho(u)+Su)}\mathbb{Q}||g||_2}$$

$$= \frac{\sqrt{\mathbb{E}[(g\mathcal{R}(u)).(g\mathcal{R}(u))+2(gS).(g\mathcal{R}(u))+(gS).(gS)]}\mathbb{Q}||u||_2}{\sqrt{\mathbb{E}[\rho(u).\rho(u)+2(Su).\rho(u)+(Su).(Su)]}\mathbb{Q}||g||_2}$$

$$= \frac{\sqrt{\mathbb{E}[(g\mathcal{R}(u)).(g\mathcal{R}(u))+(gS).(gS)]}\mathbb{Q}||u||_2}{\sqrt{\mathbb{E}[\rho(u).\rho(u)+(Su).(Su)]}\mathbb{Q}||g||_2}$$

$$= \frac{\sqrt{(\mathbb{Q}||g\mathcal{R}(u)||_2)^2+(\mathbb{Q}||gS||_2)^2}\mathbb{Q}||u||_2}{\sqrt{(\mathbb{Q}||\rho(u)||_2)^2+(\mathbb{Q}||Su||_2)^2}\mathbb{Q}||g||_2}$$

$$= \frac{\sqrt{(\frac{r\mathbb{Q}||\rho(u)||_2\mathbb{Q}||g||_2}{\mathbb{Q}||u||_2})^2+(\mathbb{Q}||gS||_2)^2}\mathbb{Q}||u||_2}{\sqrt{(\mathbb{Q}||\rho(u)||_2)^2+(\mathbb{Q}||Su||_2)^2}\mathbb{Q}||g||_2}$$

$$= \frac{\sqrt{(\frac{r\mathbb{Q}||Su||_2\mathbb{Q}||g||_2}{k\mathbb{Q}||u||_2})^2+(\mathbb{Q}||gS||_2)^2}\mathbb{Q}||u||_2}{\sqrt{(\frac{\mathbb{Q}||Su||_2}{k})^2+(\mathbb{Q}||Su||_2)^2}\mathbb{Q}||g||_2}$$

$$= \frac{\sqrt{(\frac{r\mathbb{Q}||u||_2\mathbb{Q}||g||_2}{k\mathbb{Q}||u||_2})^2+(\mathbb{Q}||g||_2)^2}\mathbb{Q}||u||_2}{\sqrt{(\frac{\mathbb{Q}||u||_2}{k})^2+(\mathbb{Q}||u||_2)^2}\mathbb{Q}||g||_2}$$

$$= \frac{\sqrt{\frac{r^2}{k^2}+1}}{\sqrt{\frac{1}{k^2}+1}}$$

$$= \frac{\sqrt{k^2+r^2}}{\sqrt{k^2+1}}$$

$$= \frac{\sqrt{k^2+(1+(r-1))^2}}{\sqrt{k^2+1}}$$

$$= \frac{\sqrt{k^2+1+2(r-1)+(r-1)^2}}{\sqrt{k^2+1}}$$

$$= \sqrt{1+\frac{2}{k^2+1}(r-1)+\frac{1}{k^2+1}(r-1)^2}$$

$$= 1+\frac{r-1}{k^2+1}+O((r-1)^2)$$

$\square$

$u$ represents the incoming activation vector of some residual block and $g$ the incoming gradient. $r$ represents a type of expectation over the ratio $\frac{GSC(b+1,0)}{GSC(b,0)} = \frac{\frac{||\mathcal{J}_{b+1}^0||_{qm}||f_{b+1}||_2}{||f_0||_2}}{\frac{||\mathcal{J}_b^0||_{qm}||f_b||_2}{||f_0||_2}} = \frac{||g_{b+1}||_2||f_{b+1}||_2}{||g_b||_2||f_b||_2} = \frac{||g_b\mathcal{R}(f_{b+1})||_2||f_{b+1}||_2}{||g_b||_2||\rho(f_{b+1})||_2} = \frac{||g\mathcal{R}(u)||_2||u||_2}{||\rho(u)||_2||g||_2}$, ignoring the skip connection. Therefore $r$ can be viewed as the growth of the GSC. Similarly, $\frac{\mathbb{Q}||g\mathcal{R}(u)+gS||_2\mathbb{Q}||u||_2}{\mathbb{Q}||\rho(u)+Su||_2\mathbb{Q}||g||_2}$ represents the growth of the GSC after the skip connection has been added.

The key assumptions are $\mathbb{E}(Su).(\rho(u)) = 0$ and $\mathbb{E}(g\mathcal{R}(u)).(gS) = 0$. In plain language, we assume that the function computed by the skip connection is uncorrelated to the function computed by the residual block and that the same is true for the gradient flowing through them. For the forward direction, this is true if either the skip connection is Gaussian / orthogonally initialized or the last layer of the residual block is linear and Gaussian / orthogonally initialized and if the randomness of the initialization is absorbed into the expectation. Unfortunately, for the backward direction, such a statement cannot be made because the gradient has a complex dependence both on $S$ and $\mathcal{R}$. However, the assumption that this dependence between the forward and backward direction is immaterial has proven realistic in mean field theory based studies (see section B.1.1). Under this assumption, as in the forward direction, we require that either the skip connection is Gaussian / orthogonally initialized or the first layer of the residual block is linear and Gaussian / orthogonally initialized. Note that even if both assumptions are only fulfilled approximately, this is not catastrophic to the theorem.

The other assumption is $\frac{\mathbb{Q}||gS||_2\mathbb{Q}||u||_2}{\mathbb{Q}||Su||_2\mathbb{Q}||g||_2} = 1$. This is true if $S$ is an orthogonal matrix and so specifically if $S$ is the identity matrix. If $S$ is Gaussian / orthogonally initialized, this is true if the randomness of the initialization is absorbed into the $\mathbb{Q}$ terms. Again, if this assumption only holds approximately, it is not catastrophic to the theorem.

An implicit assumption made is that the distribution of the incoming gradient $g$ is unaffected by the addition of the skip connection, which is of course not quite true in practice. The addition of the skip connection also has an indirect effect on the distribution and scale of the gradient as it flows further towards the input layer.

The experiments in figure 3 bear out the theory discussed here.

# G  TAYLOR APPROXIMATION OF A NEURAL NETWORK

We define the first-order Taylor approximation $T_l$ of the bottom layers up to layer $l$ recursively. Write $i_l(x)$ as the short form of $i_l(i_{l+1}(..i_L(x)..))$. Then

$$T_L(\theta, X) = f_L(\theta_L, X)$$

$$T_l(\theta, X) = i_l(T_{l+1}(\theta, X)) + r_l(\theta_l, i_{l+1}(X)) + \sum_{k=l+1}^{L} \frac{dr_l(\theta_l, i_{l+1}(X))}{di_k(X)} r_k(\theta_k, i_{k+1}(X)) \text{ for } l < L$$

The maximum number of parametrized residual functions composed in $T_l$ is 2. Otherwise, only addition and composition with fixed functions is used. Hence, the compositional depth of $T_l$ is $\min(L - l, 2)$. Hence, the network $f_{\text{Taylor}(l)} := f_0(y, f_1(..f_{l-1}(T_l(X))..))$ has compositional depth $\max(l + 1, L)$.

For ResNet architectures, as in section D.2, we divide each layer in the residual block into its initial and residual function. Then the definition of the Taylor expansion remains as above, except a term $s_l(T_m(\theta, X))$ is added at each layer $l$ where a skip connection, represented by skip function $s_l$, terminates. $T_m$ is the Taylor expansion at the layer where the skip connection begins.

# H  LOOKS-LINEAR INITIALIZATION

The looks-linear initialization ('LLI') of ReLU MLPs achieves an approximate orthogonal initial state. Consider a ReLU MLP with some number of linear layers and a ReLU layer between each

pair of linear layers. LLI initializes the weight matrix of the lowest linear layer differently from the weight matrix of the highest linear layer and differently from the weight matrices of the intermediate linear layers. Let a weight matrix $W$ have dimension $m*n$, where $n$ is the dimension of the incoming vector and $m$ is the dimension of the linear layer itself. Also, we require that the dimension of all ReLU layers, and thus the dimension of all linear layers except the highest linear layer, is even. Then the weight matrices are initialized as follows.

- Lowest linear layer: Draw a uniformly random orthogonal matrix $W'$ of dimension $\max(\frac{m}{2}, n) * \max(\frac{m}{2}, n)$. Then, for all $1 \leq i \leq \frac{m}{2}$ and $1 \leq j \leq n$, set $W(2i, j) = \max(\sqrt{\frac{m}{2n}}, 1)W'(i, j)$ and $W(2i + 1, j) = -\max(\sqrt{\frac{m}{2n}}, 1)W'(i, j)$.

- Highest linear layer: Draw a uniformly random orthogonal matrix $W'$ of dimension $\max(m, \frac{n}{2}) * \max(m, \frac{n}{2})$. Then, for all $1 \leq i \leq m$ and $1 \leq j \leq \frac{n}{2}$, set $W(i, 2j) = \max(\sqrt{\frac{2m}{n}}, 1)W'(i, j)$ and $W(i, 2j + 1) = -\max(\sqrt{\frac{2m}{n}}, 1)W'(i, j)$.

- Intermediate linear layers: Draw a uniformly random orthogonal matrix $W'$ of dimension $\max(\frac{m}{2}, \frac{n}{2}) * \max(\frac{m}{2}, \frac{n}{2})$. Then, for all $1 \leq i \leq \frac{m}{2}$ and $1 \leq j \leq \frac{n}{2}$, set $W(2i, 2j) = \max(\sqrt{\frac{m}{n}}, 1)W'(i, j)$, $W(2i + 1, 2j) = -\max(\sqrt{\frac{m}{n}}, 1)W'(i, j)$, $W(2i, 2j+1) = -\max(\sqrt{\frac{m}{n}}, 1)W'(i, j)$ and $W(2i+1, 2j+1) = \max(\sqrt{\frac{m}{n}}, 1)W'(i, j)$.

Under LLI, pairs of neighboring ReLU neurons are grouped together to effectively compute the identity function. The incoming signal is split between ReLU neurons of even and odd indeces. Each of the two groups preserves half the signal, which are then "stitched together" in the next linear layer only to be re-divided in a different way to pass through the next ReLU layer.

LLI networks can be said to be approximately orthogonal. Let $X_{l_r}$ be the representation computed by the $r$'th ReLU layer. Then let $\chi'_{l_r}(i) = X_{l_r}(2i) - X_{l_r}(2i + 1)$ for $1 \leq i \leq \frac{d_{l_r}}{2}$. Since $X_{l_r}(2i)$ or $X_{l_r}(2i + 1)$ is 0, this transformation is bijective. Finally, the transformation from $\chi_{l_r}$ to $\chi_{l_{r-1}}$ is an orthogonal transformation.

# I   EXPERIMENTAL DETAILS

## I.1   ARCHITECTURES USED

**Vanilla networks without skip connections**   All networks are MLPs composed of only fully-connected linear layers and unparametrized layers. The following types of layers are used.

- linear layer: $f_l(\theta_l, f_{l+1}) = W_l f_{l+1}$ where the entries of the weight matrix $W_l$ are the entries of the parameter sub-vector $\theta_l$. Trainable bias parameters are not used.

- ReLU layer: $f_l(f_{l+1}) = \sigma_{\text{ReLU}}.(f_{l+1})$, where the scalar function $\sigma_{\text{ReLU}}$ is applied element-wise as indicated by .() We have $\sigma_{\text{ReLU}}(a) = a$ if $a \geq 0$ and $\sigma_{\text{ReLU}}(a) = 0$ if $a < 0$.

- tanh layer: $f_l(f_{l+1}) = \sigma_{\text{tanh}}.(f_{l+1})$, where $\sigma_{\text{tanh}}(a) = \tanh(a)$.

- SeLU layer: $f_l(f_{l+1}) = \sigma_{\text{SeLU}}.(f_{l+1})$. We have $\sigma_{\text{SeLU}}(a) = c_{\text{pos}}a$ if $a \geq 0$ and $\sigma_{\text{SeLU}}(a) = c_{\text{neg}}(e^a - 1)$ if $a < 0$. We set $c_{\text{pos}} = 1.0507$ and $c_{\text{neg}} = 1.0507 * 1.6733$ as suggested by Klambauer et al. (2017).

- batch normalization layer: $f_l(f_{l+1}) = \frac{f_{l+1} - \mu}{\sigma}$, where $\mu$ is the component-wise mean of $f_{l+1}$ over the current batch and $\sigma$ is the componentwise standard deviation of $f_{l+1}$ over the current batch.

- layer normalization layer: $f_l(f_{l+1}) = \frac{f_{l+1} - \mu}{\sigma}$, where $\mu$ is mean of the entries of $f_{l+1}$ and $\sigma$ is the standard deviation of the entries of $f_{l+1}$.

- length-only layer normalization layer: $f_l(f_{l+1}) = \frac{f_{l+1}}{qm}$, where $qm$ is the quadratic mean of the entries of $f_{l+1}$.

- dot product error layer: $f_0(f_1, y) = f_1.y$

- softmax layer: $f_l(f_{l+1})(i) = \frac{e^{f_{l+1}(i)}}{\sum_j e^{f_{l+1}(j)}}$

- kl error layer: $f_0(f_1, y) = \ln f_1(y)$ where $y$ is an integer class label and $f_1$ has one entry per class.

Note that normalization layers (batch normalization, layer normalization or length-only layer normalization) do not use trainable bias and variance parameters.

A network of compositional depth $N$ contains $N$ linear layers and $N-1$ nonlinearity layers (ReLU, tanh or SeLU) inserted between those linear layers. If the network uses normalization layers, one normalization layer is inserted after each linear layer. For Gaussian noise experiments, the error layer is the dot product error layer. For CIFAR10 experiments, a softmax layer is inserted above the last linear or normalization layer and the error layer is a kl error layer.

For Gaussian noise experiments, data inputs as well as predictions and labels have dimension 100. We used a compositional depth of 50. We generally used a uniform width of 100 throughout the network. However, we also ran experiments where the width of all layers from the first linear layer to the layer before the last linear layer had width 200. We also ran experiments where linear layers alternated in width between 200 and 100. For CIFAR10 experiments, data inputs have dimension 3072 and predictions have dimension 10. We use a compositional depth of 51. The first linear layer transforms the width to 100 and the last linear layer transformed the width to 10.

The following initialization schemes for the weight matrices are used.

- Gaussian: Each entry of the weight matrix is drawn as an independent Gaussian with mean 0. The variance of this Gaussian is one over the dimension of the incoming vector except when the weight matrix follows a ReLU layer. In that case, the variance of the Gaussian is two over the dimension of the incoming vector.
- orthogonal: The weight matrix is a uniformly random orthogonal matrix. Note that this initialization scheme is only used for square matrices.
- looks-linear: See section H.

**ResNet**    In all cases, the first layer is a linear layer. After that, there are 25 skip connections. Each skip connection bypasses a block of 6 layers: a normalization layer, a nonlinearity layer, a linear layer, another normalization layer, another nonlinearity layer, and another linear layer. Above the last skip connection, a final normalization layer is inserted, followed by softmax (CIFAR10 only) and then the error layer. For Gaussian noise experiments, we use a constant width of 100. For CIFAR10, the first linear layer transforms the width from 3072 to 100, and the last skip connection as well as the last linear linear in the last residual block transform the width from 100 to 10.

Skip connections are identity skip connections, except the last skip connection in CIFAR10 experiments that is responsible for reducing the width. There, the skip connection multiplies its incoming value by a fixed $10 * 100$ submatrix of a $100 * 100$ orthogonal matrix. For Gaussian noise experiments, we also conducted some experiments where skip connections used random matrices where each entry is drawn from an independent Gaussian with mean 0 and the variance being one over the dimension of the incoming vector.

## I.2    PROTOCOL FOR GAUSSIAN NOISE EXPERIMENTS

For Gaussian noise experiments, both inputs and labels are 100-dimensional vectors were each entry is drawn from an independent Gaussian with mean 0 and variance $\frac{1}{100}$. We normalized the input vectors to have length 10. We drew 100 independent datasets of size 10.000.

For each dataset and each architecture we studied (see table 1 for the full list), we computed both the forward activations and the gradient for each datapoint. For architectures with batch normalization, all 10.000 datapoints were considered part of a single batch. Note that no training was conducted. We then computed the following metrics:

- Expected GSC: At each layer $l$, we computed $\frac{\mathbb{Q}_D ||\mathcal{J}_l^0||_{qm} \mathbb{Q}_D ||f_l||_2}{\mathbb{Q}_D ||f_0||_2}$. Note that $\mathcal{J}_l^0$ is simply the "regular gradient" of the network.
- Pre-activation standard deviation: For each nonlinearity layer $l$, we computed the standard deviation of the activations of each neuron in $f_{l+1}$ over the 10.000 datapoints, i.e.

$(\mathbb{E}_D f_{l+1}(i)^2) - (\mathbb{E}_D f_{l+1}(i))^2$ for all $1 \le i \le d_{l+1}$. We then computed the quadratic mean of those standard deviations as a summary statistic for the layer.

- Pre-activation sign diversity: For each nonlinearity layer $l$, at each neuron in $f_{l+1}$, we computed $\min(pos, 1 - pos)$, where $pos$ is the fraction of activations that were positive across the 10.000 datapoints. We then computed the mean of those values across the layer as a summary statistic.

Finally, we obtained a summary statistic for each layer and architecture by averaging the results over the 100 datasets. Results are shown in table 1, figure 1 and figure 3.

## I.3 PROTOCOL FOR CIFAR10 EXPERIMENTS

For CIFAR10 experiments, we preprocessed each feature to have zero mean and unit variance. We used the training set of 50.000 datapoints and disregarded the test set. We used batches of size 1.000 except for the vanilla batch-ReLU architecture with Gaussian initialization, for which we used a batch size of 50.000. (See section 4.5 for the explanation.)

We trained each architecture we studied (see table 2 for the full list) with SGD in two ways. First, with a single step size for all layers. Second, with a custom step size for each layer.

**Single step size** We perform a grid search over the following starting step sizes: $\{1e5, 3e4, 1e4, 3e3, .., 1e - 4, 3e - 5, 1e - 5\}$. For each of those 21 starting step sizes, we train the network until the end-of-epoch training classification error has not decreased for 5 consecutive epochs. Once that point is reached, the step size is divided by 3 and training continues. Once the end-of-epoch training classification error has again not decreased for 5 epochs, the step size is divided by 3 again. This process is repeated until training terminates. Termination occurs either after 500 epochs or after the step size is divided 11 times, whichever comes first. The starting step size that obtains the lowest final training classification error is selected as the representative step size for which results are presented in the paper.

**Custom step sizes** In this scenario, we use a different starting step size for each layer. After those step sizes are computed, smoothed and scaled as described in section I.4, we train the pre-trained network with those step sizes. As before, periodically, we divide all step sizes jointly by 3. As before, training is terminated after 11 divisions or when 500 epochs are reached, whichever comes first.

We compute the following metrics:

- Largest relative update size for each layer induced by the estimated optimal step size during the epoch where that optimal step size was estimated. See section I.4 for details.
- Effective depth throughout training: see section D.2 for details. $\lambda$-contributions are accumulated from batch to batch.
- Training classification error at the end of each epoch.
- Training classification error when compositional depth is reduced via Taylor expansion after training: see section G for details.
- GSC, pre-activation standard deviation and pre-activation sign diversity: for details, see the end of section I.2. Note that the expectations over the dataset were computed by maintaining exponential running averages across batches.
- Operator norms of residual weight matrices after training.

See table 2 and figures 2, 4, 5 and 6 for results.

## I.4 SELECTING CUSTOM STEP SIZES

We estimated the optimal step size for each linear layer under SGD for our CIFAR10 experiments. This turned out to be more difficult than expected. In the following, we describe the algorithm we used. It has five stages.

**Pre-training** We started by pre-training the network. We selected a set of linear layers in the network that we suspected would require similar step sizes. In exploding architectures (vanilla batch-ReLU with Gaussian initialization, vanilla layer-tanh, vanilla batch-tanh, SeLU), we chose the second highest linear layer through the sixth highest linear layer for pretraining, i.e. 5 linear layers in total. We expected these layers to require a similar step size because they are close to the output and the weight matrices have the same dimensionality. For vanilla ReLU, vanilla layer-ReLU, vanilla tanh and looks-linear initialization, we chose the second lowest linear layer through the second highest linear layer (i.e. 49 linear layers in total) because the weight matrices have the same dimensionality. Finally, for ResNet, we chose the second lowest through the third highest linear layer (i.e. 48 linear layers in total), because the blocks those layers are in have the same dimensionality.

We then trained those layers with a step size that did not cause a single relative update size of more than 0.01 (exploding architectures) or 0.001 (other architectures) for any of the pre-trained layers or any batch. We chose small step sizes for pre-training to ensure that pre-training would not impact effective depth. We pre-trained until the training classification error reached 85%, but at least for one epoch and at most for 10 epochs. The exact pre-training step size was chosen via grid search over a grid with multiplicative spacing of 3. The step size chosen was based on which step size reached the 85% threshold the fastest. Ties were broken by which step size achieved the lowest error.

**Selection** In the selection phase, we train each linear layer one after the other for one epoch while freezing the other layers. After each layer is trained, the change to the parameter caused by that epoch of training is undone before the next layer is trained. For each layer, we chose a step size via grid search over a grid with multiplicative spacing 1.5. The step size that achieved the lowest training classification error after the epoch was selected. Only step sizes that did not cause relative update sizes of 0.1 or higher were considered, to prevent weight instability.

Now we can explain the need for pre-training. Without pre-training, the selection phase yields very noisy and seemingly random outcomes for many architectures. This is because it was often best to use a large step size to jump from one random point in parameter space to the next, hoping to hit a configuration at the end of the epoch where the error was, say, 88%. Since we used a tight spacing of step sizes, for most layers, there was at least one excessively large step size that achieved this spurious "success". Since we only trained a single layer out of 51 for a single epoch, the error of the "correct" step size after pre-training often did not reach, say, 88%. When we trained the network for 500 epochs with those noisy estimates, we obtained very high end-of-training errors.

Pre-training ensures that training with an excessively high step size causes the error to exceed 85% again. Therefore, those step sizes are punished and step sizes that ultimately lead to a much better end-of-training error are selected.

**Clipping** Even though pre-training was used, for some architectures, it was still beneficial to add the following restriction: as we consider larger and larger step sizes during grid search, as soon as we find a step size for which the error is at least $0.1\%$ higher than for the current best step size, the search is terminated. Clipping is capable of further eliminating outliers and was used if and only it improved the end-of-training error. It was used for vanilla tanh, ResNet layer-tanh and looks-linear layer-ReLU.

For each linear layer, the largest relative update size induced by the step size obtained for that layer after the clipping phase (or after the selection phase if clipping was not used) during the epoch of training conducted in the selection phase is shown in the in figures 2A, 4A, 5A and 6A.

**Smoothing** In this stage, we built a mini-regression dataset of $(X, Y)$ points as follows. For each $X$ from 1 to 51, we include the point $(X, Y)$ where $Y$ is the largest relative update size the step size selected for linear layer $X$ after clipping induced during the epoch of training in the selection phase. We then fit a line via least-squares regression on that dataset in log scale. For each $X$, we thus obtain a smoothed value $Y'$. The ratio $\frac{Y'}{Y}$ was multiplied to the step size obtained for each layer at the end of the clipping phase.

We added this phase because we found that the end-of-training error could still be significantly improved by reducing noise among the layer-wise step sizes in this way.

**Scaling** Finally, we jointly scale all layer-wise step sizes with a single constant. That value is chosen as in the selection phase by trying a small constant, training for one epoch, rewinding that epoch, multiplying that constant by 1.5, rinse, repeat. Again, that process was terminated once any layer experiences an update of relative size at least 0.1. This stage is necessary because the size of the update on the entire parameter vector when all layers are trained jointly is $\approx \sqrt{51}$ times larger than when only single layers are trained as in the selection phase. Hence, a scaling constant less than 1 is usually needed to compensate. Again, some architectures benefited from using clipping, where we terminated the scaling constant search as soon as one exhibited an error more than 0.1% above the current best scaling constant. Vanilla tanh, vanilla layer-tanh, ResNet layer-tanh and looks-linear layer-ReLU used this clipping.

Formally, for each architecture, we trained three networks to completion. One using no clipping, one using only clipping during the scaling phase, and using the clipping phase as well as clipping during the scaling phase. Whichever of these three networks had the lowest end-of-training error was selected for presentation in the paper. To compare, for single step size training, we compared 21 end-of-training error values.

