# OpenReview forum: "Gradients explode - Deep Networks are shallow - ResNet explained"
_ICLR.cc/2018/Conference — Invite to Workshop Track_

### Official Review · AnonReviewer2 · 2017-11-26
**This paper needs some major reworking to emphasize the overall narrative introducing the Gradient Scale Coefficient and its potential impact.**

**Rating:** 3
**Confidence:** 2

**Review:**

Summary of paper - The paper introduces the Gradient Scale Coefficient and uses it to demonstrate issues with the current understanding of where and why exploding gradients occur.

Review - The paper attempts to contribute to the discussion about the exploding gradient problem by both introducing a metric for discussing this issue and by showing that current understanding of the exploding gradient problem may be incorrect. It is admirable that the authors are seeking to add to the understanding about theory of neural nets instead of contributing a new architecture with better error rates but without understanding why said error rates are lower. While the authors list 7 contributions, the current version of the text is a challenge to read and makes it challenging to distill an overarching theme or narrative to these contributions.

The authors do mention experiments on page 8, but confess that some of the results are somewhat underwhelming. Unfortunately, all tables with the experimental results are left to the appendix. As this is a mostly theoretical paper, pushing experimental results to the appendix does make sense, but the repeated references to these tables suggest that these experimental results are crucial for the authors’ overall points.

While the authors do attempt to accomplish a lot in these nearly 16 pages of text, the authors' main points and overall narrative gets lost due to the writing that is a bit jumbled at times and that relies heavily on the supplement. There are several places where it is not immediately clear why a certain block of text is included (i.e. the proof outlines on pages 8 and 10). At other points the authors default to an chronological narrative that can be useful at times (i.e. page 9), but here seems to distract from their overall narrative.

This paper has a lot of content, but not all of it appears to be relevant to the authors’ central points. Furthermore, the paper is nearly double the recommended page length and has a nearly 30 page supplement. My biggest recommendations for this paper are for the authors to 1) articulate one theme and then 2) look at each part (whether that be section, paragraph, or sentence) and ask what does that part contribute to that theme.



Pros -
* This paper attempts to add the understanding of neural nets instead of only contributing better error rates on benchmark datasets.
* At several points, the authors seek to make the work accessible by offering lay explanations for their more technical points.
* The practical suggestions on page 16 are a true highlight and could provide an outline for possible revisions.


Cons -
* The main narrative is lost in the text, leaving a reader unsure of the authors main points and contributions as they read. For example, the authors’ first contribution is hidden among the text presentation of section 2.
* The paper relies heavily on the supplement to make their central points.
* It is nearly double the recommended page length with a nearly 30 page supplement


Minor issues -
* Use one style for introducing and defining terms either use italics or single quotes. The latter is not recommended since the authors use double quotes in the abstract to express that the exploding gradient problem is not solved.
* The citation style of Authors (YEAR) at times leads to awkward sentence parsing.
* Given that many figures have several subfigures, the authors should consider using a package that will denote subfigures with letters.
* The block quotes in the introduction may be quite important for points later in the paper, but summarizing the points of these quotes may be a better use of space. The authors more successfully did this in paragraph 2 of the introduction.
* All long descriptions of the appendix should be carefully revisited and possibly removed due to page length considerations.
* In the text, figure 4 (which is in the supplement) is referenced before figure 3 (which is in the text).

=-=-=-= Response to the authors

During the initial reviewing period, I was unable to distill the significance of the authors’ contributions from the current literature in large part due to the nature of the writing style. After reading the authors responses and consulting the differences between the versions of the paper, my review remains the same. It should be noted that all three reviewers pointed out the length of the paper as a weakness of the paper, and that in the most recent draft, the authors made the main text of the paper longer.

Consulting the differences between the paper revisions, I was initially intrigued with the volume of differences that shown in the summary bar. Upon closer inspection, I read a much stronger introduction and appreciated the summaries at the ends of sections 4.4 and 6. However, I did notice that the majority of these changes were superficial re-orderings of the original text. Given the limited substantive changes to the main text, I did not deeply re-read the text of the paper beyond the introduction.

---

> ### Author Response · Authors · 2017-12-14
> **Review Response (2/2)**
>
> ###
>
> "The paper relies heavily on the supplement to make their central points."
>
> We moved both table 1 and table 2 to the main body of the paper in the revision.
>
> Because this paper is detail-oriented and each reader cares about a different set of details, we chose (a) to provide as much detail as possible and (b) move those details to the appendix into dedicated sections so that they would be easy to find by specific interested parties.
>
> Do you think there is any particular section, paragraph or detail from the appendix that should still be moved to the main body? If so, we would be glad to know and to fulfil such a request if it could be aligned with the preferences of the other reviewers.
>
> ###
>
> "... confess that some of the results are somewhat underwhelming."
>
> The goal of sections 3 through 6 is to demonstrate the pathologies of exploding gradients and collapsing domains. We made our neural networks very deep precisely so that these pathologies would be very clear and measurable. Pathological architectures, by definition, suffer from high errors. We include this information explicitly in the revision.
>
> Using very deep MLPs to study gradient pathologies is a well-established practice from previous works closely related to this paper (e.g. [1,2,4]).
>
> Note that we contrast these high error values with those achieved by ResNet and looks-linear initialized ReLU networks, which acheive much lower error, in section 7 / table 2.
>
> ###
>
> "Unfortunately, all tables with the experimental results are left to the appendix."
>
> Tables 1 and 2 have been moved to the main body in the revision as per this request.
>
> ###
>
> "There are several places where it is not immediately clear why a certain block of text is included (i.e. the proof outlines on pages 8 and 10)."
>
> In the revision, the proof outline of theorem 1 was removed and replaced by an informal explanation of the underlying mechanisms preceding the theorem. The proof outline of theorem 2 exists to highlight the important intermediate result that surjective endomorphisms exhibit an expected absolute determinant of 1, which leads to an expected qm norm greater than 1, which causes exploding gradients. We've added more references to this important relationship throughout the revision.
>
> ###
>
> Minor issues:
> - We used single quotes to define terms and italic to highlight important concepts. In the revision, we use single quotes to define terms AND important concepts for increased consistency. We still use italic to highlight important concepts.
> - We use the citation style provided by the ICLR latex template. I would prefer not to alter this setting. Also, the vast majority of ICLR 2018 submission use (YEAR) in their citations. However, I did miss some brackets around citations in the original version of the paper. Those brackets have been added in the revision.
> - Letters have been added to the subfigures. Thank you for this advice.
> - We removed 2 of the 4 block quotes in the introduction. We would like to keep the remaining ones to underscore the difference between our results and popular wisdom.
> - Appendix length: see above
> - Figures are numbered according to the order in which they appear in the paper, not the order in which they are referenced. Again, this is the default of the ICLR latex template / latex itself. Let me know if you would like me to alter this.
>
> ###
>
>
> We hope that we have addressed your concerns in this comment and the revised version of the paper. If you agree that the contributions of our paper are significant and have been sufficiently demonstrated, we hope that you agree that our paper is well-placed at ICLR. We look forward to hearing your thoughts and comments.
>
> [1] Schoenholz et al. Deep Information Propagation. ICLR 2107. https://arxiv.org/abs/1611.01232
>
> [2] Balduzzi et al. The Shattered Gradients Problem: If resnets are the answer, then what is the question?. ICML, 2017. https://arxiv.org/abs/1702.08591
>
> [3] Saxe et al. Exact solutions to the nonlinear dynamics of learning in deep linear neural networks. 2014. https://arxiv.org/abs/1312.6120
>
> [4] Yang & Schoenholz. Mean field residual networks: on the edge of chaos. NIPS, 2017. https://papers.nips.cc/paper/6879-mean-field-residual-networks-on-the-edge-of-chaos

---

> ### Author Response · Authors · 2017-12-14
> **Review Response (1/2)**
>
> Dear Reviewer,
>
> Thank you for your review.
>
> We think our paper makes important contributions deep learning theory, architecture design and optimization and presents a valuable addition to the recent line of work exploring the properties of deep gradients and the impact of skip connections (e.g. [1,2,3,4]). Therefore, we are disappointed that the paper was awarded a low rating without its scientific merit being criticized. Do you believe that our analysis is correct? Do you believe we succeed in supporting the claims we make in the introduction and conclusion of our paper?
>
> I just uploaded a revised version of the paper. We address the points raised in your review in this revision as well as in the comments below.
>
> ###
>
> "It is nearly double the recommended page length with a nearly 30 page supplement ... This paper has a lot of content, but not all of it appears to be relevant to the authors’ central points."
>
> In the revision, we removed some of the less central results (propositions 7 through 9) as well as high-level commentary to make the paper more focused.
>
> This paper pays attention to details that other papers often gloss over, such as the rigorous definition of exploding gradients or effective depth and the careful setting of layerwise step sizes. This rigor is what enables us to obtain important results. Also note that an important predecessor work [3] from NIPS 2017 is also 55 pages long.
>
> Much of our appendix is strictly optional for readers interested in certain specifics, such as implementation details for those interested in replicating our results or the extended related work section for those interested in pursuing research in deep learning theory. Do you see providing such details as a weak point of the paper?
>
> If you believe there are still specific results in the revision that you consider unimportant and should thus be moved to the appendix or removed entirely, please let us know.
>
> ###
>
> "While the authors list 7 contributions, the current version of the text is a challenge to read and makes it challenging to distill an overarching theme or narrative to these contributions. ... the authors main points and overall narrative gets lost due to the writing that is a bit jumbled at times... For example, the authors’ first contribution is hidden among the text presentation of section 2. "
>
> We expand both the introduction and conclusion section in the revision to make the implications and contributions of this work more clear and explicit as well as adding summary sections throughout the paper to remind the reader of the overarching goals, including that of contribution 1. We removed high-level commentary from the main paper in favor of low-level explanations and summaries.
>
> The overarching goal of the paper is to advance the theoretical understanding of the gradient properties of deep networks and provide practical insight for designing neural architectures. This is the same aim as many predecessor works (e.g. [1,2,3,4]). All these works combine a range of theoretical and experimental studies to paint an overall picture, just as we do. Our "narrative" is summarized in the new "Summary" section on page 16, which is followed by an extended list of practical recommendations and research implications.
>
> In the revision, is there still a specific goal of the paper that is unclear? Is there a term or piece of notation is not defined? Is there a statement that is ambiguous? Is there a paragraph you think is redundant or out of place?

---

> ### Author Response · Authors · 2018-01-12
> **Response to "=-=-=-= Response to the authors"**
>
> Dear Reviewer,
>
> Thank you for your response. You criticize that our revised version contained re-orderings of the text rather than substantive changes. I apologize if I was not able to address your criticisms in the way you wanted in the revised version.
>
> As far as I could tell, your criticisms of the original paper as well as the criticisms of the other reviewers hinged almost exclusively on the writing style of the paper, i.e. "central point gets lost", "relying on supplement", "too long" etc. Therefore that is exactly what I worked on addressing in the revision: the writing. I inserted summaries and added detail to introduction and conclusion as well as removing several propositions. So I am confused when you say that I "only" changed the writing when I thought that was precisely the reason you disliked the paper.
>
> I am sorry if I misinterpreted your original review. Could you explain in more detail what you are still unhappy with?
>
> Thanks,

---

> ### Public Comment · (anonymous) · 2018-04-22
> **Does this review make "any" technical comments?**
>
> I'm surprised that a review like this that makes largely "style" based comments is taken seriously by the AC.

---

> > ### Author Response · Authors · 2018-08-16
> > **This is an interesting comment**
> >
> > Dear anonymous commenter,
> >
> > I just saw your anonymous comment above (I'm the author). I've never seen a comment on OpenReview posted this long after the end of the review cycle. This suggests to me that you may be interested in the paper itself. If so, I wanted to provide you with an arxiv link to the latest version (see below) as well as an invitation to contact me via email (see also below) if you want to discuss further. In any case, I'm curious to know what motivated you to post this comment.
> >
> > https://arxiv.org/abs/1712.05577
> > george.philipp@email.de
> >
> > Best,
> > George

---

### Official Review · AnonReviewer3 · 2017-11-27

**Rating:** 5
**Confidence:** 4

**Review:**

Paper Summary:
This is a very long paper (55 pages), and I did not read it in its entirety. The first part (up to page 11), focuses on better understanding the exploding gradients problem, and challenges the fact that current techniques to address gradient explosion work as claimed. To do so, they first motivate a new measure of gradient size, the Gradient Scale Coefficient which averages the singular values of the Jacobian and takes a ratio of different layers. The motivation for this measure is that it is invariant to simple rescaling of layers that preserves the function. (I would have liked to have seen what was meant by preserved the function here -- did you mean preserve the same class outputs e.g.?)

They focus on linear MLPs in the paper for computational simplicity. With this setup, and assuming the Jacobian decomposes, they prove that the GSC increases exponentially (Proposition 5). They empirically test this out for networks 50 layers deep and 100 layers wide, where they find that some architectures have exploding gradients after random initialization, and others do not, but those that do not have other drawbacks.

They then overview the notion of effective depth for a residual network: a linear residual network can be written as a product of terms of the form (I + r_i).  Expanding out, each term is a product of some of the r_i and some of the identities I. If all r_i have a norm < 1, then the terms that dominate will be those that consist of fewer r_i, resulting in a lower effective depth. This is described in Veit et al, 2016. While this analysis was originally used for residual networks, they relate this to any network by letting I turn into an arbitrary initial function. Their main theoretical result from this is that deeper networks take exponentially longer to train (under certain conditions), which they test out with (linear?) networks of depth 50 and width 100.

They also propose that the reason gradients explode is because networks try to preserve their domain going forward, which requires Jacobians to have determinant 1 and leads to a higher Q-norm.

Main Comments:
This could potentially be a very nice paper, but I feel the current presentation is not ready for acceptance. In particular, the paper would benefit greatly from being made much shorter, and having more of the important details or proof outlines for the various propositions in the main text. Right now, it is quite confusing to follow, and I fail to see the motivation for some of the analysis. For example, the Gradient Scale Coefficient appears to be motivated because (bottom page 3), with other norm measurements, we could take any architecture and rescale the parameters, and inversely scale the gradients to make it "easy to train". But typically easy to train does not involve a specific preprocessing of gradients. Other propositions e.g. Theorem 1, proposition 6, could do with clearer intuition leading to them. I think the assumptions made in the results should also be clearer. (It's fine to have results, but currently I can't tell under what conditions the results apply and under what conditions they don't. E.g. are there any extensions of this that apply to non-linear networks?)

I also have issues with their experimental setup: why choose to experiment on networks of depth 50 and width 100? This doesn't really look anything like networks that are trained in practice. Calling these "popular architectures" is misleading.

In summary, I think this paper needs more work on the presentation to make clear what they are proving and under what conditions, and with experiments that are closer to those used in practice to support their claims.

---

> ### Author Response · Authors · 2017-12-14
> **Review Respone (2/2)**
>
> ###
>
> "I would have liked to have seen what was meant by preserved the function here -- did you mean preserve the same class outputs e.g"
>
> Yes, we mean the value of the prediction and error layers is invariant. In the revision, we have amended the text to reflect this.
>
> ###
>
> "They focus on linear MLPs in the paper for computational simplicity."
>
> When you say "linear MLPs", do you mean MLPs containing only linear layers? Note that all of the MLPs we study in this paper contain nonlinear layers (ReLU, tanh, SeLU, batch normalization and layer normalization) and many also contain skip connections. We do not study linear MLPs in this paper.
>
> ###
>
> "But typically easy to train does not involve a specific preprocessing of gradients."
>
> In the revision, we replace "easy to train" with "can be successfully trained" and make clear that this includes gradient rescaling. In the paper, we aim to contrast training difficulty that can be overcome by rescaling the gradient versus training difficulty that cannot be overcome in this way, as encapsulated by theorem 1. We agree that it is not always obvious how to scale the gradient in practice, but point out that techniques such as Adam, vSGD [4] or heuristics such as "scale the gradient to be proportial to the size of the weight matrix" are often quite successful.
>
> ###
>
> "Other propositions e.g. Theorem 1, proposition 6, could do with clearer intuition leading to them. I think the assumptions made in the results should also be clearer."
>
> We added additional explanations to the leadup of both theorem 1 and proposition 6 in the revision.
>
> Unfortunately, we were unable to include the assumptions made in theoretical results in the main body of the paper due to space reason, and because we think it would significantly detract from the readability of the paper. For example, consider the full statement of theorem 1. While some readers will be interested in this full statement, other readers may find it distracting. However, the assumptions are given and discussed in detail in sections E and F. Is there a specific section, paragraph or detail from the appendix you believe we should include in the main body?
>
> ###
>
> We hope that we have addressed your concerns in this comment and the revised version of the paper. We also refer you to our new introduction and conclusion section that make the contributions and implications of our paper even more clear. If you agree that our paper makes important contributions that are also well-supported (taking into account that we do not just use linear MLPs) we hope that you agree our paper is well-placed at ICLR. We look forward to hearing your thoughts and comments.
>
> [1] Schoenholz et al. Deep Information Propagation. ICLR 2107. https://arxiv.org/abs/1611.01232
>
> [2] Balduzzi et al. The Shattered Gradients Problem: If resnets are the answer, then what is the question?. ICML, 2017. https://arxiv.org/abs/1702.08591
>
> [3] Yang & Schoenholz. Mean field residual networks: on the edge of chaos. NIPS, 2017. https://papers.nips.cc/paper/6879-mean-field-residual-networks-on-the-edge-of-chaos
>
> [4] Schaul et al. No More Pesky Learning Rates. ICML, 2013. https://arxiv.org/abs/1206.1106
>
> [5] Saxe et al. Exact solutions to the nonlinear dynamics of learning in deep linear neural networks. 2014. https://arxiv.org/abs/1312.6120

---

> ### Author Response · Authors · 2017-12-14
> **Review Response (1/2)**
>
> Dear Reviewer,
>
> Thank you for your review. As far as I can tell, you agree that our paper makes important contributions to deep learning theory, architecture design and optimization. I just uploaded a revised version of the paper. We address the points raised in your review in this revision as well as in the comments below.
>
> ###
>
> "In particular, the paper would benefit greatly from being made much shorter, ..."
>
> In the revision, we removed some of the less central results (propositions 7 through 9) as well as high-level commentary to make the paper more focused.
>
> This paper pays attention to details that other papers often gloss over, such as the rigorous definition of exploding gradients or effective depth and the careful setting of layerwise step sizes. This rigor is what enables us to obtain important results. Also note that an important predecessor work [3] from NIPS 2017 is also 55 pages long.
>
> Much of our appendix is strictly optional for readers interested in certain specifics, such as implementation details for those interested in replicating our results or the extended related work section for those interested in pursuing research in deep learning theory. Do you see providing such details as a weak point of the paper?
>
> If you believe there are still specific results in the revision that you consider unimportant and should thus be moved to the appendix or removed entirely, please let us know.
>
> ###
>
> "... and having more of the important details or proof outlines for the various propositions in the main text."
>
> In the revision, we add a significant number of further explanations and clarifications throughout the main body of the paper.
>
> Because this paper is detail-oriented and each reader cares about a different set of details, we chose (a) to provide as much detail as possible and (b) move those details to the appendix into dedicated sections so that they would be easy to find by specific interested parties. We hope that you will find this strategy acceptable and refer you to sections E and F for theoretical details.
>
> Nonetheless, we do outline the proofs for theorems 1 and 2 in the main paper. (The proof of theorem 3 is already quite short.)
>
> If you believe there is any particular section, paragraph or detail from the appendix that should still be moved to the main body, we would be glad to know and to fulfil such a request if it could be aligned with the preferences of the other reviewers.
>
> ###
>
> "I also have issues with their experimental setup: why choose to experiment on networks of depth 50 and width 100? This doesn't really look anything like networks that are trained in practice. Calling these "popular architectures" is misleading."
>
> In the revision, we replace the phrase "popular architectures" with "architectures with popular layer types".
>
> We agree that 50-layer MLPs without skip connections are seldom used in practice. However, this is mainly because of the very pathologies explored in this paper that lead to training difficulty. We deliberately chose this high depth so that those difficulties could be clearly demonstrated on those networks. Using very deep MLPs to study gradient pathologies is a well-established practice from previous works (e.g. [1,2,3,5]). There is significant evidence that the deep learning theory community cares about those kinds of networks. Also, we believe that our networks are not that far removed from networks used in practice. For example, we use MLPs with tanh nonlinearities and MLPs with ReLU nonlinearities and batch normalization, which are popular choices.
>
> In addition to plain MLPs, we investigate MLPs with skip connections (ResNets). For ResNet, a depth of 50 is not impractical.
>
> Regarding the layer width of 100: we studied different layer widths in section 3. There was no evidence that any results presented in this paper depend on layer width in a significant way. We don't think using a width of, say, 1000, would have made a difference.
>
> We did not extend our results to convolutional networks due to space reasons, though we plan to study this case in future work. Again, many recent works also focused on MLPs.

---

### Official Review · AnonReviewer1 · 2017-11-29
**Claiming much of common intuition around tricks for avoiding gradient issues are incorrect.**

**Rating:** 8
**Confidence:** 1

**Review:**

The paper makes some bold claims. In particular about commonly accepted intuition for avoiding exploding/vanishing gradients and why all the recent bag of tricks (BN, Adam) do not actually address the problems they set out to alleviate.

This is either a very important paper or the analysis is incorrect but it's not my area of expertise. Actually understanding it at depth and validating the proofs and validity of the experiments will require some digestion. It's possible some of the issues arise from the particular architectures they choose to investigate and demonstrate on (eg I have mostly seen ResNets in the context of CNNs but they analyze on FC topologies, the form of the loss, etc) but that's a guess and there are some further analysis in the supp material for these networks which I haven't looked at in detail.

Regardless - an important note to the authors is that it's a particularly long and verbose paper, coming in at 16 pages of the main paper(!) with nearly 50 (!) pages of supplementary material where the heart and meat of the proofs and experiments reside. As such it's not even clear if this is proper for a conference. The authors have already provided several pages worth of additional comments on the website on further related work. I view this as an issue in and of itself. Being succinct and applying rigour in editing is part of doing science and reporting findings, and a wise guideline to follow. While the authors may claim it's necessary to use that much space to make their point I will argue that this length is uncalibrated to standards. I've seen many papers that need to go through much more complicated derivations and theory and remain within a 8-10 page limit by being precise and strictly to the point. Perhaps Godel could be a good inspiration here, with a 21 page PhD thesis that fundamentally changed mathematics.

In addition to being quite bold in claims, it is also somewhat confrontational in style. I understand the authors are trying to make a very serious claim about much of the common wisdom, but again, having reviewed papers for many years, this is highly unusual and it is questionable whether it is necessary.

So, while I cannot vouch for the correctness, I think it can and should go through a serious revision to make it succinct and that will likely considerably help in making it accessible to a wider readership and aligned to the expectations from a conference paper in the field.

---

> ### Author Response · Authors · 2017-12-14
> **Review Response**
>
> Dear Reviewer,
>
> Thank you for your review and for your honesty in stating that this paper does not fall within your area of expertise. I just uploaded a revised version of the paper. We address the points raised in your review in this revision as well as in the comments below.
>
> ###
>
> "an important note to the authors is that it's a particularly long and verbose paper"
>
> In the revision, we removed some of the less central results (propositions 7 through 9) as well as high-level commentary to make the paper more focused.
>
> This paper addresses many issues in detail that other papers often gloss over, such as the rigorous definition of exploding gradients or effective depth and the careful setting of layerwise step sizes. This rigor is what enables us to obtain important results. Also note that an important predecessor work [3] from NIPS 2017 is also 55 pages long.
>
> Much of our appendix is strictly optional for readers interested in certain specifics, such as implementation details for those interested in replicating our results or the extended related work section for those interested in pursuing research in deep learning theory.
>
> ###
>
> "It's possible some of the issues arise from the particular architectures they choose to investigate and demonstrate on"
>
> While we believe that all results discussed in the paper apply to convolutional and other networks in a similar fashion, we do not discuss or test the applicability to these networks specifically, for space reasons. However, using very deep MLPs as a testbed to advance the study of exploding gradients and related problems is a well-established practice (e.g. [1,2,3,4]).
>
> ###
>
> "it is also somewhat confrontational in style"
>
> I apologize if my writing style appeared confrontational. Do you mean the paragraph that starts with "These claims are mistaken. ..."? I reformulated that paragraph in the revision. It now starts with "We argue that these claims are overly optimistic..."
>
> ###
>
> "making it accessible to a wider readership and aligned to the expectations from a conference paper in the field"
>
> We do not necessarily agree that ICLR papers should appeal to a wide readership. Many program synthesis papers are targeted at those interested in program synthesis. Many machine translation papers are targeted at those interested in NLP etc. Our paper is targeted at those interested in the theory of neural networks and foundational principles of neural network architecture design. We accept that this is a subset of the entire ICLR audience and do not see anything wrong with that.
>
> [1] Schoenholz et al. Deep Information Propagation. ICLR 2107. https://arxiv.org/abs/1611.01232
>
> [2] Balduzzi et al. The Shattered Gradients Problem: If resnets are the answer, then what is the question?. ICML, 2017. https://arxiv.org/abs/1702.08591
>
> [3] Yang & Schoenholz. Mean field residual networks: on the edge of chaos. NIPS, 2017. https://papers.nips.cc/paper/6879-mean-field-residual-networks-on-the-edge-of-chaos
>
> [4] Saxe et al. Exact solutions to the nonlinear dynamics of learning in deep linear neural networks. 2014. https://arxiv.org/abs/1312.6120

---

### Author Response · Authors · 2017-11-16
**Further related work (4/4)**

+++++ ODE-based ResNets +++++


Recently, [1-4] proposed ResNet architectures inspired by dynamical systems and numerical methods for ordinary differential equations. The central claim is that these architectures are stable at arbitrary depth, i.e. both forward activations and gradients (and hence GSC) are bounded as depth goes to infinity. They propose four practical strategies for building and training ResNets: (a) ensuring that residual and skip functions compute vectors orthogonal to each other by using e.g. skew-symmetric weight matrices (b) ensuring that the Jacobian of the skip function has eigenvalues with negative real part by using e.g. weight matrices factorized as -C^TC (c) scaling each residual function by 1/B where B is the number of residual blocks in the network and (d) regularizing weights in successive blocks to be similar via a fusion penalty.


architecture             GSC (base 10 log)            GSC dilution-corrected  (base 10 log)
batch-ReLU (i)             0.337                               4.23
batch-ReLU (ii)            0.329                               4.06
batch-ReLU (iii)           6.164                              68.37
batch-ReLU (iv)            0.313                               7.22
layer-tanh (i)             0.136                               2.17
layer-tanh (ii)            0.114                               1.91
layer-tanh (iii)           3.325                               5.46
layer-tanh (iv)            0.143                               2.31
Table 1


We evaluated those strategies empirically. In table 1, we show the value of the GSC across the network for 8 different architectures in their initialized state applied to Gaussian noise (see section 9.9.2 for details). All architectures use residual blocks containing a single normalization layer, a single nonlinearity layer and a single linear layer. We initialize the linear layer in four different ways: (i) Gaussian initialization, (ii) skew-symmetric initialization, (iii) initialization as -C^TC where C is Gaussian initialized and (iv) Gaussian initialization where weight matrices in successive blocks have correlation 0.5. Initializations (ii), (iii) and (iv) mimic strategies (a), (b) and (d) respectively. To enable the comparison of the four initialization styles, we normalize each weight matrix to have a unit qm norm. We study all four initializations for both batch-ReLU and layer-tanh.

Initialization (ii) improves slightly over initialization (i). This is expected given theorem 3. One of the key assumptions is that skip and residual function be orthogonal in expectation. While initialization (i) achieves this, under (ii), the two functions are orthogonal with probability 1.

Initialization (iii) has gradients that grow much faster than initialization (i). On the one hand, this is surprising as [2] states that eigenvalues with negative real parts in the residual Jacobian supposedly slow gradient growth. On the other hand, it is not surprising because introducing correlation between the residual and skip path breaks the conditions of theorem 3.

Initialization (iv) performs comparably to initialization (i) in reducing gradient growth, but requires a larger amount of dilution to achieve this result. Again, introducing correlation between successive blocks and thus between skip and residual function breaks the conditions of theorem 3 and weakens the power of dilution.

While we did not investigate the exact architectures proposed in [2,3], our results show that more theoretical and empirical evaluation is necessary to determine whether architectures based on (a), (b) and (d) are indeed capable of increasing stability. Of course, those architectures might still confer benefits in terms of e.g. inductive bias or regularization.

Finally, strategy (c), the scaling of either residual and/or skip function with constants is a technique already widely used in regular ResNets. In fact, our study suggests that in order to bound the GSC at arbitrary depth in a regular ResNet, it is sufficient to downscale each residual function by only 1/sqrt(B) instead of 1/B as [1-4] suggest.


[1] E. Haber, L. Ruthotto, E. Holtham. Learning Across Scales - Multiscale Methods for Convolution Neural Networks. arXiv 2017. https://xtract.ai/wp-content/uploads/2017/05/Learning-Across-Scales.pdf

[2] E. Haber, L. Ruthotto. Stable Architectures for Deep Neural Networks. arXiv 2017. https://arxiv.org/abs/1705.03341

[3] B. Chang, L. Meng, E. Haber, L. Ruthotto, D. Begert, E. Holtham. Reversible Architectures for Arbitrarily Deep Residual Neural Networks. arXiv 2017. https://export.arxiv.org/abs/1709.03698

[4] Anonymous. Multi-level residual networks from dynamical systems view. ICLR 2018 submission. https://openreview.net/forum?id=SyJS-OgR-

---

### Author Response · Authors · 2017-11-16
**Further related work (3/4)**

+++++ Mean field analysis continued +++++

[3] uses a framework similar to [1,2] to propose to combat gradient growth by downscaling the weights on the residual path in a ResNet. This corresponds to increased dilution, which indeed reduces gradient growth as shown in section 7. However, we also show in proposition 10 that the reduction achievable in this way may be limited. [3] also proposes to combat the exploding gradient problem by changing the width of intermediate layers. Our analysis in section 4.4 strongly suggests that this is not effective in reducing the growth of the GSC. [3] concludes that changing the width combats the exploding gradient problem because they implicitly assume that the pathology of exploding gradients is determined by the scale of individual components of the gradient vector rather than the length of the entire vector or the GSC. They do not justify this assumption. We propose the GSC as a standard for assessing pathological exploding gradients to avoid such ambiguity.


[1] B. Poole, S. Lahiri, M. Raghu, J. Sohl-Dickstein, S. Ganguli. Exponential expressivity in deep neural networks through transient chaos. NIPS 2016. https://arxiv.org/abs/1606.05340v1

[2] S. Schoenholz, J. Gilmer, S. Ganguli, J. Sohl-Dickstein. Deep information propagation. ICLR 2017. https://openreview.net/forum?id=H1W1UN9gg

[3] Anonymous. Deep Mean Field Theory: Variance and Width Variation by Layer as Methods to Control Gradient Explosion. ICLR 2018. https://openreview.net/forum?id=rJGY8GbR-


PS: There seems to be another relevant paper: "Mean Field Residual Networks: On the Edge of Chaos" that will be published at NIPS this year. Unfortunately, I have been unable to obtain a copy so far. If you have a link to this paper, I would love to have it.

---

### Author Response · Authors · 2017-11-16
**Further related work (2/4)**

+++++ Mean field analysis continued +++++

We do not use the assumption of infinite width in our analysis. The only possible exception is that the SSD assumption in proposition 10 can be viewed as implying infinite width.

While [2] conjectures that stability is necessary for training very deep networks, our paper provides somewhat contrary evidence. Our two best performing vanilla architectures, SeLU and layer-tanh, are both inside the chaotic regime whereas ReLU, layer-ReLU and tanh, which are all stable, exhibit a higher training classification error. Clearly, chaotic architectures avoid pseudo-linearity. The difference between our experiments and those in [2] is that we allowed the step size to vary between layers. This had a large impact, as can be seen in table 2. We believe that our results underscore the importance of choosing appropriate step sizes when comparing the behavior of different neural architectures or training algorithms in general.

In section 4, we present a rigorous argument for the harmful nature of exploding gradients, and thus of chaos, at high depth.

It is not clear a priori whether a unit limit correlation is harmful for accuracy. After all, correlation information is a rather small part of the information present in the data, so the remaining information might be sufficient for learning. In section 6, we show how pseudo-linearity can arise under unit limit correlation and explain how it can harm expressivity and thus accuracy.

---

### Author Response · Authors · 2017-11-16
**Further related work (1/4)**

Dear Reviewers,

I have recently become aware of two lines of work that are quite relevant to this work: ODE-based ResNets and Mean field analysis of deep networks. I will address both these strands in the next revision of the paper, mostly in section 9 but making references throughout the main body of the paper where appropriate. Below, I give a preview (note that this is split between 3 comments).


+++++ Mean field analysis +++++

[2] and its precessor [1] are the closest works to our paper. The authors use infinitely wide networks to study the expected behavior of forward activations and gradients in the initialized state. They identify two distinct regimes, order and chaos, based on whether an infinitesimal perturbation shrinks or grows in expectation respectively as it is propagated forward. This corresponds to the expected qm norm of the layer-Jacobian being smaller or larger than 1 respectively. They show that in the chaotic regime, gradients explode whereas in the ordered regime, gradients vanish. Further, they show that for tanh MLPs the correlation between forward activations corresponding to two different data inputs converges to 1 (`unit limit correlation') in the ordered regime as activations are propagated forward and to some value less than 1 in the chaotic regime. Specifically, in a tanh MLP without biases, in the chaotic regime, the correlation converges to 0.

Like [1,2], much of our analysis relies on the expected behavior of networks in their randomly initialized state. Further, it is clear that the order / chaos dichotomy bears similarity to the exploding gradient problem / collapsing domain problem dichotomy as presented in this paper. However, there are also important differences.

- We argue in this paper that the GSC is a better measure for the presence of pathological exploding or vanishing gradients than the raw scale of the gradient. Using the GSC, we obtain very different regions of order, chaos and stability for popular architectures. For a tanh MLP with no biases, using raw gradients, order is achieved for $\sigma_w < 1$, stability for $\sigma_w = 1$ and chaos for $\sigma_w > 1$. For a tanh MLP with no biases, using the GSC, order is impossible, stability is achieved for $\sigma_w \le 1$ and chaos for $\sigma_w > 1$. For a ReLU MLP with no biases, using raw gradients, order is achieved for $\sigma_w < \sqrt{2}$, stability for $\sigma_w = \sqrt{2}$ and chaos for $\sigma_w > \sqrt{2}$. For a ReLU MLP with no biases, using the GSC, stability is inevitable.
- While [1] showed that order / chaos corresponds to unit limit correlation / non-unit limit correlation in a tanh MLP, this is not true in general. In a ReLU MLP with no biases and $\sigma_w > \sqrt{2}$, infinitesimal noise grows (chaos), yet correlation still converges to 1. Exploding gradient problem / collapsing domain problem is not a strict dichotomy and is thus able to accomodate such cases.

Similarly, the concepts of unit limit correlation and the collapsing domain problem are not the same. In fact, the former can be seen as a special case of the latter. In a tanh MLP with no bias and $\sigma_w$ slightly larger than 1, correlation converges to 0 and eventually, gradients explode. Yet the domain can still collapse dramatically in the short term as shown in figure 1 to cause pseudo-linearity. In a tanh MLP with no bias and $\sigma_w$ very large, again, correlation converges to 0 and gradients explode. However, the tanh layer maps all points close to the corners of the hypercube, which corresponds to domain collapse.

---

### Author Response · Authors · 2017-11-16
**Legend in figure 5 is incorrect**

The legend located in the top center graph in figure 5 is incorrect. From top to bottom it should be layer-tanh, batch-tanh, layer-ReLU, batch-ReLU, layer-SeLU. This colors match those in figure 3.

---

### Decision · Program_Chairs · 2018-01-29
**ICLR 2018 Conference Acceptance Decision**

**Decision:**

Invite to Workshop Track

**Comment:**

The paper sets out to analyze the problem of exploding gradients in deep nets which is of fundamental importance. Reviewers largely acknowledge the novelty of the main ideas in the paper towards this goal, however it is also strongly felt that the writing/presentation of the paper needs significant improvement to make it into a coherent and clean story before it can be published. There are also some concerns on networks used in the experiments not being close to practice.

I recommend invitation to the workshop track as it has novel ideas and will likely generate interesting discussion.